# CDKN1A is a target for phagocytosis-mediated cellular immunotherapy in acute leukemia

Awatef Allouch [1,2,3] ✉, Laurent Voisin[1,2], Yanyan Zhang[2,4], Syed Qasim Raza[1,2,5], Yann Lecluse[2,6], Julien Calvo [7,8,9,10], Dorothée Selimoglu-Buet [2,11], Stéphane de Botton[2], Fawzia Louache[2,12], Françoise Pflumio [7,8,9,10], Eric Solary [2,11] & Jean-Luc Perfettini [1,2] ✉

Targeting the reprogramming and phagocytic capacities of tumor-associated macrophages (TAMs) has emerged as a therapeutic opportunity for cancer treatment. Here, we demonstrate that tumor cell phagocytosis drives the pro-inflammatory activation of TAMs and identify a key role for the cyclin-dependent kinase inhibitor CDKN1A (p21). Through the transcriptional repression of Signal-Regularity Protein α (*SIRPα*), p21 promotes leukemia cell phagocytosis and, subsequently, the pro-inflammatory reprogramming of phagocytic macrophages that extends to surrounding macrophages through Interferon γ. In mouse models of human T-cell acute lymphoblastic leukemia (T-ALL), infusion of human monocytes (Mos) engineered to overexpress p21 (p21TD-Mos) leads to Mo differentiation into phagocytosis-proficient TAMs that, after leukemia cell engulfment, undergo pro-inflammatory activation and trigger the reprogramming of bystander TAMs, reducing the leukemic burden and substantially prolonging survival in mice. These results reveal p21 as a trigger of phagocytosis-guided pro-inflammatory TAM reprogramming and highlight the potential for p21TD-Mo-based cellular therapy as a cancer immunotherapy.

Phagocytosis of cancer cells by tumor-associated macrophages (TAMs), which are abundant in the tumor microenvironment (TME), plays a critical role in cancer immunosurveillance[1,2]. Cancer cells can evade macrophage-mediated phagocytosis by upregulating cell-surface expression of "don't eat me signals", such as CD47, PD-L1, β2M, and CD24, which bind to the phagocytosis-inhibiting receptors SIRPα, PD-1, LILRB1 and Siglec-10, respectively[1,3-6]. These interactions trigger intracellular cascades of inhibitory signals in macrophages to

---

[1]Université Paris-Saclay, Inserm UMR1030, Laboratory of Molecular Radiotherapy and Therapeutic Innovation, Villejuif F-94805, France. [2]Gustave Roussy Cancer Center, Villejuif F-94805, France. [3]NH TherAguix, Meylan F-38240, France. [4]Inserm U955, Université Paris-Est Créteil (UPEC), Créteil F-94100, France. [5]Institute of Biochemistry and Biotechnology, University of Veterinary and Animal Sciences-UVAS, Lahore, Pakistan. [6]Université Paris-Saclay, UMS 3655 CNRS / US 23 Inserm, Imaging and Cytometry Platform, Villejuif F-94805, France. [7]Université de Paris, Inserm, CEA, Stabilité Génétique Cellules Souches et Radiations, Fontenay-aux-Roses F-92260, France. [8]Université Paris-Saclay, Inserm, CEA, Stabilité Génétique Cellules Souches et Radiations, Fontenay-aux-Roses F-92260, France. [9]Equipe Niche et Cancer dans l'Hématopoièse, équipe labellisée Ligue Nationale contre le Cancer, UMR Stabilité Génétique Cellules Souches et Radiations, Fontenay-aux-Roses F-92260, France. [10]OPALE Carnot Institute, The Organization for Partnerships in Leukemia, Saint-Louis Hospital, Paris F-75010, France. [11]Université Paris-Saclay, Inserm UMR1287, Hematopoietic stem cells and the development of myeloid malignancies, Villejuif F-94805, France. [12]Université Paris-Saclay, Inserm UMR-S-MD1197, Hôpital Paul Brousse, Villejuif F-94800, France. ✉e-mail: awatef.allouch@gustaveroussy.fr; jean-luc.perfettini@gustaveroussy.fr

block cytoskeletal rearrangements, phagocytic synapse formation, and cancer cell engulfment[1,3–7]. Several immunotherapies aim to disrupt these interactions through the use of macrophage immune checkpoint (MIC) blockers (such as blocking antibodies against CD47 or antagonistic engineered SIRPα variants) to circumvent signaling that inhibits phagocytosis, enabling macrophages to engulf and clear cancer cells[1,3–6,8,9]. However, several tumors exhibit resistance to these prophagocytic therapeutic modalities, revealing the existence of still unknown phagocytosis-regulating mechanisms that can act at the levels of both cancer cells and TAMs.

In addition to the impairment in their phagocytic functionalities, TAMs demonstrate an immunosuppressive phenotype that contributes to tumor progression, treatment resistance, and poor clinical outcomes[10,11]. Due to their functional plasticity, these myeloid cells can be reprogrammed to acquire a proinflammatory phenotype and promote tumor clearance[10]. Several therapeutic approaches targeting TAMs to alleviate their immunosuppressive properties or promote their phagocytic capacities to engulf tumor cells have shown therapeutic benefits in several tumor types[12–15], but therapeutic strategies aimed at promoting both the tumor cell phagocytosis and proinflammatory reprogramming of TAMs are rarely considered therapeutic options for cancer treatment. Nevertheless, human macrophages genetically engineered to express a chimeric antigen receptor (CAR) specific for HER2 were recently shown to overcome the resistance of solid tumors to macrophage phagocytosis and to support a T cell-based antitumor immune response through the conversion of immunosuppressive TAMs into a proinflammatory phenotype[16]. Although the proinflammatory reprogramming of transduced macrophages was induced by an adenoviral vector, regardless of CAR expression and tumor cell phagocytosis induction[16], CAR macrophage-based therapy reveals that overcoming both tumor cell phagocytosis resistance and TAM immunosuppressive functions sustains the development of an effective antitumor immune response. A better understanding of the interplay between the abrogation of tumor cell phagocytosis resistance and the proinflammatory functional reprogramming of TAMs is urgently needed and should pave the way for the development of antitumor myeloid cell-based therapies.

In the present study, we reveal a non-cell-autonomous modality of macrophage proinflammatory activation that starts with the phagocytosis of live cancer cells. We identify CDKN1A (p21) as a transcriptional repressor of the phagocytosis inhibitory receptor SIRPα and consequently as a therapeutic target for overcoming the CD47-dependent phagocytosis resistance mechanism of cancer cells (such as that of T-cell acute lymphoblastic leukemia (T-ALL) cells) and for inducing the proinflammatory activation of macrophages after cancer cell engulfment. Finally, using preclinical mouse models of human T-ALL, we demonstrate that human monocytes (Mos) engineered to express p21 could be used to home the tumor bed, to promote the phagocytosis of cancer cells after Mo differentiation into TAMs, to subvert anti-inflammatory TAMs and to reduce tumor growth. These results show that phagocytosis-guided Mo-based cellular therapy has the potential to significantly improve the outcomes of patients with T-ALL.

## Results

### Tumor cell phagocytosis triggers the proinflammatory activation of macrophages

To decipher the interplay between the abrogation of tumor cell phagocytosis resistance mechanisms and the proinflammatory activation of macrophages, we first characterized the molecular mechanisms involved in the regulation of tumor cell phagocytosis by macrophages. Human blood monocyte-derived macrophages (MDMs), which possess an anti-inflammatory phenotype[17], were cocultured with several leukemia cell lines (Fig. 1a). MDMs rapidly engulfed Jurkat T cells

(Fig. 1b) and delivered them to lysosomal compartments for degradation, as visualized by confocal microscopy (Supplementary Fig. 1a). MDMs demonstrated phagocytic activity toward two other T-ALL cell lines (MOLT4 and CEM) without engulfment of primary peripheral blood (PB) lymphocytes (PBLs) (Fig. 1c). Under these coculture conditions, MDMs also engulfed acute myeloid leukemia (AML) cells, such as the THP1 cell line (Fig. 1c) and primary AML blasts (Fig. 1d) (Supplementary Table 1), but failed to engulf erythroleukemia HEL cells and chronic myelogenous leukemia K562 cells (Fig. 1c), indicating that MDMs preferentially engulf acute leukemia cells. In accordance with the previously demonstrated role of ROCK-dependent signaling pathways in phagocytosis[18], MDM engulfment of leukemia cells and cell lines may depend on ROCK activity, as the pharmacological inhibition of ROCK kinase activity with Y27632 (as indicated by inhibition of myosin light chain 2 (MLC2) phosphorylation on serine 19 (MLC2S19*) (Supplementary Fig. 1b)) abolished Jurkat cell phagocytosis (Supplementary Fig. 1c). Neither a broad-spectrum caspase inhibitor (ZVAD) (Fig. 1c, d) nor recombinant human annexin V (Supplementary Fig. 1d, e), which impair apoptosis and the engulfment of phosphatidylserine-exposing and dying cells (as shown in cisplatinum (CDDP)-treated cells (Supplementary Fig. 1d, e)), respectively, increased or reduced the engulfment of Jurkat cells and primary AML blasts by macrophages, thus excluding the possibility that the detection of tumor cell phagocytosis was associated with the clearance of dying tumor cells by efferocytosis[19]. We also excluded the role of target cell geometry since no correlation was identified between the phagocytosis of leukemia cells and target cell volume and surface area, as determined by confocal microscopy (Supplementary Fig. 1f, g). Despite the fact that we did not observe a negative correlation between the expression of the major "don't eat me signal" CD47 on the surface of leukemia cells[13] and their phagocytosis (Supplementary Fig. 1h, i), the reduction in CD47 cell-surface expression in MOLT4 cells achieved through stable lentiviral transduction of specific CD47 CRISPR guide RNAs (gRNAs) and the CAS9 gene (CrCD47 MOLT4) (Supplementary Fig. 2a–d) significantly enhanced the percentage of phagocytosis and the efficiency of leukemia cell engulfment compared to the control expression in MOLT4 cells transduced with control lentiviral vectors (CrCo.) (Supplementary Fig. 2e–g), thus demonstrating that CD47 is a key determinant in the repression of leukemic T-cell phagocytosis.

To explore the functional impact of leukemia cell phagocytosis on the MDM phenotype, we cultured MDMs with MOLT4 cells for 2 h before sorting phagocytic (Phago⁺ MDMs) and nonphagocytic (Phago⁻ MDMs) cells. Phago⁺ MDMs (Fig. 1e-g) showed complete degradation of engulfed cells at 96 h after cell sorting (Fig. 1h, i). Gene expression analysis indicated that after target cell degradation, Phago⁺ MDMs underwent proinflammatory activation (Fig. 1j and Supplementary Data 1), as indicated by the upregulation of 20 genes and downregulation of 16 genes, which were previously associated with proinflammatory or anti-inflammatory activation, respectively, identified by comparison with Phago⁻ MDMs (Fig. 1j and Supplementary Data 1). Consistently, Phago⁺ MDMs demonstrated decreased expression of the cell-surface scavenger receptor CD163 (Fig. 1k), increased expression of the transcription factor Interferon regulatory factor 5 (IRF5, Fig. 1l), and enhanced release of proinflammatory cytokines, including IL1β, IL6, IL8, IFNγ, SERPIN E1, GM-CSF, IL23, Gro-α, IL1-ra, MIF, and IL27 (Fig. 1l-n and Supplementary Fig. 3a) compared with Phago⁻ MDMs. Similarly, Phago⁺ MDMs that had engulfed primary AML blasts demonstrated increased secretion of IL8 (Supplementary Fig. 3b).

Considering that Interferon γ (IFNγ), which is secreted by Phago⁺ MDMs (Fig. 1m, n), could convert immunosuppressive TAMs into immunostimulatory macrophages[20,21], we next explored the ability of secreted IFNγ to promote the proinflammatory reprogramming of surrounding Phago⁻ MDMs. Phago⁺ MDMs in the upper chamber of Transwell devices were cocultured with Phago⁻ MDMs in the bottom

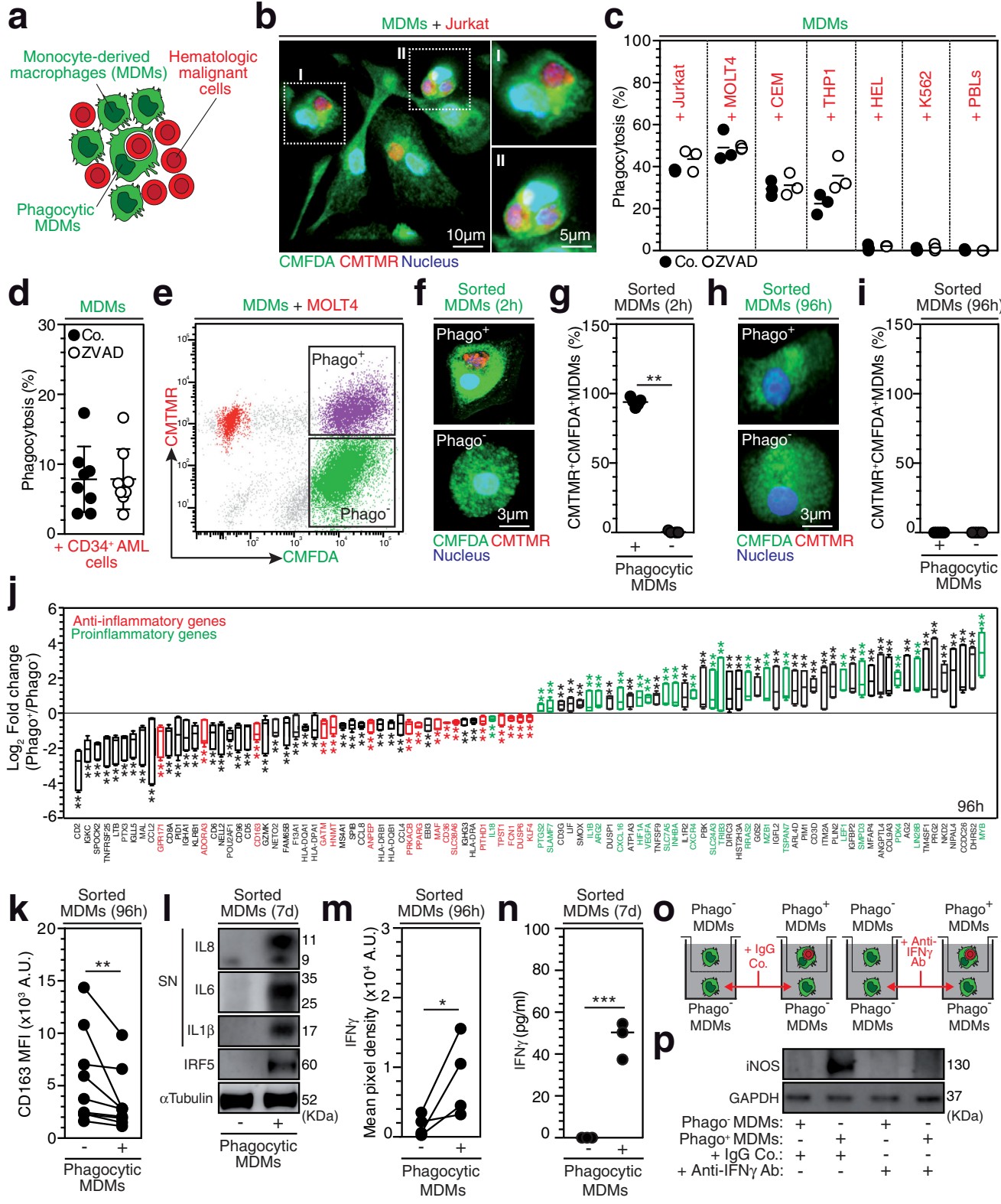

chamber in the absence or presence of control IgG or anti-IFNγ blocking antibodies for 15 d (Fig. 1o). While increased expression of the proinflammatory factor inducible oxide synthase (iNOS) was detected in Phago⁻ MDMs in the presence of control IgG, this effect was lost in the presence of the anti-IFNγ blocking antibodies (Fig. 1p), demonstrating that Phago⁺ MDMs could polarize bystander anti-inflammatory MDMs into a proinflammatory phenotype through the secretion of IFNγ.

## p21 acts as a master regulator of tumor cell phagocytosis through SIRPα transcriptional repression

Considering (i) the shared cellular features between macrophages and senescent cells[22], (ii) the ability of senescent cells to phagocytose live tumor cells[23], and (iii) the pivotal roles of p21 in the senescence[24], terminal differentiation, and survival of macrophages[25–28], we assessed the role of p21 to identify molecular mechanisms regulating the macrophage-mediated phagocytosis of leukemia cells. Using specific

**Fig. 1 | Tumor cell phagocytosis triggers the proinflammatory activation of macrophages. a** Schematic representation of the coculture of CMFDA-labeled MDMs with CMTMR-labeled malignant hematologic cells. **b** Confocal micrograph of MDMs and Jurkat cells after 8 h of coculture. **c, d** Percentage of phagocytosis of leukemia cells or PBLs (**c**) or CD34⁺ AML cells (**d**) in control (Co.)- or ZVAD (100 μM)-treated cocultures. **e** FACS dot plot of Phago⁺ MDMs or Phago⁻ MDMs sorted after 2 h of coculture with MOLT4 cells. **f, g** Confocal micrographs and percentages of CMTMR⁺CMFDA⁺ MDMs at 2 h (**f** p = 0.0079) (**f, g**) and 96 h (**h, i**) after Phago⁺ MDMs and Phago⁻ MDMs sorting. **j–n** Phago⁺ MDMs were analyzed in comparison to Phago⁻ MDMs to characterize modulated genes by a microarray (**f** p = 0.0022, *p = 0.0152) (**j**), CD163 membrane expression by FACS (**f** p = 0.0039) (**k**), and IRF5 expression by western blot (WB) analysis (**l**) and the supernatant (SN) was evaluated for indicated proinflammatory cytokines by WB analysis (**l**) or for IFNγ by a cytokine microarray (*p = 0.0286) (**m**) or ELISA (***p = 0.0008) (**n**) at 96 h (**j, k, m**) and 7 d (**l, n**) after FACS sorting. **o, p** Transwell coculture model of Phago⁺ MDMs and Phago⁻ MDMs at 2 h after FACS sorting (**o**), and iNOS expression identified by WB analysis of Phago⁻ MDMs cocultured in the bottom chambers for 15 d (**p**). In **b, l, p** and **e, f, h**, the data are representative of n = 3 and n = 5 donors. In **c, n, d**, and **g, i**, the data are presented as the mean±SEM from n = 3, n = 6, and n = 5 donors. In **d**, the CD34⁺ cells were from n = 4 AML patients. In **j**, box plots show centre line as median, box limits as upper and lower quartiles, and whiskers as a minimum to maximum values, from n = 3 donors. In **k** and **m**, the data are donor matched from n = 9 and n = 4 donors. Exact p-values are indicated and determined with two- (**g**) or one-tailed (**m**) unpaired Mann-Whitney test, the Kolmogorov–Smirnov test (**j**), a two-tailed paired Wilcoxon test (**k**), and a two-tailed unpaired t test (**n**). Source data are provided as a Source Data file.

small interfering RNA (siRNA), MDMs were depleted of p21 (Fig. 2a) and analyzed for phagocytic activity. Interestingly, p21 depletion strongly reduced the phagocytosis of Jurkat cells (Fig. 2b, c), MOLT4 cells (Fig. 2d), and primary human AML blasts (Fig. 2e), without affecting the cell cycle progression, proliferation or viability of MDMs (Supplementary Fig. 4a–e). We also noticed that p21 depletion did not alter the phagocytosis of pHrodo Green⁺ bacterial *Escherichia coli* bioparticles (Fig. 2f). These results suggest that p21 is a key regulator of the MDM-mediated phagocytosis of leukemia cells. Consistently, knockdown of the transcription factor p53 in MDMs reduced the expression of p21 and significantly impaired the phagocytosis of MOLT4 cells without affecting MDM viability (Supplementary Fig. 5a–c), suggesting that the spontaneous phagocytosis of leukemia cells is under the control of p53-dependent p21 expression. Conversely, increased p21 expression (Supplementary Fig. 5d, g, i) induced by treating MDMs with phorbol myristate acetate (PMA)[29], the histone deacetylase (HDAC) inhibitor MS275[30] or immobilized immunoglobulins (intravenous immunoglobulins, IVIg), which bind macrophage Fcγ receptors and upregulate p21 expression[31], significantly enhanced the phagocytosis of Jurkat cells and MOLT4 cells (Supplementary Fig. 5e, f, h, j), confirming that p21 regulates the MDM-mediated phagocytosis of leukemia cells. Accordingly, silencing or upregulating p21 expression in MDMs through transduction of lentiviral vectors encoding short hairpin RNA (shRNA) targeting the 3′ untranslated region of p21 (sh3′UTRp21) or p21 cDNA (p21TD) resistant to sh3′UTRp21, respectively, impaired or enhanced the phagocytosis of MOLT4 cells, as indicated by comparison to control MDMs (Co.TD) (Supplementary Fig. 5k, l). Importantly, the exogenous expression of p21 cDNA in p21-depleted MDMs (p21TD + sh3′UTRp21) restored the phagocytic activity of these complemented MDMs as compared to that of control or p21-depleted cells (Co.TD or sh3′UTRp21) (Supplementary Fig. 5k, l), further corroborating the specific key role of p21 in promoting the phagocytosis of leukemia cells by macrophages. To elucidate the molecular mechanisms regulating the p21-dependent phagocytosis of leukemia cells, p21-overexpressing (p21TD) or control (Co.TD) MDMs were cocultured with control (CrCo.) or CD47-depleted (CrCD47) MOLT4 cells and analyzed to assess phagocytosis. The coculture of p21TD-MDMs with CrCo.MOLT4 cells exhibited significantly increased MDM phagocytosis of MOLT4 cells, while the cocultures of Co.TD-MDMs or p21TD-MDMs with CrCD47MOLT4 cells did not show significant enhancement in the phagocytosis of leukemia cells (Fig. 2g). Accordingly, coculturing p21TD-MDMs with MOLT4 cells in the presence of anti-CD47 blocking antibodies did not increase the phagocytosis of MOLT4 cells compared to coculturing p21TD-MDMs with MOLT4 cells in the presence of control IgG or coculturing Co.TD-MDMs with MOLT4 cells in the presence of anti-CD47 blocking antibodies (Supplementary Fig. 6). Altogether, these results suggest that p21 overexpression enables the phagocytosis of MOLT4 cells through the disruption of the antiphagocytic CD47-SIRPα axis in phagocytic macrophages.

To test this hypothesis, we evaluated the impact of p21 modulation on the expression of SIRPα, which is the phagocytosis inhibitory receptor of CD47 expressed on myeloid cells[32]. Depletion of p21 in macrophages increased SIRPα protein expression and mRNA levels (Fig. 2h–j and Supplementary Fig. 7a), indicating that p21 downregulates SIRPα through inhibition of its transcription. Reciprocally, upregulation of p21 through transduction of MDMs with a p21-expressing lentiviral vector (p21TD) (Fig. 2k) or treatment with PMA (Supplementary Fig. 7b) decreased the protein expression of SIRPα protein. A decrease in SIRPα cell-surface expression was detected until 15 d after lentiviral transduction (Supplementary Fig. 7c). Consistently, SIRPα mRNA expression was decreased in p21-transduced (p21TD) macrophages compared with control-transduced (Co.TD) macrophages (Fig. 2l, m), further indicating that p21 represses the transcription of SIRPα. Various mechanisms have been shown to account for the ability of p21 to negatively regulate gene transcription[17,33,34]. Using chromatin immunoprecipitation (ChIP) qPCR assays, p21 was detected at the SIRPα promoter in MDMs, and p21 overexpression was observed to significantly increase its binding to the SIRPα promoter (Fig. 2n). When MDMs with p21 knockdown (Fig. 2o) or overexpression (Fig. 2p) were transduced with a lentiviral vector expressing a luciferase reporter gene under the control of the endogenous macrophage *SIRPα* promoter, we observed an increase or a decrease in *SIRPα* promoter-dependent luciferase activity, respectively, further demonstrating that p21 could repress *SIRPα* gene transcription. To evaluate the impact of the p21-SIRPα axis on the phagocytosis of leukemia cells, MOLT4 cells stably expressing the fluorescent *mCherry* reporter gene (mCherry⁺MOLT4 cells) were cocultured with MDMs generated from genetically engineered human Mos transduced with an empty lentiviral vector (Co.TD-Mos), a *CDKN1A*-expressing lentiviral vector (p21TD-Mos), a *SIRPα*-expressing (SIRPαTD-Mos) lentiviral vector or a combination of the lentiviral vectors (p21 + SIRPαTD-Mos). The p21-mediated repression of *SIRPα* was overcome by using a promoter that was distinct from endogenous macrophage *SIRPα* promoter and did not respond to p21 repression. As expected and without affecting the cell cycle progression, proliferation, or viability of transduced MDMs (Supplementary Fig. 8a–e), enforced p21 expression decreased the endogenous expression of the SIRPα protein (Fig. 2q) and enhanced the phagocytosis of mCherry⁺ MOLT4 cells (Fig. 2r, s), while enforced expression of SIRPα (Fig. 2q) inhibited phagocytosis (Fig. 2r, s). Most importantly, the transduction of a *SIRPα* cDNA sequence that was insensitive to p21 repression abrogated the phagocytosis of mCherry⁺ MOLT4 cells by p21-overexpressing macrophages (Fig. 2q–s), demonstrating that p21-induced phagocytosis depends on SIRPα biological activity. Moreover, p21 was unlikely to be involved in the proinflammatory reprogramming of MDMs following leukemia cell engulfment, since p21 overexpression and p21 knockdown failed to modulate the expression of CD163 (Supplementary Fig. 8f) and IRF5 (Supplementary Fig. 8g), respectively.

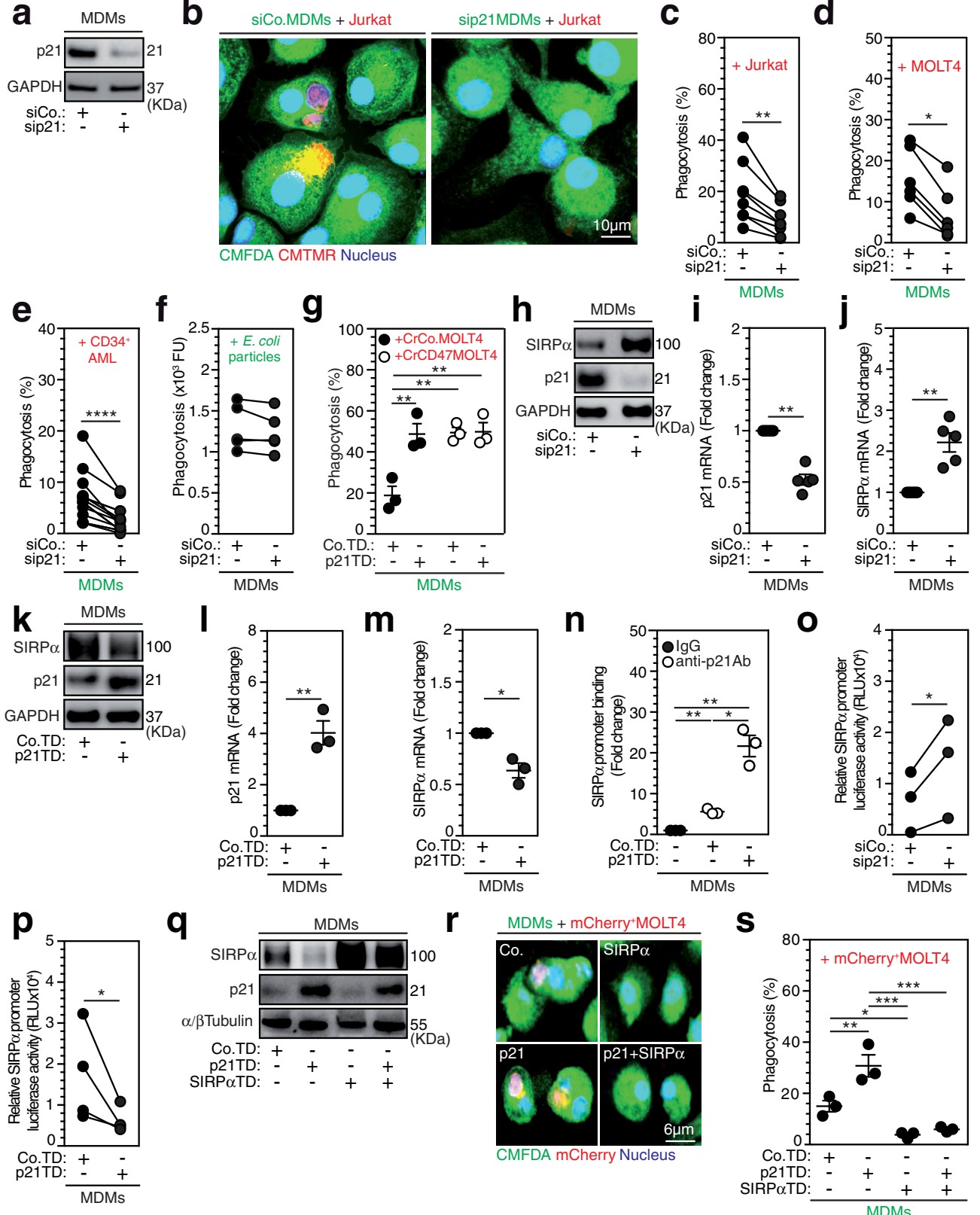

**In situ differentiation of adoptively transferred p21-engineered Mos into phagocytosis-proficient TAMs reduces the tumor burden and prolongs the survival of T-ALL-bearing NSG mice**

To explore how the p21-mediated modulation of MDM phagocytic activity toward leukemia cells could be used therapeutically, we first showed the adoptive transfer of CFSE-labeled human Mos into total

body-irradiated NOD.Cg-*Prkdc^scid^Il2rg^tm1Wjl^/SzJ* (NSG) mice generated CFSE+ anti-inflammatory macrophages, which were detected in the bone marrow (BM) and the spleen within 7 d after Mo adoptive transfer (Supplementary Fig. 9a–c, f). mCherry+ MOLT4 cells were engrafted in NSG mice to develop a mouse model of human T-ALL, which was confirmed by the presence of mCherry+ cells in the PB by d 21 after

**Fig. 2 | Macrophage-expressed p21 governs the phagocytosis of leukemia cells through the repression of SIRPα transcription. a** p21 expression in control (siCo.) or p21-silenced (sip21) MDMs after 24 h silencing. **b** Confocal micrograph of siCo. or sip21 CMFDA⁺ MDMs cocultured with CMTMR⁺ Jurkat cells. **c**–**e** Percentages of Jurkat (**c**), MOLT4 (**d**) or patient CD34⁺ AML (**e**) phagocytosis by siCo. or sip21 MDMs (**p = 0.0078, *p = 0.0312, ****p = 6.1e-5). **f** Phagocytosis (Fluorescence Units (FU)) of pHrodo Green *E. coli* bioparticle by siCo. or sip21 MDMs. **g** Percentages of CrCD47MOLT4 or CrCo.MOLT4 phagocytosis by MDMs transduced with control (Co.TD) or p21-expressing (p21TD) lentiviral vectors (LVs) after 72 h of transduction (**p = 0.0061, **p = 0.0053, **p = 0.0048). **h**–**m** SIRPα and p21 proteins (**h, k**) or mRNAs (**i, j, l, m**) in siCo. or sip21 MDMs (**p = 0.0024, **p = 0.0020) (**h**–**j**) or in Co.TD-MDMs or p21TD-MDMs (**p = 0.0063, *p = 0.0291) (**k**–**m**). **n** ChIP-qPCR assays performed with control MDMs, Co.TD-MDMs (**p = 0.0016) or p21TD-MDMs (**p = 0.0017, *p = 0.0194) immunoprecipitated with control IgG or anti-p21 antibodies and analyzed by qPCR at SIRPα promoter. **o, p**

Luciferase activities in siCo. or sip21 MDMs (**o**) or in Co.TD-MDMs or p21TD-MDMs (**p**) transduced with a lentiviral vector expressing a *luciferase* reporter gene under the control of the SIRPα promoter (*p = 0.0431, *p = 0.0458). **q**–**s** MDMs derived from monocytes transduced with Co.TD and/or p21TD LVs and/or SIRPα-expressing LVs (SIRPαTD) assessed for the indicated proteins (**q**) and for phagocytosis of mCherry⁺ MOLT4 cells (**p = 0.0079, *p = 0.0472, ***p = 0.0002, ***p = 0.0004) (**r, s**). In **a, b, h, k, q** and **r**, the data are representative of n = 3 donors. In **c, d, e** and **o, p**, the data are donor matched from n = 8, n = 6, n = 9, n = 4 and n = 3 donors. In **e**, the CD34⁺ cells were from n = 6 AML patients. In **i, j** and **g, l, m, n, s**, the data are presented as the mean±SEM from n = 5 and n = 3 donors. Exact p-values are indicated and determined with two-tailed paired Wilcoxon (**c, d, e**), two- (**i, j, l, n**) or one-tailed (**m**) ratio-paired t, and one-tailed paired t (**o, p**) tests and two-way ANOVA with Sidak's (**g**) or one-way ANOVA Tukey's (**s**) multiple comparison tests. Source data are provided as a Source Data file.

engraftment (Supplementary Fig. 9d, e, and g) and body weight loss, BM invasion and marked splenomegaly by d 35 after engraftment (Supplementary Fig. 9h, i). Using lentiviral vectors encoding a p21 cDNA sequence or control vectors that encoded a GFP reporter gene (Co-GFPTD or p21-GFPTD), we revealed that human Mos were effectively transduced and that GFP expression was stably detected after Mo differentiation into macrophages until 30 d after lentiviral transduction (Supplementary Fig. 10). We then adoptively transferred CFSE⁺ p21TD-Mos or Co.TD-Mos into tumor-free NSG mice. No toxic effect was observed after the adoptive transfer of the engineered Mos (Supplementary Fig. 11a, b). p21TD-Mo- or Co.TD-Mo-derived CFSE⁺ MDMs were equally distributed in the BM and spleen of NSG mice, without being detected in PB or liver (Supplementary Fig. 11c-e). More than 96% of the CFSE⁺ MDMs detected in the spleen expressed both human leucocyte (hCD45) and macrophage (hCD68) markers (Supplementary Fig. 11f, g). After 72 d, FACS-sorted Co.TD-MDMs or p21TD-MDMs from BM and spleen exhibited integrated lentiviral copy numbers (4 to 6 lentiviral vector copies per MDM) similar to those of MDMs transduced and differentiated in vitro over 7 d (Supplementary Fig. 12a, b), indicating that Co.TD-MDMs and p21TD-MDMs stably persisted for at least 72 d in NSG mice. Therefore, we then transferred genetically engineered CFSE⁺ p21TD-Mos or Co.TD-Mos into NSG mice engrafted with mCherry⁺ MOLT4 cells (Fig. 3a). Interestingly, p21TD-Mo adoptive transfer prevented body weight loss (Fig. 3b, c) and splenomegaly development (Fig. 3d, e) in the mCherry⁺ MOLT4 cell-engrafted NSG mice. These clinical parameters were associated with significant reductions in the leukemic burdens in the PB, BM, spleen and liver of p21TD-Mo-treated mice compared with those in the same tissues of Co.TD-Mo-treated mice (Fig. 3f-j and Supplementary Fig. 13a). Consistently, a significant increase in survival was also observed in p21TD-Mo-treated leukemia-bearing mice (Fig. 3k). To exclude any impacts of different steps of the cell manufacturing process on the antitumor effect observed, prior to engraftment of mCherry⁺ MOLT4 cells, NSG mice were not treated with adoptive transfer or infused with control untransduced human monocytes (UTD-Mos); human monocytes treated with Vpx viral-like particles (Vpx-Mos), which were used to enhance the lentiviral transduction efficiency of myeloid cells[35,36]; p21TD-Mos; or Co.TD-Mos. We observed that only mice infused with p21TD-Mos showed significant prolongation of survival (Supplementary Fig. 13b). Altogether, these results demonstrated that the adoptive transfer of p21TD-Mos strongly reduced the progression of leukemia.

To decipher the biological mechanisms involved in the beneficial effect of p21TD-Mos, mCherry⁺ MOLT4 cell-engrafted NSG mice (established as shown in Fig. 3a) were treated with liposomes containing clodronate or liposomes without clodronate 21 d after the adoptive transfer of p21TD-Mos or Co.TD-Mos. At 24 h after treatment, clodronate had selectively depleted p21TD-Mo- or Co.TD-Mo-derived CFSE⁺ macrophages (Supplementary Fig. 14a–c), and the clodronate-

containing liposomes strongly reduced the survival of p21TD-Mo-treated NSG mice (Fig. 3l). These results further demonstrate the key role of p21TD-Mo-derived macrophages in the inhibition of leukemia progression. To address the impact of p21-mediated SIRPα repression on disease regression in vivo, p21TD-Mos, SIRPαTD-Mos, p21+SIRPαTD-Mos or Co.TD-Mos were engineered (as shown in Fig. 2q) and transferred into NSG mice engrafted with mCherry⁺ MOLT4 cells (Fig. 3a). As expected, while p21TD-Mo-treated NSG mice lived longer, a significant decrease in survival was observed for mice that received SIRPαTD-Mos (Fig. 3m). In agreement with previous results (Fig. 2r, s), exogenous expression of a *SIRPα* cDNA sequence resistant to p21 repression in p21TD-Mos abrogated the positive effect of engineered p21TD-Mo transfer on mouse survival (Fig. 3m), further demonstrating the role of p21-mediated SIRPα repression in the regulation of MDM function. The decreased survival of mice observed after depletion of Co.TD-Mo-derived macrophages with clodronate-containing liposomes (Fig. 3l) and adoptive transfer of SIRPαTD-Mos (Fig. 3m) confirmed our in vitro results (Figs. 1 and 2) and demonstrated that the lentiviral transduction of p21 potentiated the intrinsic capacity of anti-inflammatory macrophages to engulf leukemia cells in vivo. To investigate the antitumor effect of the combination of p21TD-Mo-based cellular therapy with CD47 blockade[37], mCherry⁺ MOLT4 cell-engrafted NSG mice were treated with anti-CD47 antibodies 15 d after adoptive transfer of p21TD-Mos, SIRPαTD-Mos, p21+SIRPαTD-Mos or Co.TD-Mos. The anti-CD47 antibodies significantly prolonged the survival of treated mice, with the exception of the mice infused with p21TD-Mos, which did not show an additional enhancement of survival compared with that of the corresponding control mice (Fig. 3n). These results were confirmed by adoptively transferring p21TD-Mos or Co.TD-Mos into NSG mice engrafted with CD47-depleted (CrCD47MOLT4) or control (CrCo.MOLT4) MOLT4 cells (Supplementary Fig. 15). Altogether, these results highlighted that adoptive transfer of p21TD-Mos promotes the phagocytosis of CD47-expressing leukemia cells and significantly inhibits leukemia progression through the abrogation of the antiphagocytic CD47-SIRPα axis mediated by repressing SIRPα expression in TAMs differentiated from p21TD-Mos.

## Phagocytic p21-engineered TAMs undergo proinflammatory activation and reprogram surrounding TAMs into proinflammatory macrophages

Considering the proinflammatory fate of Phago⁺ MDMs detected in vitro (Fig. 1 and Supplementary Fig. 3), we sought to investigate this skewing in vivo and its relevance to p21TD-Mo-based therapy. Twenty-one days after engineered CFSE⁺ Mos were adoptively transferred, CFSE⁺ Co.TD-Mo- or p21TD-Mo-derived macrophages engulfing mCherry⁺ MOLT4 cells were detected in the spleen of treated mice (Fig. 4a). More importantly, flow cytometric (on d 21) and fluorescence microscopy (on d 21 and 35) analyses of FACS-sorted nonphagocytic CFSE⁺ (Phago⁻) or phagocytic (Phago⁺) CFSE⁺mCherry⁺ MDMs from the

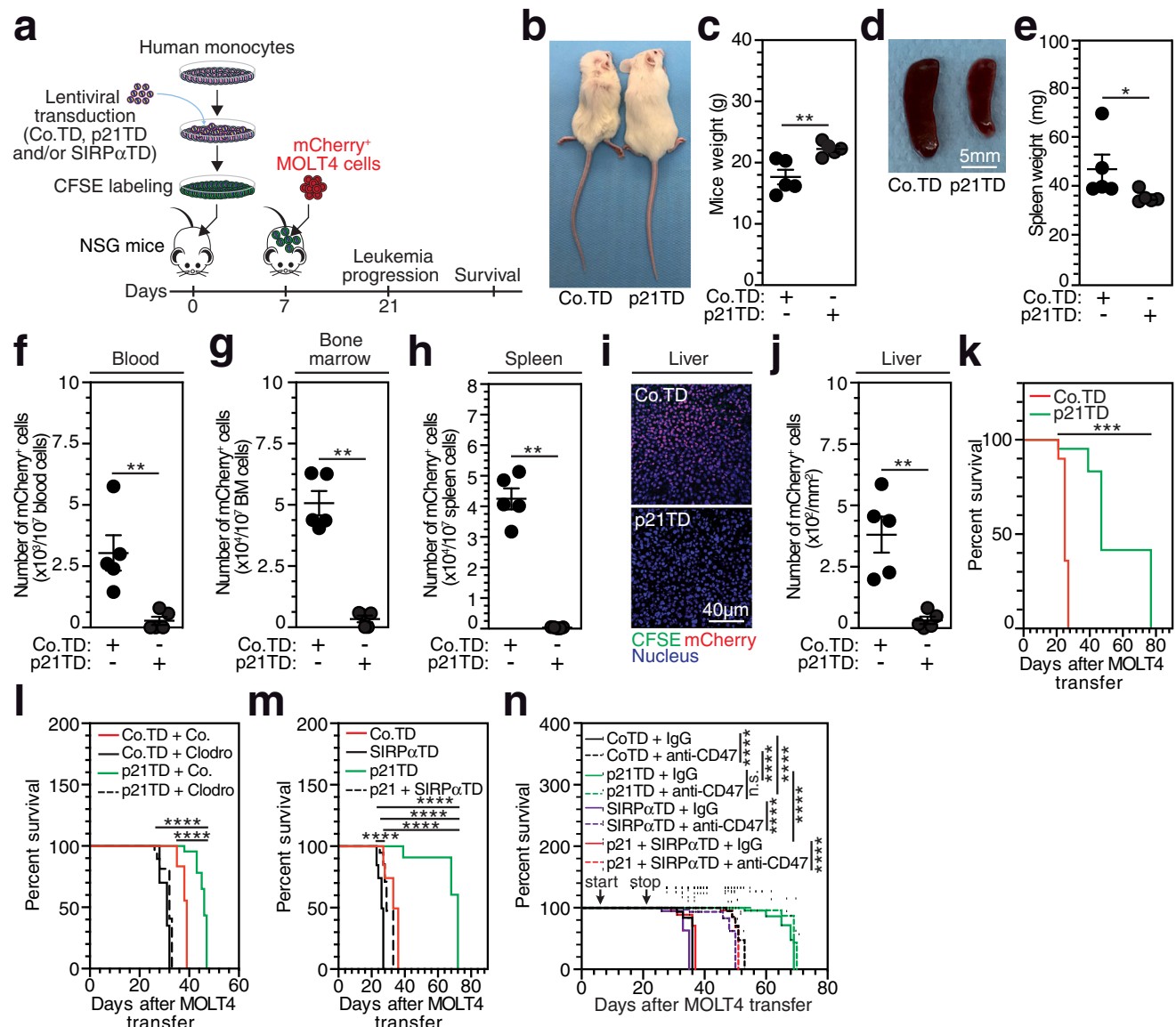

**Fig. 3 | Prophylactic adoptive transfer of p21TD-Mos decreases the leukemic burden and prolongs mouse survival in a model of human T-ALL. a** Schematic representation showing prophylactic adoptive transfer of CFSE-labeled control (Co.TD), p21 (p21TD) and/or SIRPα (SIRPαTD) genetically engineered human monocytes (Mos) into NSG mice previously treated with total body irradiation (TBI), which were then engrafted with mCherry⁺ MOLT4 cells 7 days later. **b–k** Engrafted mice that received the indicated monocytes were assessed at 21 d to measure body (**p = 0.0079) (**b**, **c**) and spleen (*p = 0.00317) weights (**d**, **e**); the leukemic burdens in the blood, bone marrow (BM) and spleen by FACS (**p = 0.0079) (**f**, **g**, **h**) and in liver tissues by confocal microscopy (**p = 0.0079) (**i**, **j**); and survival (***p = 0.0003) (**k**). **l** Survival of engrafted mice that received indicated monocytes and were treated 21 d after monocyte transfer with control (Co.) or clodronate (Clodro)-containing liposomes (***p = 4.8e-9, ****p = 0.1.1e-5). **m** Survival of engrafted mice that received the indicated monocytes (****p = 3.2e-5, ****p = 7e-10, ****p = 2.1e-7, ****p = 1e-6). **n** Survival of engrafted mice that received Co.TD-Mos, p21TD-Mos, SIRPαTD-Mos or p21+SIRPαTD-Mos and were treated on d 15 after Mo transfer for 14 d with daily injections of isotype control (IgG) or anti-CD47 blocking antibodies (100 μg/mouse) (****p = 9.4e-12, p = 0.0942 (n.s.), ****p = 2.9e-7, ***p = 2.7e-6, ****p < 1e-15, ***p = 4.5e-14, ***p = 9e-10). In **b**, **d** and **i**, the data are representative of n = 5 mice/group. In **c**, **e**, **f**, **g**, **h** and **j**, the data are presented as the mean±SEM from n = 5 mice/group. In **k**, **m**, **n** and **l**, the survival data are from n = 5 and n = 6 mice/group. Exact p-values are indicated and deter- mined with two-tailed unpaired Mann-Whitney (**c**, **e**, **f**, **g**, **h**, **j**) and log-rank Mantel−Cox (**k**, **l**, **m**, **n**), tests. Source data are provided as a Source Data file.

spleen and the BM of treated mice showed, respectively, that Phago⁺ MDMs exhibited decreased membrane expression of the scavenger receptor CD163 (compared with Phago⁻ MDMs, Fig. 4b) and highly expressed iNOS (Fig. 4c). These results confirmed that in vivo, pha- gocytosis of leukemia cells promoted the activation of MDMs toward a proinflammatory phenotype. In contrast to MDMs sorted from tumor-free NSG mice infused with Co.TD-Mos or p21TD-Mos (Fig. 4d), Phago⁻ MDMs sorted from mCherry⁺ MOLT4 cell-engrafted NSG mice injected with p21TD-Mos exhibited a significant increase in iNOS expression on d 35 (Fig. 4e), suggesting that Phago⁺ MDMs can

support the proinflammatory reprogramming of surrounding TAMs in the TME. To test this hypothesis, FACS-sorted Phago⁻ and Phago⁺ MDMs from Co.TD-Mo- or p21TD-Mo-infused, mCherry⁺ MOLT4 cell- engrafted NSG mice were seeded in the upper chambers of Transwell devices and cocultured with hCD14⁺hCD11b⁺hCD163⁺CFSE⁺ MDMs (TAMs) (from mCherry⁺ MOLT4 cell-engrafted NSG mice infused with CFSE⁺ human Mos) in the bottom chambers for 15 d; then, iNOS expression was analyzed (Supplementary Fig. 16). In accordance with our in vitro findings (Fig. 1p), TAMs were cocultured with Phago⁺ MDMs sorted from tumor-bearing mice injected with Co.TD-MDMs

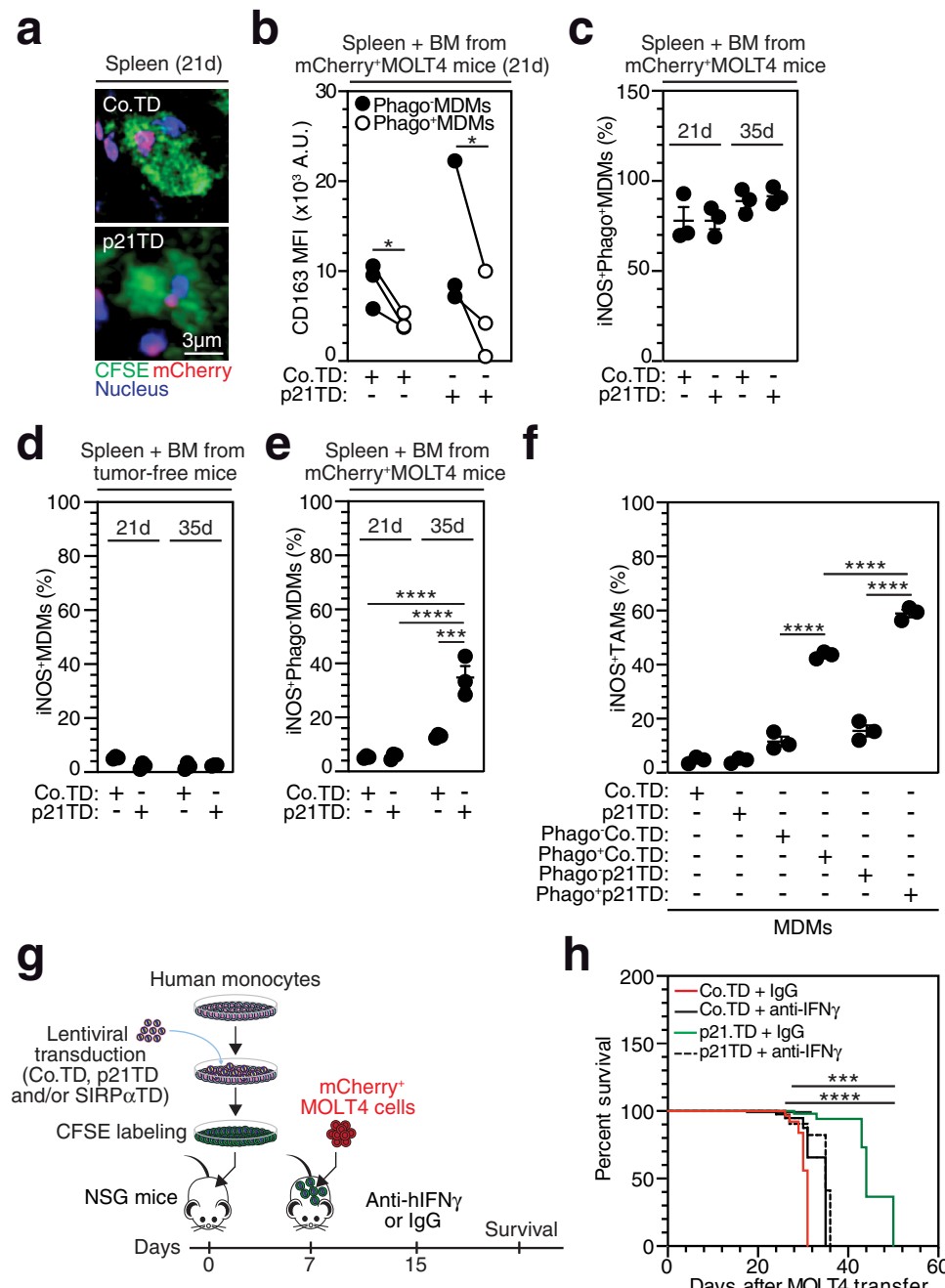

**Fig. 4 | Prophylactic adoptive transfer of p21TD-Mos triggers the proin-flammatory activation of TAMs and prolongs the survival of human T-ALL-engrafted mice in an IFNγ-dependent manner. a, b** Confocal micrographs of phagocytic (mCherry⁺CFSE⁺) macrophages in the spleen (**a**) and CD163 membrane expression identified by FACS on sorted single-positive CFSE⁺ (Phago⁻) MDMs or phagocytic mCherry⁺CFSE⁺ (Phago⁺) MDMs from the spleen and BM (*p = 0.033) (**b**) of mCherry⁺ MOLT4 cell-engrafted NSG mice that received Co.TD-Mos or p21TD-Mos, as shown in Fig. 3a, on d 21 after Mo transfer. **c–e** Percentages of iNOS-expressing (iNOS⁺) cells, determined by immunofluorescence staining, among Phago⁺ MDMs (**c**), MDMs (**d**) or Phago⁻ MDMs (****p = 4.6e-5, ****p = 5e-5, ***p = 0.0004) (**e**) sorted from tumor-free (**d**) or mCherry⁺ MOLT4 cell-engrafted (**c**, **e**) NSG mice that received Co.TD-Mos or p21TD-Mos; cells collected on d 21 and 35 after Mo transfer. **f** Percentages of iNOS⁺ cells among human TAMs (hCD14⁺hCD11b⁺hCD163⁺CFSE⁺), sorted from the BM and spleen of mCherry⁺ MOLT4 cell-engrafted NSG mice adoptively transferred with CFSE⁺ human Mo; cells collected on d 35 after

Mo transfer. These cells were cocultured, as shown in Supplementary Fig. 16, in the bottom chambers of Transwell devices with Phago⁻ MDMs or Phago⁺ MDMs sorted from the spleen and BM of mCherry⁺ MOLT4 cell-engrafted NSG mice adoptively transferred with Co.TD-Mos or p21TD-Mos (d 35 after Mo transfer) or with control MDMs differentiated in vitro from Co.TD-Mos or p21TD-Mos (****p = 1.4e-8, ****p = 4.2e-5, ****p = 3.8e-10). **g, h** Schematic representation (**g**) and survival Fig. 4h (****p = 1.2e-7, ***p = 0.0002) (**h**) of mCherry⁺ MOLT4 cell-engrafted NSG mice that received Co.TD-Mos or p21TD-Mos and were treated 15 d after Mo transfer with isotype control (IgG) or anti-IFNγ blocking antibodies. In (**a**), the data are representative of n = 5 mice/group. In **b**, the data are mouse matched from n = 3 mice/group. In **c–f**, the data are presented as the mean±SEM from n = 3 mice/group. In **h**, the survival data are from n = 7 mice/group. Exact p-values are indicated and determined with one-way ANOVA with Friedman's (**b**) and Sidak's (**f**) or two-way ANOVA with Tukey's (**e**) multiple comparison tests or determined with log-rank Mantel-Cox (**h**), test. Source data are provided as a Source Data file.

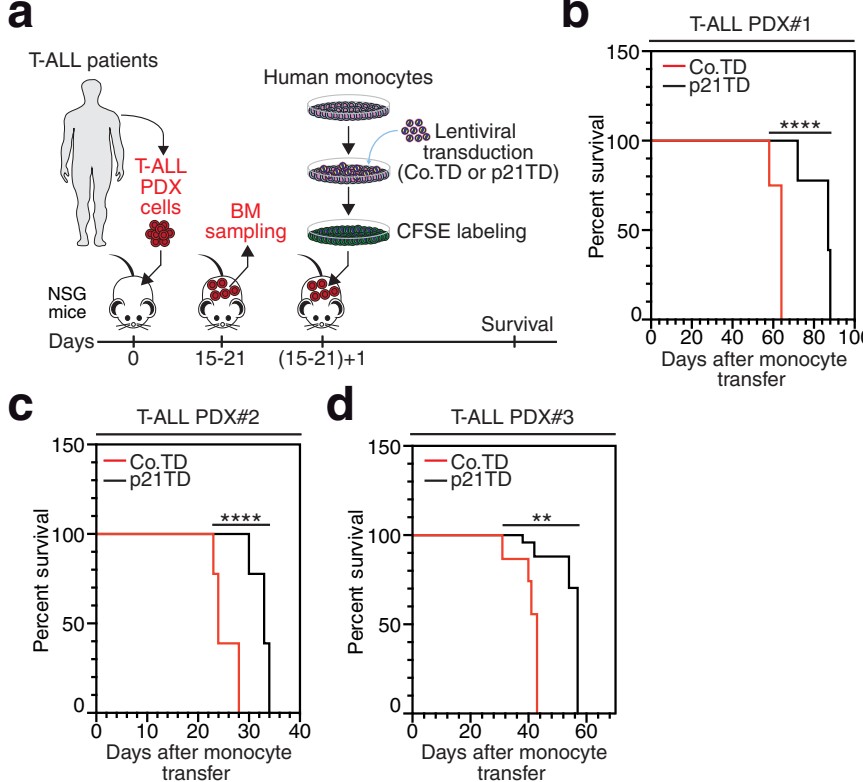

**Fig. 5 | Curative adoptive transfer of p21TD-Mos prolongs mouse survival in diagnosed or relapsed T-ALL-derived PDX models. a** Schematic representation showing curative adoptive transfer of Co.TD-Mos or p21TD-Mos into NSG mice engrafted with T-ALL PDXs. **b−d** Survival of mice engrafted with T-ALL PDX#1 (****$p$ = 1.1e-5) (**b**), PDX#2 (****$p$ = 1.9e-6) (**c**) or PDX#3 (**$p$ = 0.003) (**d**) and treated curatively, as shown in **a**, with Co.TD-Mos or p21TD-Mos. PDX#1 and PDX#2 were from the same patient but isolated at the diagnosis and relapse stages, respectively. In **b−d**, the survival data are from $n$ = 5 mice/group. Exact $p$-values are indicated and determined with the log-rank Mantel-Cox (**b−d**) test. Source data are provided as a Source Data file.

or p21TD-MDMs showed a significant increase in iNOS expression (Fig. 4f). We noticed that TAMs that were cocultured with Phago⁺ p21TD-MDMs exhibited significantly increased expression of iNOS compared to those that were cocultured with Phago⁺ Co.TD-MDMs. To exclude a potential role for mouse macrophages in the antitumor effect elicited by p21TD-Mo-based therapy, the expression of proinflammatory mouse macrophage markers (murine cell-surface MHCII (Supplementary Fig. 17a–d, i) and iNOS (Supplementary Fig. 17e–h, j)) was analyzed. No increased expression of these markers was detected in mCD45⁺mF4/80⁺ mouse macrophages sorted on d 21 or 35 after Mo transfer from the spleen or BM of mCherry⁺ MOLT4 cell-engrafted or tumor-free NSG mice adoptively infused with Co.TD-Mos or p21TD-Mos as compared to corresponding cells sorted from control mice (Supplementary Fig. 17), suggesting that the observed bystander proinflammatory activation of human TAMs (Fig. 4e, f) mainly depended on factors secreted by the p21TD-Phago⁺ MDMs. Hence, given the key role of IFNγ secretion by Phago⁺ MDMs during in vitro proinflammatory reprogramming of surrounding Phago⁻ MDMs (Fig. 1), the contribution of this proinflammatory cytokine to the antitumor effect elicited by the adoptive transfer of p21-engineered Mos was determined. Specifically, p21TD-Mo-transferred NSG mice engrafted with mCherry⁺ MOLT4 cells were treated with anti-human IFNγ (hIFNγ) blocking antibodies 15 d after Mo transfer and analyzed for survival (Fig. 4g). Treatment with the anti-hIFNγ blocking antibodies strongly reversed the increase in survival observed for mice that received p21TD-Mos (Fig. 4h), confirming that IFNγ secreted by phagocytic p21-engineered TAMs and/or in response to the proinflammatory reprogramming of surrounding TAMs is a central cytokine in the antitumor effect detected after Mo transfer. Altogether, these results indicate that p21TD-Mo-based cellular therapy drives the

engraftment of p21-transduced phagocytes, which, in addition to directing the elimination of leukemia cells, triggers the secretion of the proinflammatory cytokine IFNγ and supports the proinflammatory reprogramming of TAMs, which in turn participate in the regression of leukemia.

## p21TD-Mo-based cellular therapy prolongs mouse survival in diagnosed or relapsed T-ALL patient-derived xenograft (PDX) models

To further explore the therapeutic effect of p21TD-Mo-based cellular therapy, we treated NSG mice engrafted with patient-derived T-ALL cells (Supplementary Table 2), including two PDX models derived from the same patient at diagnosis (PDX#1) and at relapse (PDX#2) (Fig. 5a and Supplementary Fig. 18). Adoptive transfer of p21TD-Mos significantly increased the survival of T-ALL PDX-engrafted mice compared with adoptive transfer of Co.TD-Mos (Fig. 5b–d). Moreover, the p21TD-Mo-based cellular therapy remained effective in mice engrafted with T-ALL cells at relapse after one year of chemotherapy (Supplementary Table 2). Altogether, these results support the translational potential of the therapeutic induction of p21-mediated tumor cell phagocytosis for the treatment of T-ALL.

## Discussion

Taken together, our findings demonstrate that macrophage-expressed p21 is responsible for the proinflammatory activation of macrophages through the induction of tumor cell phagocytosis. Our findings provide direct genetic and functional evidence that macrophage-expressed p21 is a key regulator of leukemia cell phagocytosis, acting by repressing the transcription of the phagocytosis inhibitory receptor SIRPα. After the degradation of engulfed cells, phagocytic

macrophages undergo proinflammatory activation and, through the secretion of IFNγ, convert surrounding anti-inflammatory macrophages into proinflammatory cells. Supporting the translational potential of our findings, the adoptive transfer of p21TD-Mos into mouse models of human leukemia was shown to lead, after p21TD-Mo migration and differentiation into TAMs in the spleen and BM, to phagocytosis of leukemia cells and proinflammatory reprogramming of both the engineered TAMs and neighboring nonphagocytic TAMs. The p21-initiated, SIRPα-repressed, phagocytosis-guided, IFNγ-dependent proinflammatory macrophage activation of TAMs strongly reduced the leukemic burden and prolonged mouse survival, thus revealing a non-cell-autonomous mechanism to direct the killing of CD47-expressing leukemia cells and harness antitumor innate immunity in a way that could be targeted for cellular therapy against leukemia.

Our findings also identify SIRPα as a target of the transcriptional corepressor p21[17,33,34]. Considering that simply depleting macrophage SIRPα does not lead to the phagocytosis of CD47-expressing cancer cells[38–40], instead requiring adjuvant immune stimulation such as exposure of proinflammatory cytokines[38], TLR agonists[39] or radiation[40] to trigger tumor cell phagocytosis, future studies should identify the upstream and downstream effectors of p21 that enable p21 binding to the SIRPα promoter and contribute to the repression of SIRPα transcriptional activity and should elucidate the molecular mechanisms involved in the tumor cell-phagocytosing activity and proinflammatory activation of engineered phagocytic macrophages. Our findings also reveal that IFNγ plays a dominant role in the antitumor effect exerted by p21TD-Mo-derived macrophages. In addition to its direct effect on malignant cells (Supplementary Fig. 19 and refs. [41–44]), IFNγ released by phagocytic macrophages can induce the proinflammatory reprogramming of neighboring nonphagocytic anti-inflammatory macrophages, thus supporting the possibility that inducing the TME accumulation of p21TD-Mo-derived macrophages with enhanced abilities to engulf leukemia cells and trigger the proinflammatory reprogramming of TAMs in situ could be an attractive alternative approach to overcome tumor immune evasion strategies[12].

Our findings also support the hypothesis that in syngeneic immunocompetent murine models of leukemia, the effectiveness of p21TD-Mo-based cellular therapy would be increased, since phagocytic macrophages in the TME would recruit and activate host immune cells in addition to bystander macrophages to support antitumor T-cell priming and favor an adaptive antitumor immune response. The therapeutic feasibility and safety of autologous macrophage-based cellular therapy have been previously shown[16,45]. However, genetic engineering of primary myeloid cells with clinically approved vectors, such as a self-inactivating (SIN) human immunodeficiency virus 1 (HIV-1)-based lentivirus, has remained a major difficulty for a long time[16]. In this study, we circumvented the resistance of primary Mos and macrophages to lentiviral transduction by cotransducing a p21-expressing lentiviral vector with viral-like particles containing the Vpx protein, which was shown to degrade the SAMHD1 viral restriction factor without affecting p21 expression[17] and thus enabled us to evaluate the potential of p21TD-Mo-based cellular therapy. Adoptive transfer of p21TD-Mos would represent a broad-spectrum therapy for leukemia because it can enhance the macrophage capacity for phagocytosis without affecting antigen pressure selection, which could induce the emergence of clonal resistance[46]. Finally, considering the current difficulties associated with eradicating acute leukemia, adoptive transfer of p21TD-Mos should be considered a strategy that could complement commonly used chemotherapeutic approaches.

## Methods

All research conducted was compliant with all relevant ethical regulations. Human samples were obtained and experimented after approval of the ethical committees of Etablissement Français du Sang (EFS), Comité Consultatif sur le Traitement de l'Information en matière de Recherche dans le domaine de la Santé (CCTIRS), Commission Nationale de l'Informatique et des Libertés (CNIL) and the institutional review board (IRB00003888) of Institut National de la Santé et de la Recherche Médicale (INSERM). Animal studies were approved by the ethical committee (Comité d'Ethique en Expérimentation Animale N°026 (CEEA26)) of the Ministère de l'éducation nationale, de l'enseignement supérieur et de la recherche (France).

### Primary cells and cell lines

Mos, MDMs, and PBLs were obtained and differentiated as previously described[17,47] from peripheral blood mononuclear cells (PBMCs) in the buffy coats of healthy donors from the French blood bank, after approval of Etablissement Français du Sang (EFS) ethical committee (convention N°14EFS003) in accordance with French law and with the written informed consent of each donor. Mos were isolated from PBMCs by adherence to plastic in macrophage medium (MM) (RPMI medium supplemented with 200 mM L-glutamine, 100 U/ml penicillin, 100 µg/ml streptomycin, 10 mM HEPES, 10 mM sodium pyruvate, 50 µM β-mercaptoethanol, 1% minimum essential medium vitamins, and 1% nonessential amino acids; all from Gibco) containing 2% (vol/vol) heat inactivated (HI, 1 h at 56 °C) human AB serum (hABS) (Sigma, #H3667). After extensive washing with DPBS (Gibco, #14190-094) to eliminate nonadherent cells, Mos were incubated overnight in MM containing 10% HI hABS before in vitro differentiation or in MM containing 2% HI fetal bovine serum (FBS) before genetic engineering and/or adoptive transfer. Mo purity was analyzed by FACS, which revealed that 90 to 96% of cells expressed the markers hCD11b and hCD14 and did not express hCD56 (NK cells), hCD3 (T cells), and hCD20 (B cells). For differentiation into MDMs, Mos were cultured for 6 to 7 d in hydrophobic Teflon dishes (Lumox, #94.6077.305) in MM containing 20% HI hABS, yielding adherent nonproliferating cells, among which 90-96% expressed macrophage (hCD11b, hCD14, hCD71) and M2-like (hCD163 and hCD206) markers. MDMs were then cultured in MM containing 10% HI FBS at $1 \times 10^6$/ml for experiments. The isolation, PHA-P/IL2 activation, and culture of PBLs were performed as previously described[47]. Jurkat (Jurkat, CloneE6-1) (#TIB-152), MOLT4 (MOLT-4) (#CRL-1582), CEM (CCRF-CEM) (#CCL-119), HEL (HEL 92.1.7) (#TIB-180) and K562 (K-562#) (CCL-243) cells were all obtained from American Type Culture Collection (ATCC) and cultured in RPMI medium supplemented with 10% HI FBS, except for K562 cells, which were grown in IMDM (Gibco, #21980-032) supplemented with 10% HI FBS. No further authentication of the cell lines was performed. For phagocytosis assays, all target cells were confirmed to be negative for mycoplasma contamination and were cocultured with MDMs in MM supplemented with 10% HI FBS.

### Cell cycle distribution, cell proliferation and cell viability assays

The cell cycle distribution of MDMs was analyzed using propidium iodide (PI) (Sigma, #P4864) and FACS analysis. Briefly, MDMs ($1 \times 10^6$) were scraped at the indicated time points, fixed for 30 min with a cold 70% ethanol (400 µl) solution at −20 °C, washed extensively twice with DPBS and incubated protected from light for 30 min at 37 °C in 400 µl DPBS containing PI (20 µg/ml), DNase-free RNase A (Macherey-Nagel, #740505) (100 µg/ml) and EDTA (20 mM) before FACS analysis. The cell proliferation and viability of both MOLT4 cells and engineered MDMs were assessed with a Cell Proliferation Reagent WST-1 kit (Roche, #5015944001) and using trypan blue (Sigma, #T8154) according to the manufacturer's instructions. MOLT4 cells ($0.5 \times 10^6$/ml) given control treatment or treated with carrier-free recombinant hIFNγ (RD, #285-IF-100/CF) (20 IU/ml, 200 IU/ml or 1000 IU/ml) were analyzed with the WST-1 assay, and the viable cell number was counted on d 1, 3, and 6 after treatment.

## Patient samples

PB and BM samples from AML patients (Supplementary Table 1) were prospectively collected after obtaining informed consents from human participants to be included and for publishing their information in the MYELOMONO2 study of Groupe Francophone des Myélodysplasies conducted according to the Declaration of Helsinki and after approval by the relevant ethical committees (Comité Consultatif sur le Traitement de l'Information en matière de Recherche dans le domaine de la Santé (CCTIRS N°14-266) and Commission Nationale de l'Informatique et des Libertés (CNIL N°914283)). Mononucleated cells in the PB or BM were then enriched for CD34[+] cells with an AutoMacs system using a human CD34 MicroBead kit (Miltenyi Biotec, #130-046-702) according to the manufacturer's instructions. The CD34 cell purity was checked by FACS, and cells were cultured in IMDM containing 10% HI FBS supplemented with a human cytokine and growth factor cocktail from PeproTech (hTPO (#300-18) (10 ng/ml), hSCF (#300-07) (25 ng/ml), hIL3 (#200-03) (10 ng/ml), hIL6 (#200-06) (10 ng/ml), and hFlt-3 (#AF-300-19) (10 ng/ml)) for at least three days before phagocytosis assays. The study of T-ALL PDX samples (Supplementary Table 2) was approved by the institutional review board (IRB00003888) of Institut National de la Santé et de la Recherche Médicale (INSERM) (project N° 13-105-2) and informed consents were obtained in accordance with the Declaration of Helsinki as previously described[48]. The human participants or their legal representatives also provided informed consents for publishing their information. No compensation was given for patient samples used in this study.

## Plasmids, lentiviral vectors, and transduction

For genetic engineering of Mos, MDMs and MOLT4 cells, SIN lentiviral vectors were used. The p21 cDNA sequence used for p21TD-Mo-based cellular therapy was cloned into the pAIP lentiviral vector[49] under the control of the SFFVp promoter. P21 cDNA (used for assessing transduction efficiency by detecting GFP in live cells by fluorescence microscopy) and SIRPα cDNA sequences were cloned into the pRRL-EF1-PGK-GFP lentiviral vector under the control of the EF1 promoter. A mCherry cDNA sequence was cloned into the pRRL lentiviral vector under the control of the EF1 promoter, and this vector was used to establish the stable mCherry[+] MOLT4 cell line. For the establishment of the CD47-depleted MOLT4 cell line (CrCD47MOLT4), two different CRISPR CD47 gRNAs (gRNA1: ATCGAGCTAAAATATCGTGT; gRNA2: CTACTGAAGTATACGTAAAG), which were previously designed and validated[50], were cloned into the eSpCas9-LentiCRISPRv2 lentiviral vector expressing the *CAS9* gene (GenScript). The control MOLT4 cell line (CrCo.MOLT4) was obtained by transduction with the empty eSpCas9-LentiCRISPRv2 vector. The shRNA targeting the 3' untranslated region of p21 (sh3'UTRp21 CGCTCTACATCTTCTGCCTTA) cloned into the pLKO.1 lentiviral vector was predesigned and validated by Sigma MISSION (#TRCN0000287021). The SIRPA 2 promoter sequence (EP026655) identified by the Eukaryotic Promoter Database (EPD) to be expressed in human primary Mos and macrophages was fused to a *luciferase* (Luc) reporter gene and cloned into the pRRL-EF1-PGK-GFP lentivirus after the insertion of three stop codons at the 3' end of the E2F1 promoter. The initiator motif of the SIRPA 2 promoter is present at −530 bp before the transcription start site (TSS) of the *SIRPα* gene. Control (AIP, RRL-EF1-PGK-GFP, eSpCas9-LentiCRISPRv2 and LKO.1), gene-encoding (AIP-p21, RRL-EF1-p21-PGK-GFP, RRL-EF1-SIRPα-PGK-GFP, RRL-EF1-3xstop codons-SIRPA 2-Luc-PGK-GFP and RRL-mCherry), gRNAs-encoding (eSpCas9-LentiCRISPRv2-CD47gRNA1 and eSpCas9-LentiCRISPRv2-CD47gRNA2) and shRNA-encoding (LKO.1-sh3'UTRp21) lentiviral vectors were produced by separately transfecting $5 \times 10^6$ HEK293T (ATCC#CRL-3216) cells in a T150 plate in 10 ml Opti-MEM medium (Gibco, #31985-070) (containing 2% HI FBS, 100 U/ml penicillin, 100 μg/ml streptomycin) with 4 μg pDM2-VSV-G, 10 μg pΔ8.91 packaging and 8 μg lentiviral vector plasmid using Fugene (Promega, #E2312) according manufacturer's instructions. The

cell supernatants were removed after 24 h of transfection, and the cells were refed with by 10 ml fresh Opti-MEM medium (containing 2% HI FBS, 100 U/ml penicillin, and 100 μg/ml streptomycin) for an additional 48 h. Then, the cell supernatants were harvested and filtered through a 0.45-μm-pore size filter. Viral stocks were quantified for viral CAp24 content by enzyme-linked immunosorbent assay (ELISA) (Perkin Elmer, # NEK050A) and by qPCR quantification of integrated lentiviral copies in the human genome of MDMs as previously described[47,51]. Viral-like particles containing the Vpx protein (VLPs-Vpx[+]) were produced, as described above for lentiviral vectors, by cotransfecting $5 \times 10^6$ HEK293T cells with 4 μg pSIV3[+](Vpx[+]) (kindly provided by A. Cimarelli, ENS Lyon) and 4 μg pDM2-VSV-G. Mos ($10^7$) or MDMs ($10^6$) were treated for 1 h at 37 °C with 1 ml VLPs-Vpx[+] containing 5 μg/ml polybrene (Sigma, #H9268) for SAMHD1 degradation[17,51] before the addition of 20 μg CAp24 of each control or/ and encoding-gene lentiviral vectors with 5 μg/ml polybrene for 2 h of transduction. After extensive washing with medium, the Mos were differentiated for 7 d in MM containing 20% HI hABS before western blot (WB) analysis and phagocytosis assays. The MDMs were cultured for 72 h before qPCR analysis, luciferase reporter assays with a luciferase assay system (Promega, #E1500), ChIP-qPCR assays, WB analysis or phagocytosis assays. Genetically engineered Mos (Co.TD-Mos or p21TD-Mos) used for adoptive transfer into NSG mice were further incubated for 30 min with a 1 μM CellTracer CFSE cell proliferation kit (Invitrogen, #C34554) before extensive washing with DPBS and intravenous (IV) injection into mice. MOLT4 cells ($5 \times 10^6$) were transduced with 100 μg CAp24 RRL-mCherry by 1 h of spinoculation at 1200 g at 25 °C and 1 h of incubation at 37 °C. After 72 h, the MOLT4 cells expressing the *mCherry* gene were sorted by FACS and cultured in RPMI medium supplemented with 10% HI FBS. CrCD47MOLT4 or CrCo.MOLT4 cells were obtained by separately transducing mCherry[+] MOLT4 cells ($5 \times 10^6$) with 50 μg CAp24 of each eSpCas9-LentiCRISPRv2-CD47gRNA1 and eSpCas9-LentiCRISPRv2-CD47gRNA2 or with 100 μg CAp24 SpCas9-LentiCRISPRv2 by 1 h of spinoculation at 1200 g at 25 °C and 1 h of incubation at 37 °C. After 72 h, CrCD47MOLT4 or CrCo.MOLT4 cells were selected with puromycin (InvivoGen, #ant-pr) (500 ng/ml) for 15 d before the assessment of cell-surface CD47 depletion by FACS. For the in vitro and in vivo experiments performed with the CrCD47MOLT4 or CrCo.MOLT4 cell lines, puromycin was removed from the culture medium at least 72 h or 7 d earlier, respectively.

## Macrophage phagocytosis assays

MDMs ($0.125 \times 10^6$ cells) were allowed to adhere for 2 h in 125 μl MM supplemented with 10% HI FBS in one well of an 8-chamber tissue culture treated glass slide (Falcon, #354118), labeled with 20 μM Cell-Tracker Green CMFDA (Invitrogen, #C2925) for 1 h at 37 °C and washed three times with 1 ml medium before coculture with target cells. Leukemia cells, PBLs or CD34[+] AML cells were incubated at $1 \times 10^6$/ml in RPMI medium supplemented with 10% HI FBS containing 100 μM ZVAD (Bachem, #4027403.0005) and 10 μM CellTracker Orange CMTMR (Invitrogen, #C2927) for 1 h at 37 °C. After extensive washing, CMTMR[+] target cells ($0.125 \times 10^6$ cells) were added to adherent CMFDA[+] MDMs (1:1 ratio) at $1 \times 10^6$ cells/ml in MM supplemented with 10% HI FBS for 8 h of coculture in the presence of ZVAD (100 μM), Y27632 (30 μM) (Tocris, #1254), human recombinant Annexin V (5 μg/ml) (Invitrogen, #BMS306) or the same amounts of the respective solvents (DMSO or H2O). MDMs were pretreated with Y27632 (30 μM, 24 h), PMA (30 ng/ml, 32 h) (Sigma, #P1585), MS275 (1 μM, 32 h) (Enzo, #ALX-270-378-M001) or the same volume amount of DMSO before coculturing with target cells. MDMs were also allowed to adhere to wells precoated with 100 mg human IVIg (CSL Behring) for 24 h before extensive washing of the MDMs and cocultures. MDMs for p21 (sip21) or p53 (sip53) silencing and their corresponding controls (siCo.) were cocultured with target cells or with pHrodo Green

bacterial *E. coli* bioparticles (according to the manufacturer's instructions) (Life Technologies, #P35366) at 24 h after siRNA transfection. Jurkat or CD34$^+$ AML cells were pretreated with 50 µM cis-platinum (CDDP) (Mylan) for 24 h to induce apoptosis before they were cocultured with MDMs. After 8 h of coculture, the cell supernatant (containing any target cells not internalized by the MDMs) was removed, and the MDMs were washed extensively and fixed for 5 min with 2% paraformaldehyde (PFA) before confocal microscopy analysis. Phagocytosis assays performed with anti-CD47 blocking antibodies were performed as previously described[37]; specifically, MDMs were cultured in serum-free IMDM (Gibco, #12440-53) for 2 h before being cocultured with MOLT4 cells in the same medium with the InVivoMAb anti-human CD47 antibody clone B6.H12 (7 µg/ml) (Bio Cell, #BE0019-1) or an IgG1 isotype control clone MOPC-21 (Bio Cell, #BE0083) for an additional two hours.

### In vitro functional assays performed with Phago$^-$ MDMs and Phago$^+$ MDMs

MDMs ($10 \times 10^6$) resuspended at $1 \times 10^6$/ml in MM supplemented with 10% HI FBS were labeled with 2.5 µM CMFDA for 1 h at 37 °C in a 50 ml Falcon tube. Target cells (MOLT4 or CD34$^+$ AML cells) ($10 \times 10^6$) were resuspended at $1 \times 10^6$/ml in RPMI medium supplemented with 10% HI FBS containing 100 µM ZVAD for staining with 1.25 µM CMTMR for 1 h at 37 °C in a 50 ml Falcon tube. After extensive washing with medium, MDMs and target cells (1:1 ratio) were cocultured in 50-ml Falcon tubes at $1 \times 10^6$/ml in MM supplemented with 10% HI FBS for 2 h at 37 °C. Then, Phago$^+$ MDMs (CMTMR$^+$CMFDA$^+$) MDMs and Phago$^-$ MDMs (CMFDA$^+$) were sorted by FACS. The purity of the sorted populations was checked with FACS. Phago$^+$ MDMs and Phago$^-$ MDMs (at least $0.01 \times 10^6$ each) were then allowed to adhere to the wells of 8-chamber tissue culture-treated glass slides for 2 h and 96 h, respectively, before 2% PFA fixation and confocal microscopy analysis of target cell internalization and degradation by MDMs. For gene expression microarray analysis, CD163 membrane expression analysis by FACS, and cell supernatant analysis with a proteome profiler human cytokine panel A array (according to the manufacturer's instructions) (RD, #ARY005), Phago$^+$ MDMs and Phago$^-$ MDMs (at least $0.1 \times 10^6$ each) were cultured in MM supplemented with 10% HI FBS for 96 h. The mean pixel densities of analyzed cytokines were determined with Gene tools from Syngene software version 4.02.03 (Syngene). For quantification of human IFNγ secretion into the cell supernatants of Phago$^+$ MDMs and Phago$^-$ MDMs by Quantikine ELISA (RD, #DIF50C) (according to the manufacturer's instructions), the degradation of internalized cells was assessed at 96 h and after extensive washing, fresh medium was added to the MDMs for up to 7 d for quantification. For WB analysis, Phago$^+$ MDMs and Phago$^-$ MDMs were cultured for 7 d. For the study of bystander activation of macrophages, 2 h after FACS sorting, Phago$^-$ MDMs ($0.1 \times 10^6$ in 1 ml of MM supplemented with 10% HI FBS) were allowed to adhere to the bottom chambers of 24-well plates; Phago$^-$ MDMs or Phago$^+$ MDMs ($0.1 \times 10^6$ in 200 µl) adhered to 0.4 µm-pore membranes of Transwell cell culture inserts (Falcon, #353095) were used in 15-d cocultures performed in the presence of 1 µg/ml anti-hIFNγ antibody clone 25718 (RD, #MAB285) or mouse IgG2A isotype control (RD, #MAB003). The Phago$^-$ MDMs in the bottom chambers were then analyzed by WB.

### Mouse models of human T-ALL

Mouse studies were performed in accordance with protocols approved by the French Ethical Committee (Comité d'Ethique en Expérimentation Animale N°026 (CEEA26)) of the Ministère de l'éducation nationale, de l'enseignement supérieur et de la recherche (Project N°2012-022) and following recommendations for proper use and care during animal experimentation. Mouse experiments were performed using 6- to 8-week-old female (Fig. 3a–n, Fig. 4a–h, Fig. 5a–d, Supplementary Fig. 9a–i, Supplementary Fig. 11a–g,

Supplementary Fig. 12a, b, Supplementary Fig. 13a, b, Supplementary Fig. 14a–c, Supplementary Fig. 16, Supplementary Fig. 17a–h and Supplementary Fig. 18a–f) and male (Supplementary Fig. 15, Supplementary Fig. 19c) NOD.Cg-*Prkdc*$^{scid}$*Il2rg*$^{tm1Wjl}$/SzJ (NSG) immunodeficient mice purchased from Charles River Laboratories; the mice were maintained in specific pathogen-free (SPF) grade room (at 20–22 °C ambient temperature, humidity (45–60%) and 12 h (7:00 a.m.–7:00 p.m.) light/dark cycle) and randomized to homogenous mouse body weight groups (20 g to 23 g) before experiments. IV injections of 200 µl cell suspension in DPBS/mouse were performed via the retro-orbital sinus under isoflurane gas anesthesia. FACS-sorted mCherry$^+$ MOLT4 cells ($10^6$/mouse), obtained by RRL-mCherry lentiviral vector transduction as described above, were intravenously injected into NSG mice and then after 30 d, mCherry$^+$ leukemia cells that engrafted in the BM were sorted by FACS and cultured in RPMI medium supplemented with 10% HI FBS for in vitro macrophage-mediated phagocytosis assays and mouse injections. To characterize leukemia progression, mCherry$^+$ MOLT4 cells ($10^6$/mouse) were intravenously injected into NSG mice 7 d after total body irradiation (TBI) with 1 Gy. The percentage of mCD45$^-$hCD45$^+$mCherry$^+$ cells detected in PB was determined every week. After 35 d, the mice were sacrificed, and the presence of leukemia cells in the BM and spleen was analyzed by FACS. Similarly, T-ALL PDX cells ($10^5$/mouse for PDX#1 and PDX#2 and $2 \times 10^5$/mouse for PDX#3) were intravenously injected into NSG mice. PDX cell engraftment were assessed on 21 d (for PDX#1 and PDX#2) and 15 d (for PDX#3) by femoral BM sampling as previously described[48] and FACS detection of hCD45$^+$hCD7$^+$ PDX cells. The ethical endpoints for mice sacrifice are the loss of 20% of body weight and/or clinical signs of illness (general prostration and weakness).

### Adoptive Mo-based cellular therapy

Human Mos ($5 \times 10^6$/mouse) were labeled with 1 µM CFSE (Invitrogen, #C34554) for 30 min at 37 °C, extensively washed and intravenously injected into NSG mice 24 h after 1 Gy total TBI with an X-RAD-320 irradiator (Precision X Ray). Seven days after Mo transfer, the NSG mice were sacrificed, and CFSE$^+$ cells were sorted from BM and spleen cells and analyzed for human anti-inflammatory macrophage markers (hCD11b, hCD14, and hCD163) to assess the in vivo differentiation of human Mos into macrophages. Then, the engraftment and persistence of genetically engineered Mos in mice were analyzed. CFSE-labeled Co.TD-Mos or p21TD-Mos ($5 \times 10^6$/mouse) were intravenously injected into NSG mice given 1 Gy TBI, and the mice were sacrificed after 21 d, 35 d, or 72 d. The presence of CFSE$^+$ Mo-derived macrophages in PB, BM, spleen, and liver at 21 d and 35 d were evaluated by FACS and/or confocal microscopy and/or fluorescence microscopy. After 72 d, the engrafted Co.TD-MDMs or p21TD-MDMs were analyzed for the expression of human leukocyte and macrophage (hCD45, hCD11b, hCD71, and hCD14) markers, sorted and analyzed by qPCR for the quantification of the integrated lentiviral copy number in the genomic DNA as we previously described[47]. The integrated copy number was normalized to the MDM cell number determined by qPCR quantification of the endogenous reference gene albumin[47] from a standard curve generated with serial dilutions of MDMs. The integrated copy numbers detected from Co.TD-MDMs and p21TD-MDMs differentiated for 7 d in vitro were compared with those of Co.TD-Mo- and p21TD-Mo-derived macrophages 72 d after Mo transfer. For the prophylactic adoptive transfer of engineered Mos, NSG mice treated with 1 Gy TBI were intravenously injected with CFSE-labeled Co.TD-Mos or p21TD-Mos ($5 \times 10^6$/mouse) and then intravenously injected with mCherry$^+$ MOLT4, CrCo.MOLT4 or CrCD47MOLT4 cells ($10^6$/mouse) 7 d later. The mice were then analyzed for overall survival or sacrificed on 21 d to determine the leukemic burden (in the PB, BM, spleen, and liver by FACS and confocal microscopy). For curative adoptive transfer of engineered Mos, NSG mice were randomized 1 d after BM sampling showing 40–70% PDX cell engraftment, intravenously injected with

Co.TD-Mos or p21TD-Mos ($5 \times 10^6$/mouse), and monitored for overall survival.

### In vivo and ex vivo assays performed with Phago⁻ MDMs and Phago⁺ MDMs

In vivo proinflammatory activation of engineered macrophages was also determined by sorting Phago⁺ (mCherry⁺CFSE⁺ cells) and Phago⁻ (CFSE⁺) Co.TD- or p21TD-derived MDMs from BM and spleen cells and analyzing hCD163 membrane expression by FACS on 21 d and iNOS expression by immunofluorescence staining on 21 d and 35 d after Mo transfer. Briefly, Phago⁺ MDMs ($5 \times 10^2$) and Phago⁻ MDMs ($5 \times 10^3$) sorted from the spleen and BM by FACS were plated in one well of an 8-chamber tissue culture-treated glass slide (Falcon, #354118) or on one 10-mm-diameter microscope cover slip (Knittel Glass, #100045) in a 48-well plate, respectively, allowed to adhere for 24 h in RPMI medium containing 2% HI FBS and then fixed with 2% PFA for 5 min before iNOS staining. Additionally, at 21 d and 35 d, murine NSG macrophages were sorted from the spleen and BM by FACS based on the expression of the murine leukocyte marker CD45 (mCD45) and a macrophage marker (F4/80) and immunophenotyped by FACS for the murine proinflammatory marker MHCII (mMHCII) or adhered ($2.5 \times 10^4$) on a 10 mm-diameter microscope cover slip (Knittel Glass, #100045) in a 48-well plate for 24 h in RMPI medium containing 2% HI FBS and then fixed with 2% PFA for 5 min before iNOS staining for immunofluorescence analysis. mMHCII and iNOS staining of NSG macrophages was performed simultaneously with staining of wild-type proinflammatory activated murine Raw264.7 macrophages treated for 48 h with recombinant murine IFNγ (mIFNγ) (PeproTech, #315-05) (1 μg/ml). To determine the bystander proinflammatory activation induced by Phago⁺ MDMs, as illustrated in Supplementary Fig. 16, CFSE⁺ human TAMs (hTAMs), which were derived from adoptively transferred CFSE⁺ human Mos into mCherry⁺ MOLT4 cell-engrafted NSG mice, were sorted by FACS at 35 d after Mo transfer and immunophenotyped by FACS to assess the expression (98%) of hCD14, hCD11b, and hCD163. CFSE⁺ hTAMs ($5 \times 10^3$ in 500 μl of RPMI medium supplemented with 2% HI FBS) were then adhered to a 13-mm-diameter microscope cover slip (Knittel Glass, #100048) in bottom chamber of a 24-well plate containing Phago⁻ MDMs or Phago⁺ MDMs ($5 \times 10^2$ in 200 μl) adhered to 0.4-μm-pore membranes of Transwell cell culture inserts (Falcon, #353095) for 15 d of coculture. Phago⁻ MDMs or Phago⁺ MDMs in the upper chambers, as illustrated in Supplementary Fig. 16, were sorted by FACS from BM and spleen of mCherry⁺ MOLT4 cell-engrafted NSG mice adoptively transferred with Co.TD-Mos or p21TD-Mos collected on d 35 after Mo transfer. Control TAMs were cocultured with Co.TD-MDMs or p21TD-MDMs differentiated in vitro from Co.TD-Mos or p21TD-Mos, respectively. hTAMs on cover slips in the bottom chambers were then analyzed for iNOS expression by immunofluorescence staining, as previously described[52].

### In vivo functional studies

To determine the impacts of Co.TD-Mo- or p21TD-Mo-derived macrophages and their related hIFNγ secretion on the overall survival of mice, treated NSG mice were also intravenously injected with 200 μl/mouse clodronate- or control PBS-containing liposomes (Liposoma, #CP-005-005) or intraperitoneally injected with 100 μg/mouse IgG isotype control (RD, #MAB003) or anti-hIFNγ antibody (clone 25718; RD, #MAB285) 21 d or 15 d after Mo transfer, respectively. The clodronate-mediated depletion of CFSE-labeled Co.TD-Mo- or p21TD-Mo-derived macrophages was assessed by FACS 24 h after treatment. The role of IFNγ in leukemia progression was also evaluated by intraperitoneal injection of 10 μg/mouse carrier-free recombinant human hIFNγ (RD, #285-IF-100/CF) into mCherry⁺ MOLT4 cell-engrafted NSG mice on d 7 after MOLT4 cell injection. To study the effects of p21-mediated SIRPα repression on the overall survival of engrafted mice, prophylactic Mo adoptive transfer ($5 \times 10^6$/mouse) was performed with

genetically engineered Co.TD-Mos, p21TD-Mos, SIRPαTD-Mos or p21+SIRPαTD-Mos, which were transduced as described above with equal amounts of the AIP + RRL-PGK-GFP, AIP-p21+RRL-PGK-GFP, AIP + RRL-SIRPα-PGK-GFP or AIP-p21+RRL-SIRPα-PGK-GFP lentiviral vectors, respectively. To evaluate the effect of SIRPα-CD47 MIC blockade mediated by p21, the same mouse groups were treated 15 d after Mo transfer (7 d after MOLT4 transfers) with 100 μg/mouse InVivoMAb anti-human CD47 antibody clone B6.H12 (Bio Cell, #BE0019-1) or IgG1 isotype control clone MOPC-21 (Bio Cell, #BE0083) as previously described[37]; the antibodies were administered daily for 14 d by intraperitoneal injections.

### Gene knockdown in human macrophages

Gene knockdown in MDMs was performed as we previously described[17,47] using 50 nM or 25 nM (for p21 or p53 knockdown, respectively) of nontargeting pool control siRNAs (UGGUUUACAU GUCGA CUAA, UGGUUUACAUGUUGUGUGA, UGGUUUACAUGUUUU CUGA, and UGGUUU ACAUGUUUUCCUA), a p21 smart pool selected siRNA (AGACCAGCAUGACAGAUUU) or p53 smart pool siRNAs (GAAAUUUGCGUGUGGAGUA, GUGCAGCUGUGGGUU GAUU, GCAGU CAGAUCCUAGCGUC, and GGAGAAUAUUUCACCCUUC); all siRNAs were purchased from Dharmacon. MDMs were treated with p21- or p53-specific siRNA for 24 h before WB analysis or coculture with target cells.

### Gene expression

Determination of *p21* and *SIRPα* mRNA levels by RT-qPCR was performed as we previously described[17,53] using predesigned TaqMan gene expression probes for SIRPα (Hs00757426 S1), p21 (Hs00355782 m1) and GAPDH (Hs02758991 g1). The RT-qPCR data were analyzed with qPCR CFX Maestro Version 4.0.2325.0418. (Biorad). Gene expression analysis of Phago⁻ MDMs and Phago⁺ MDMs was performed with an Agilent® SurePrint G3 Human GE 8x60K Microarray (Agilent Technologies, AMADID 28004) using Microarray images Extraction software version (10.7.3.1) (Agilent Technologies) and Microarray raw intensities limma package for R v2.13.0 software (Agilent Technologies), with two technical replicates for each sample. Data were generated with each analyzed donor by determining the mean of two technical replicates of the Log2-fold change (FC) in the gene intensity ratio of Phago⁺ MDMs/Phago⁻ MDMs to produce a comprehensive analysis of modulated genes known for their ability to regulate anti-inflammatory or proinflammatory macrophage activation (Supplementary Data 1).

### ChIP-qPCR assays

ChIP assays were performed with MDMs using a ChIP assay kit (Millipore, #17−295) according to the manufacturer's instructions. Briefly, cross-linked cell chromatin was sheared by sonication for 10 min at 40 W (duty factor: 20%, peak incident power: 200, cycles per burst: 200) with a Covaris S220 (Woodingdean, UK). The chromatin fragments were then immunoprecipitated with 20 μg anti-p21 Waf1/Cip1 (12D1) antibodies (Cell Signaling Technology, #2947) or with equal amounts of a rabbit IgG isotype control. The DNA bound to the chromatin immunoprecipitates was eluted and analyzed by qPCR using Power SYBR Green PCR Master Mix (Applied Biosystems #4367659) for the detection of the SIRPα promoter sequence (SIRPA 2) with specific primers (SIRPA 2 forward (5′CCACCGAGACACCTG GCCAG3′ and SIRPA 2 reverse (5′AAGTGAACGCAGGGGGAAGG3′). The specificity of promoter sequence detection was confirmed with negative qPCR primer controls that targeted an unrelated promoter sequence at −2000 bp upstream of the 5′ end of the SIRPA 2 promoter (Untarget-SIRPA 2 forward (5′CCGTGGGTCTCAATGGCTTC3′ and Untarget-SIRPA 2 reverse (5′GGGGGATTAGGAAACTGGAG3′). The DNA amount was normalized by quantifying *albumin* gene copies with qPCR using a human genomic DNA (20 ng/μl) standard (Roche, #11691112001) as we previously described[17].

## Western blot analysis

WBs were performed as we previously described[17,53] using anti-p21 Waf1/Cip1 (12D1) (#2947, diluted 1:1000), anti-Myosin Light Chain 2 (MLC2) (#3672, diluted 1:500), anti-Phospho (Ser19) Myosin light chain 2 (MLC2S19*) (#3671, diluted 1:500), anti-αTubulin (#3873, diluted 1:1000) and anti-α/βTubulin (#2148, diluted 1:1000) antibodies from Cell Signaling Technology, anti-IL8 (#ab106350, diluted 1:500), anti-IRF5 (#ab21689, diluted 1:1000), anti-IL1β (#ab2105, diluted 1:1000) and anti-iNOS (#ab3523, diluted 1:500) antibodies from Abcam; an anti-IL6 antibody (RD, #AB-206-NA, diluted 1:500); an anti-SIRPα antibody (Invitrogen, #PA1-30537, diluted 1:2000); an anti-p53 (DO-1) antibody (Santa Cruz, #sc-126, diluted 1:1000) and an anti-GAPDH antibody (EMD Millipore, #MAB374, diluted 1:1000). Blot scan images were analyzed using GeneSys software version v1.3.9.0 (Genesys). The uncropped and unprocessed scans of all the blots are provided in the Source Data file.

## Flow cytometry

Cell samples were analyzed by FACS using a BD LSR2 or BD LSRFortessa (BD Biosciences) and sorted using a BD FACSAria III (BD Biosciences). Cell staining was performed in DPBS containing 2% (for cultured cells) or 5% (for mouse cells) HI FBS at 4 °C for 2 h with a 1:100 antibody dilution. For blood samples, red blood cells were lysed with ACK lysis buffer before antibody incubations and/or FACS analysis for mCherry+ or CFSE+ cells. Mouse cell samples were filtered through 100-μm-pore membrane filters before FACS staining and analyses. The antibodies used were the PE anti-hCD14 clone 61D3 (eBioscience, #12-0149-42, diluted 1:100); APC/Cy7 anti-hCD11b clone ICRF44 (#301342, diluted 1:100), PE anti-hCD34 clone 561 (#343606, diluted 1:100) and PE/Dazzle 594 anti-hCD7 clone CD7-6B7 (#343120, diluted 1:100) from BioLegend; and Alexa Fluor 647 anti-hCD163 clone GHI/61 (#562669, diluted 1:100), PE anti-hCD71 (#555537, diluted 1:100), FITC anti-hCD206 (#551135, diluted 1:100), PE Cy7 anti-hCD56 clone B159 (#560916, diluted 1:100), FITC anti-hCD3 (#555339, diluted 1:100), APC anti-hCD20 (#559776, diluted 1:100), PE anti-hCD47 (#558046, diluted 1:100), anti-hCD45 (#555485, diluted 1:50), and from BD Pharmingen. V450 anti-mCD45 clone 30-F11 (#560501, diluted 1:200) and BUV395 anti-hCD45 clone HI30 (#563792, diluted 1:200) are from BD Horizon. The FITC anti-hCD47 clone B6H12 (#11-0479-42, diluted 1:100), APC anti-hSIRPα clone 15-414 (#17-1729-42, diluted 1:100), APC anti-mF4/80 clone BM8 (#17-4801-82, diluted 1:100), Alexa Fluor 700 anti-mMHCII (I-A/I-E) clone M5/114.15.2 (#56-5321-82, diluted 1:100), and PE-Cyanine7 anti-hCD71 clone OKT9 (#25-0719-42, diluted 1:100) were purchased from Invitrogen. The membrane expression levels of SIRPα, CD163 and CD47 were determined from the mean fluorescence intensities (MFIs). FACS data were analyzed with FACSDiva version 8 (BD).

## Immunofluorescence staining and confocal microscopy

After fixation, cocultures containing single-positive CMFDA+ MDMs and phagocytic CMTMR+CMFDA+ MDMs were permeabilized with 0.1% Triton in DPBS for 5 min at room temperature, extensively washed, incubated with DPBS containing 10% HI FBS and 1:1000 Hoechst 33342 (Invitrogen, #H3570) for nuclear staining and mounted onto microscope cover slips with Fluoromount G (Southern Biotech, #0100-01). Immunofluorescence staining of fixed cells with an anti-LAMP2 (H4B4) antibody (Santa Cruz, #sc-18822, diluted 1:100) or anti-iNOS antibody (Abcam, #ab3523, diluted 1:500) and of formalin-fixed, paraffin-embedded sections (NSG mouse spleen, 4 μm/section) with anti-human CD45 clone JE03-05 (Invitrogen, #MA5-37809, diluted 1:50) and Alexa Fluor 647 anti-human CD68 clone Y1/82 A (BD Pharmingen, #562111, diluted 1:500) antibodies and acquisition by confocal microscopy (SP8, Leica) analysis of fixed stained cells and tissue sections were performed as we previously described[47]. The percentage of phagocytosis was determined by dividing the number of CMTMR +CMFDA+

MDMs by the total number of MDMs quantified in at least 5 fields containing at least 100 MDMs for each sample. The percentages of iNOS+ cells and live GFP+ MDMs were determined with DMI8 fluorescence microscopy (Leica) by dividing the number of cells expressing iNOS or GFP by the total cell number determined by detecting stained nuclei or cells using phase-contrast microscopy, respectively, to assess at least 5 fields of the cover slip or well containing at least 100 cells. The image analysis was performed using LASX software (Leica). For determination of cell surfaces and volumes, target cells were labeled with CMTMR at 10 μM and Hoechst 33342 at 1:1000 for 30 min at 37 °C and live imaged in ibidi 8-well chamber slides (ibiTreat, #80826) by confocal microscopy (SP8, Leica) with z-stack (0.22 μm step) acquisitions covering from the top to the bottom of the cells. After 3D construction, cell volumes and surfaces were measured with Volocity software version 6.2 (Quorum Technologies). For the detection of CFSE+ and mCherry+ cells engrafted in mouse tissues, fresh intact pieces of spleen, liver or longitudinally cut femur BM were incubated in 500 μl HI FBS containing 0.3% Triton and 1:250 Hoechst 33342 for 12 h at 4 °C. The medium was then replaced with cold HI FBS containing 1:250 Hoechst 33342, and the samples were stored on ice until confocal microscopy acquisition of each organ tissue in ibidi-8 well chamber slides. Mouse tissues (at least 2 mm²/organ) were imaged by confocal microscopy (SP8, Leica) using a 40x oil objective and hybrid detectors (pinhole airy: 0.6; pixel size: 284 nm, surface of each image field surface is 290.62 × 290.62 μm²) at an optimal optical sectioning (OOS) of 1 μm. Organ cell layers then underwent 3D construction with Imaris 5.7 software (Bitplane AG), and CFSE+ and mCherry+ cells were quantified and normalized to 10 mm² of tissue surface.

## Statistics

Statistical analysis was performed with GraphPad Prism 8.0 (GraphPad). Statistical tests, adjustments for multiple comparisons, exact calculated $p$ values, and n sizes are indicated for each figures in the corresponding figure legend. For all figures, statistical significances is indicated as *$p < 0.05$, **$p < 0.01$, ***$p < 0.001$, and ****$p < 0.0001$.

## Reporting summary

Further information on research design is available in the Nature Portfolio Reporting Summary linked to this article.

## Data availability

Source data are provided with this paper. The microarray gene expression data of the modulated genes in phagocytic macrophages with respect to nonphagocytic macrophages are provided in Fig. 1j, Supplementary Data 1 and Source Data file. The microarray gene expression large dataset is available in ArrayExpress repository under the access code E-MTAB-12280. The SIRPA 2 promoter sequence (EP026655) is available in the Eukaryotic Promoter Database (EPD) (https://epd.epfl.ch//index.php). The remaining data generated in this study are provided in the Article, Supplementary Information or Source Data file. Source data are provided with this paper.

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

## Acknowledgements

We gratefully acknowledge D. M. Ojcius for helpful scientific discussions, NH Theraguix for supporting our projects against cancer, and M. Morabito and L. Bencheikh for their technical support. This work benefited from the facilities and expertise of the Imaging and Cytometry Platform (S. Salome-Desnoulez and T. Manoliu), Genomic and Bioinformatic Platforms (N. Droin and G. Meurice) and UMS 3655 CNRS / US 23 INSERM (Gustave Roussy Cancer Campus Villejuif, France). This work was supported by funds from the Agence Nationale de la Recherche (ANR-10-IBHU-0001, ANR-10-LABX33, ANR-11-IDEX-003-01 and ANR Flash COVID-19 "MacCOV"), the Fondation ARC pour la recherche sur le cancer (www.fondation-arc.org), the Fondation de France, the Fondation Gustave Roussy, the Institut National du Cancer (INCa 9414 and INCa 16087), the SIRIC Stratified Oncology Cell DNA Repair and Tumor Immune Elimination (SOCRATE), the Care network (directed by X. Mariette, Kremlin Bicêtre AP-HP) and Université Paris-Saclay (to J.-L. Perfettini) and The SIRIC Stratified Oncology Cell DNA Repair and Tumor Immune Elimination 2.0 (SOCRATE 2.0, INCA-DGOS-INSERM 12551) (to A. Allouch).

## Author contributions

J.-L.P. conducted the study. A.A. and J.-L.P. designed the study. A.A., L.V., Y.Z., S.Q.R., and Y.L. performed the experiments. J.C., D.S.-B., S.dB., E.S., F.L. and F.P. provided PDX and AML samples. A.A., Y.L., F.P., and J.-L.P. analyzed the results. A.A. assembled the figures. A.A. and J.-L.P. wrote the manuscript. A.A., L.V., Y.Z., S.Q.R., Y.L., J.C., D.S.-B., S.dB., E.S., F.L., F.P., and J.-L.P. provided advice and edited the manuscript.

## Competing interests

A.A. and J.-L.P. are listed as co-inventors on a patent application related to p21TD-Mo-based cellular therapy (p21 expressing monocytes for cancer cell therapy, WO2021013764). J.-L.P. is a founding member of Findimmune SAS, an immuno-oncology biotech company. J.-L.P. discloses research funding not related to this work from NH TherAguix SAS. The remaining authors declare no competing interests.
