## [Peer Review File · Nature Communications]

CDKN1A is a target for phagocytosis-mediated cellular immunotherapy in acute leukemiaREVIEWER COMMENTS

Reviewer #1 (Remarks to the Author); expert on macrophages as therapeutic approach:

In the manuscript entitled “CDKN1A is a target for phagocytosis-mediated cellular immunotherapy in acute leukemia”, the authors investigate the capacity of macrophages to engulf leukemic cells and its potential exploitation for therapeutic purposes. To this aim the authors obtained macrophages by differentiating human monocytes (MDMs) employing a previously described protocol, which is based on the addition to the cell culture medium of 20% human serum. MDMs efficiently engulfed leukemic cells lines (e.g. up to 40% MDMs engulfed leukemic JURKAT cells) but not healthy donor blood lymphocytes. Surprisingly, phagocytosis by MDMs seemed to be independent of the expression of CD47, a known phagocytosis inhibitor, on the leukemic cells. The authors then reported that MDMs that phagocyte leukemic cells in vitro upregulate “pro-inflammatory genes”, including IFN-gamma which in turn could activate non phagocytic (phago-) MDMs in a paracrine manner. The authors found that the CDKN1A-encoded p21 mediates SIRP1-alpha (the receptor for CD47) downregulation, which in turn enables phagocytosis of leukemic cells and that p21 overexpression or down-regulation enhances or inhibit phagocytosis, respectively. The authors then exploited p21-overexpressing MDMs in a cancer therapy model. They found that NSG mice infused with p21-overexpressing MDM mice had increased survival upon challenge with MOLT4 leukemic cells compared to mice infused with control MDM. A similar finding was obtained in mice transplanted with primary leukemic blasts from ALL patients. The authors then conclude that their findings may provide proof-of-principle of a new therapeutic strategy.

The manuscript is of interest and well written. There are, however, some important mechanistic insights that are missing and should be addressed by the authors to justify their interpretations and overall claims.

In vitro studies

1. The authors claim that phago+ MDMs produce IFN-gamma upon phagocytosis of leukemic cells. It will be important to determine if IFN-gamma is indeed secreted at biologically active levels by MDMs, a finding contrary to most current knowledge. An alternative possibility is that IFN-gamma detected by the authors (using cytokine arrays) is derived from the leukemic cells. To this aim the authors should quantify IFN-gamma using ELISA and better document this finding.
2. The authors identify a number of genes differentially expressed in MDMs upon phagocytosis of leukemic cells (Phago+ MDMs) compared to MDMs that did not phagocyte leukemic cells (Phago- MDMs). The authors define some of these genes as pro-inflammatory whereas other genes are called anti-inflammatory. Most of the pro-inflammatory genes are upregulated in Phago+ MDMs. It is unclear what parameters and references the authors are using to assign to these genes a determined inflammatory status, indeed there are some genes defined by the authors as anti-inflammatory, such as HLA genes, CCL2, CCL4, PTX3, CCL8, that are known to be associated with inflammation, therefore pro-inflammatory.
3. The authors claim that p21 activation is not mediated by p53. This conclusion is based on the observation that shRNAs targeting p53 do not impair phagocytosis. However, if one compares the efficiency of the knock down of p21 with that of p53, it is clear that p53 knock down is very inefficient. The authors should use more efficient shRNAs or other strategies to target p53 and thus exclude its role in phagocytosis. The authors could also use a p53 inhibitor.
4. The authors overexpress p21 using a lentiviral vector in MDMs and report that transduced cells had increased phagocytosis of leukemic cells. Is cell cycle and viability of macrophages impaired when overexpressing p21? Is there any correlation between proliferation and phagocytic capacity of MDMs?
5. What is the mechanism whereby SIRP1-alpha expression (suppressed by p21) inhibits macrophage-mediated phagocytosis of leukemic cells, if independent on CD47 expression?

Probably most importantly, there are several outstanding issues pertaining the mechanism(s) underlying the anti-tumor effect observed in vivo in the mouse model that need to be addressed to provide confidence on the proposed interpretations.

6. The authors show that adoptively transferred p21- overexpressing MDMs have a survival effect on immunodeficient mice challenged with MOLT4 ALL cells. According to fig. 3o there would be no cross-activation of human phago- by phago+ MDMs in vivo. One would then question whether the fraction of infused phago+ MDM and their in vivo survival could be sufficient to explain the observed benefit. What is the percentage of macrophages transduced with p21 LV? How long do they survive in vivo? Are mouse macrophages cross-activated by phago+ MDMs? If so, through what mechanism, as human IFN-gamma may have species-specific activity.

7. The authors abate the therapeutic effect of p21-overexpressing MDMs by delivering clodronate after 21 days after MDM transfer. It is known that macrophages need CSF1R activation to survive, whereas mouse M-CSF is not active on human CSF1R. How long do the human MDMs survive in NSG mice? CFSE+ cells in the spleen may not be human MDMs but mouse spleen macrophages that uptake CFSE from apoptotic/debris MDMs. Authors should label hCD45 to identify human MDMs in mice at 21 days post infusion.

8. Is the therapeutic effect mediated by phagocytosis or by a direct effect of human IFN-gamma on the leukemic cells? The authors show that the therapeutic effect can be abolished when using anti-human IFN-gamma antibodies, as shown in Fig.3p. Can they obtain the same effect by administering IFN-gamma and omitting MDM? The authors could also attempt to block CD47 in vivo using specific antibodies; indeed, even if correlation studies performed in vitro suggest that CD47 does not play a role in inhibiting MDM-mediated phagocytoses, CD47 may still play an important function in vivo.

9. In these survival studies the control arm is (correctly) performed by infusing mice with untransduced MDM but there is no data showing the survival of mice not infused with any cell. Do untransduced MDM have an effect?

10. The authors used MDMs as a model of human macrophages. However, it is well established that macrophages are polarized according to the microenvironment (e.g. organ, tissue, etc) where they are found. Authors' finding may not apply to in vivo (endogenously) differentiated macrophages but it would still be interesting as manipulation/interventional strategy. This should be discussed.

Minor comments

1. Abstract: Refer to CDKN1A as p21 once the term p21 has been introduced.

2. The authors exclude efferocytosis as the mechanism of MDM-mediated uptake of leukemic cells, since caspase inhibitor and anti-annexin antibodies did not impair human MDM engulfment of cancer cells (Jurkat and AML blasts). However, CDDP is not described in the results. This figure should be described better in the results section.

3. Authors claim that MDMs were treated with intravenous immunoglobulins (IVIG). Here the IVIG term seems ambiguous. Authors should explain better what IVIGs are and how MDMs were treated.

4. Extended Data Fig. 1e and d are quoted before Extended Data Fig. 1b, c. Authors should quote figures in a chronological order.

5. It is unclear the rationale used by the authors to investigate the role of p21. There are many transcription factors, including the IRF5, that could be in part responsible for the observed phenotype. If possible, the authors should briefly indicate their rationale in the result section before describing the strategy to knock down p21.

Reviewer #2 (Remarks to the Author); expert on T-ALL:

In this work, the authors identified p21 as a trigger of phagocytosis-guided pro-inflammatory reprogramming of TAMs and demonstrate the potential for p21TD-Mo-based cell therapy in cancer immunotherapy. I have a few questions that need to be clarified before this paper can be considered for publication.

Specific points:

1. Figure 1c. MDMs fail to engulf HEL and K562 cells. Why is that?
2. Extended data Fig 1. It is remarkable that leukemic T cells could be refractory to phagocytosis induced by anti-CD47 blockade. What could be the reason for this?
3. Next, the authors compare Phago+MDM with Phago-MDM cells. Although a variety of differences are identified. The authors decided to focus on IFN γ . Why did the authors pick this specific cytokine?
4. Next, the authors focus on p21 to identify molecular mechanisms that regulate the macrophage-mediated phagocytosis of leukemic cells. Why did they pick p21?
5. How would the authors envision clinical translation of their findings in daily clinical practice?

Reviewer #3 (Remarks to the Author); expert on macrophage, immunosurveillance and cancer:

In this manuscript, the authors discovered a novel role of CDKN1A in regulating macrophage phagocytosis of leukemic cells. They showed that CDKN1A (p21) regulated the expression of Sirp α , a receptor for “don’t eat me” signal CD47. This is an exciting discovery and the human monocytes engineered to overexpress p21 demonstrated impressive anticancer efficacy in in vivo MOLT4 cell model and T-ALL PDX models.

While their findings are novel and exciting, the mechanism of p21 in regulating phagocytosis was not well explored in this study, raising certain big concerns. The authors revealed a correlation between p21 and Sirp α expression - knockdown of p21 promoted Sirp α expression whereas overexpression of p21 inhibited Sirp α expression in MDMs. These data would suggest that p21 regulate phagocytosis through Sirp α signaling. However, the data in Fig1 showed that phagocytosis of the leukemic T cells was independent of their CD47 expression and was resistant to CD47 blockade, excluding a possible role of CD47-Sirpa signaling axis in this process. So the interpretation would be that p21 regulates the expression of Sirp α which controls phagocytosis ability of MDM through a novel CD47-independent mechanism - however, this has not been addressed at all in this manuscript.

1. It's very interesting that the authors showed IFN γ secreted by Phago+ MDMs stimulated Phago-MDMs to a pro-inflammatory phenotype. The authors should investigate would such changes of phenotype on Phago- MDMs also enhance their phagocytic ability.
2. Experiments performed in Fig1 were not well connected to the rest of the paper. In addition, the author started to investigate the role of p21 in macrophage phagocytosis but there was a lack of interpretation of why was p21 studied for this purpose and what the connection is between p21 and the experiments performed in Fig1 for the inflammatory phenotype of Phago+ or Phago- MDMs.
3. In fig2, western blot was used for assessing the expression of p21 and Sirp α in MDMs. It would be more informative if FACS can be performed to examine cell surface Sirpa (to evaluate the percentage of cells whose surface expression of Sirpa were impacted) of these MDMs. The efficiency of MDM transduction by lentiviruses should be examined as well – what is the percentage of MDMs that were transduced by lentiviruses?

4. The authors' finding regarding leukemic T cells are resistant to CD47 blockade-induced phagocytosis is contradictory to many of previous studies (There are many studies showing Jurkat cells were efficiently phagocytosed upon CD47 blockade, eg. Weiskopf et al, Science, 2013; Peluso et al, JITC, 2020). In addition, it's very surprising that as shown in Fig1c etc. the phagocytosis rate of Jurkat cells (and MOLT4, CEM, THP1) by resting MDMs (without treatment of antibodies etc.) could reach 20-50%, which seems to be inconsistent with previous studies in which phagocytosis rate by resting MDMs were usually 5-10% or lower.

An interpretation of such inconsistency is needed.

5. It's difficult to interpret the data in Fig2 that p21 regulated phagocytosis through the regulation of Sirp α . If the changes of Sirp α have such significant effects on phagocytosis as shown in Figure 2c-e and Fig2r, why didn't blockade of CD47 have any effects (Extended Fig 1J.), given that CD47 functions through binding to Sirp α to send phagocytosis-inhibitory signals? Were the findings in figure2 due to certain mechanisms independent of CD47- Sirp α axis? If so, such mechanisms should be investigated.

6. Is p21 differentially expressed in Phago+ vs Phago- MDMs? Is the expression of p21 in Phago+ and Phago- MDMs correlated to their phagocytosis capacity?

7. Does p21 depletion or overexpression have an impact on the viability of MDMs? Could the reduction of phagocytic ability of p21 knockdown MDMs be due to their compromised viability?

8. Given that p21 is a multi-functional regulators of macrophage functions, the authors should perform a function rescue experiment to exclude the off-target effects – eg. To express p21 in p21 knockdown MDMs and examine whether phagocytosis can be reversed to a similar level as that of the WT cells.

9. In the experiment depicted in figure 3, a control group is missing – mice only transplanted with MOLT4 cells but not the MDMs. As the authors showed in figure 1, WT MDMs demonstrated significant basal level phagocytosis of MOLT4 cells, therefore, it would be expected that the mice transplanted with Co. TD MDMs should demonstrate certain level of inhibition of MOLT4 cells, as compared to the mice only transplanted with MOLT4 but not MDMs.

Minor issues:

Fig 1g, was the x-axis "phagocytic MDMs - +" mis-labeled?

Fig 1j-m, please indicate how long after the initiation of phagocytosis was the comparison for CD163, IRF5, etc. performed?

Point-by-point response to Reviewer's comments:

Reviewer #1 (expert on macrophages as therapeutic approach)

General critique raised by reviewer #1: *In the manuscript entitled “CDKN1A is a target for phagocytosis-mediated cellular immunotherapy in acute leukemia”, the authors investigate the capacity of macrophages to engulf leukemic cells and its potential exploitation for therapeutic purposes. To this aim the authors obtained macrophages by differentiating human monocytes (MDMs) employing a previously described protocol, which is based on the addition to the cell culture medium of 20% human serum. MDMs efficiently engulfed leukemic cells lines (e.g. up to 40% MDMs engulfed leukemic JURKAT cells) but not healthy donor blood lymphocytes. Surprisingly, phagocytosis by MDMs seemed to be independent of the expression of CD47, a known phagocytosis inhibitor, on the leukemic cells. The authors then reported that MDMs that phagocytose leukemic cells in vitro upregulate “pro-inflammatory genes”, including IFN-gamma which in turn could activate non phagocytic (phago-) MDMs in a paracrine manner. The authors found that the CDKN1A-encoded p21 mediates SIRP1-alpha (the receptor for CD47) downregulation, which in turn enables phagocytosis of leukemic cells and that p21 overexpression or down-regulation enhances or inhibits phagocytosis, respectively. The authors then exploited p21-overexpressing MDMs in a cancer therapy model. They found that NSG mice infused with p21-overexpressing MDM mice had increased survival upon challenge with MOLT4 leukemic cells compared to mice infused with control MDM. A similar finding was obtained in mice transplanted with primary leukemic blasts from ALL patients. The authors then conclude that their findings may provide proof-of-principle of a new therapeutic strategy.*

The manuscript is of interest and well written. There are, however, some important mechanistic insights that are missing and should be addressed by the authors to justify their interpretations and overall claims.

Our response. We thank reviewer #1 for her/his positive constructive critiques and for recognizing the interest of our discovery. The reviewer #1 correctly points out the need for a better understanding of the molecular mechanisms involved in p21-induced, phagocytosis-guided proinflammatory reprogramming of tumor-associated macrophages. In the revised manuscript, a particular attention has been paid to improve and clarify all the issues pointed by reviewer #1.

Major point of critique 1 raised by reviewer #1: *The authors claim that phago⁺ MDMs produce IFN-gamma upon phagocytosis of leukemic cells. It will be important to determine if IFN-gamma is indeed secreted at biologically active levels by MDMs, a finding contrary to most current knowledge. An alternative possibility is that IFN-gamma detected by the authors (using cytokine arrays) is derived from the leukemic cells. To this aim the authors should quantify IFN-gamma using ELISA and better document this finding.*

Our response. We agree with the major point of critique 1 raised by reviewer #1 that the secretion of IFN γ by phagocytic macrophages that we reported in submitted manuscript seems at first glance quite surprising. Despite the fact that IFN γ is known to be primarily produced by NK cells, T helper 1 (T_H1) lymphocytes and CD8⁺ cytotoxic lymphocytes (Hu and Ivashkiy, *Immunity* (2009)), we and others also reported that macrophages can also produce IFN γ in

response to cancer treatments (such as radiotherapy (Wu *et al.*, *Cell Death Differ* (2018))), to cytokine stimulation (Munder *et al.*, *J Exp Med* (1998); Darwich *et al.*, *Immunology*, (2008); Robinson *et al.*, *J. Innate Immun.* (2010)) and pathogens infections (Rothfuchs *et al.*, *J. Immunol*, (2004)). In the submitted manuscript, we assessed the secretion of cytokines, chemokines and chemo-attractants by phagocytic (Phago⁺) and nonphagocytic (Phago⁻) MDMs. We developed a specific methodology to avoid that cytokines, chemokines and chemo-attractants released by leukemic cells should be detected during this assay. As described in the submitted Methods section (pages 16-18), CMFDA-labeled MDMs were cocultured with CMTMR-labeled MOLT4 leukemic cells during 2 hours and sorted by FACS on the basis of their phagocytic activity. Two hours after cell sorting, sorted CMTMR⁺CMFDA⁺(Phago⁺) MDMs and CMTMR⁻CMFDA⁺ (Phago⁻) MDMs were analyzed for phagocytosis using fluorescence microscopy. We observed that all sorted cells were labeled with CMFDA, thus demonstrating that no single (CMTMR⁺) MOLT4 cells were present in sorted cell populations (Point-by-point reply (PPR) Fig. 1a). Ninety-six hours after cell sorting, we also analyzed purified MDMs and noticed that no CMTMR⁺MOLT4 cells were detected in both Phago⁺ and Phago⁻ cell populations (PPR Fig. 1a). These results support the specificity of our cell sorting and demonstrate that mCherry⁺ MOLT4 cells that were engulfed by MDMs did not escape lysosomal degradation (as shown in submitted Extended Fig. 1a), thus excluding the possibility that MOLT4-derived cytokines, chemokines and chemoattractants may be detected during this assay. This experimental procedure allowed us to demonstrate that after MOLT4 cell engulfment and degradation macrophages secreted proinflammatory cytokines (such IL1 β , IL6, IL23, IL27, GM-CSF, MIF and IFN γ) (submitted Fig. 1l,m and extended Fig. 2a), chemokines (IL8 and GRO- α) (submitted Fig. 1l and extended Fig. 2a), the chemo-attractant SERPIN E1 (submitted Fig. 1m) and the compensatory anti-inflammatory cytokine IL1-ra (submitted extended Fig. 2a). According to reviewer 1's recommendation, we also confirmed using ELISA that Phago⁺ MDMs secreted IFN γ 7 days after cell sorting and showed that Phago⁻ MDMs that did not (PPR Fig. 1b).

Point-by-point reply Figure 1. Phagocytic macrophages secrete IFN γ after degradation of engulfed leukemic cells. *a*, Percentage of CMFDA⁺ MDMs engulfing CMTMR⁺ MOLT4 (CMTMR⁺CMFDA⁺ MDMs) detected 2 h and 96 h after the sorting of Phago⁻ MDMs and Phago⁺ MDMs (as shown in submitted Fig. 1e-i) (**** p <0.0001). *b*, IFN γ secretion determined by ELISA quantification, 7 d after the sorting of Phago⁻ MDMs and Phago⁺ MDMs. These cells were assessed for the degradation of engulfed MOLT4 cells at 96h (*a*), extensively washed and supplemented with fresh medium for additional 3 days (p =0.0008). In (*a*, *b*) the data are presented as the mean \pm SEM from n =3 donors. *** p <0.001 and **** p <0.0001; determined with ANOVA with Tukey's multiple comparison test (*a*) and unpaired two-tailed t test (*b*).

NM_020689	SLC24A3	2.509869054	2.487945778	0.24417411	0.29596393	0.855669424	0.686652881	1.180045863	1.047391562	23
NM_016459	MZBF1	2.583294848	2.596940739	0.723107334	0.847418516	0.723107334	0.847418516	1.386897881	0.933715208	24
NM_014220	TM4SF1	3.684633887	3.664781802	1.28603437	1.322213046	1.28603437	1.322213046	2.094316753	1.234284799	
NM_002648	PMI1	2.426849072	2.423825868	0.429656285	0.366686471	1.503533027	1.466063122	1.436102308	0.907656914	
NM_001145033	AG2	3.301542123	3.32959808	1.297876972	1.280885528	1.297876972	1.280885528	1.994770334	1.046389336	
NR_026597	DIRC3	2.407209551	2.404885824	0.272469241	0.05800101	0.792124093	0.832259623	1.127791557	1.033770692	
NM_001172292	NIPAL4	3.697098425	3.724208584	0.386233055	0.357622115	2.441272989	2.528298357	2.189122254	1.510887475	
ENST00000380464	PLIN2	2.870003562	2.72541771	0.871565548	0.813998109	1.432938739	1.492002491	1.700987693	0.895128106	
NM_033120	NKD2	2.411773556	2.215007278	2.978825795	2.738906968	1.21440627	1.288940654	2.141310087	0.737945891	
NM_002404	MFAP4	2.736404754	2.75575098	1.847561408	1.806745917	0.977980575	1.007566721	1.855335059	0.784591106	
NM_021158	TRIB3	3.216638594	3.232921566	0.211681968	0.189167337	0.153481855	0.188647295	1.198756436	1.569470086	25
NM_139314	ANGPT14	3.330031124	3.44457626	1.140097981	0.91885867	1.183578945	1.19680129	1.868990712	1.180912556	
NM_000597	IGFBP2	1.396431888	1.434434584	2.865212024	2.81262923	1.226250159	1.166577261	1.816922524	0.798159749	
NM_001853	COL3A1	1.076008673	1.19733003	3.400763795	3.217354319	1.457647379	1.231997201	1.930183566	1.076741237	
NM_001005464	HIST1H3A	0.233668323	0.164024855	2.432292745	2.451638157	1.347452045	1.439846691	1.344820469	1.004551206	
NM_018667	SMPD3	1.538149607	1.624892199	3.043850524	2.911168449	0.859509374	1.013238869	1.83180147	0.935792179	26
NM_018492	PBK	0.354317436	0.235393346	2.579898956	2.694358302	0.529973589	0.422286034	1.136037545	1.167226926	
NM_004867	ITMLA	2.267706361	2.28824222	2.570112004	2.646682524	0.152224439	0.197676711	1.687107376	1.180949555	
NR_033916	CCDC26	1.467317138	1.522804404	3.902704634	3.934413137	1.358918834	1.179927602	2.227647291	1.315025728	
NM_001004317	LN28B	1.271591361	1.399339878	3.158333171	3.24915891	1.364422982	1.548020693	1.998144499	0.938460457	27
NM_002612	PDR4	1.884871379	1.962521516	1.552354138	1.426121895	2.388744452	2.28023544	1.915808136	0.382493573	28

Fold changes (FC) are shown.

References of PPR Table 1

- Kang, K. *et al.* Interferon-gamma Represses M2 Gene Expression in Human Macrophages by Disassembling Enhancers Bound by the Transcription Factor MAF. *Immunity* **47**, 235-250 e234, doi:10.1016/j.immuni.2017.07.017 (2017).
- Martinez, F. O., Gordon, S., Locati, M. & Mantovani, A. Transcriptional profiling of the human monocyte-to-macrophage differentiation and polarization: new molecules and patterns of gene expression. *J Immunol* **177**, 7303-7311, doi:10.4049/jimmunol.177.10.7303 (2006).
- Honore, C. *et al.* The innate pattern recognition molecule Ficolin-1 is secreted by monocytes/macrophages and is circulating in human plasma. *Mol Immunol* **45**, 2782-2789, doi:10.1016/j.molimm.2008.02.005 (2008).
- Carson, W. F. *et al.* Enhancement of macrophage inflammatory responses by CCL2 is correlated with increased miR-9 expression and downregulation of the ERK1/2 phosphatase Dusp6. *Cell Immunol* **314**, 63-72, doi:10.1016/j.cellimm.2017.02.005 (2017).
- Cuevas, V. D. *et al.* MAFB Determines Human Macrophage Anti-Inflammatory Polarization: Relevance for the Pathogenic Mechanisms Operating in Multicentric Carpotalar Osteolysis. *J Immunol* **198**, 2070-2081, doi:10.4049/jimmunol.1601667 (2017).
- Gundra, U. M. *et al.* Alternatively activated macrophages derived from monocytes and tissue macrophages are phenotypically and functionally distinct. *Blood* **123**, e110-122, doi:10.1182/blood-2013-08-520619 (2014).
- Liao, X. *et al.* Kruppel-like factor 4 regulates macrophage polarization. *J Clin Invest* **121**, 2736-2749, doi:10.1172/JCI45444 (2011).
- Desterke, C., Turhan, A. G., Bennaceur-Griscelli, A. & Griscelli, F. PPARgamma Cistrome Repression during Activation of Lung Monocyte-Macrophages in Severe COVID-19. *iScience* **23**, 101611, doi:10.1016/j.isci.2020.101611 (2020).

9. Yasuda, K., Nakanishi, K. & Tsutsui, H. Interleukin-18 in Health and Disease. *Int J Mol Sci* **20**, doi:10.3390/ijms20030649 (2019).
10. Na, Y. R. *et al.* Protein Kinase A Catalytic Subunit Is a Molecular Switch that Promotes the Pro-tumoral Function of Macrophages. *Cell Rep* **31**, 107643, doi:10.1016/j.celrep.2020.107643 (2020).
11. Jha, A. K. *et al.* Network integration of parallel metabolic and transcriptional data reveals metabolic modules that regulate macrophage polarization. *Immunity* **42**, 419-430, doi:10.1016/j.immuni.2015.02.005 (2015).
12. Schridde, A. *et al.* Tissue-specific differentiation of colonic macrophages requires TGFbeta receptor-mediated signaling. *Mucosal Immunol* **10**, 1387-1399, doi:10.1038/mi.2016.142 (2017).
13. Ashley, J. W. *et al.* Polarization of Macrophages toward M2 Phenotype Is Favored by Reduction in iPLA2beta (Group VIA Phospholipase A2). *J Biol Chem* **291**, 23268-23281, doi:10.1074/jbc.M116.754945 (2016).
14. Granata, F. *et al.* Production of vascular endothelial growth factors from human lung macrophages induced by group IIA and group X secreted phospholipases A2. *J Immunol* **184**, 5232-5241, doi:10.4049/jimmunol.0902501 (2010).
15. Orecchioni, M., Ghosheh, Y., Pramod, A. B. & Ley, K. Macrophage Polarization: Different Gene Signatures in M1(LPS+) vs. Classically and M2(LPS-) vs. Alternatively Activated Macrophages. *Front Immunol* **10**, 1084, doi:10.3389/fimmu.2019.01084 (2019).
16. Tian, X. *et al.* CXCR4 knockdown prevents inflammatory cytokine expression in macrophages by suppressing activation of MAPK and NF-kappaB signaling pathways. *Cell Biosci* **9**, 55, doi:10.1186/s13578-019-0315-x (2019).
17. Simmons, D. P. *et al.* SLAMF7 engagement superactivates macrophages in acute and chronic inflammation. *Sci Immunol* **7**, eabf2846, doi:10.1126/sciimmunol.abf2846 (2022).
18. Yoon, B. R., Oh, Y. J., Kang, S. W., Lee, E. B. & Lee, W. W. Role of SLC7A5 in Metabolic Reprogramming of Human Monocyte/Macrophage Immune Responses. *Front Immunol* **9**, 53, doi:10.3389/fimmu.2018.00053 (2018).
19. Sierra-Filardi, E. *et al.* Activin A skews macrophage polarization by promoting a proinflammatory phenotype and inhibiting the acquisition of anti-inflammatory macrophage markers. *Blood* **117**, 5092-5101, doi:10.1182/blood-2010-09-306993 (2011).
20. Wang, T. *et al.* HIF1alpha-Induced Glycolysis Metabolism Is Essential to the Activation of Inflammatory Macrophages. *Mediators Inflamm* **2017**, 9029327, doi:10.1155/2017/9029327 (2017).
21. Borrell-Pages, M., Romero, J. C., Juan-Babot, O. & Badimon, L. Wnt pathway activation, cell migration, and lipid uptake is regulated by low-density lipoprotein receptor-related protein 5 in human macrophages. *Eur Heart J* **32**, 2841-2850, doi:10.1093/eurheartj/ehr062 (2011).

22. Xue, J. *et al.* Transcriptome-based network analysis reveals a spectrum model of human macrophage activation. *Immunity* **40**, 274-288, doi:10.1016/j.immuni.2014.01.006 (2014).
23. Villa-Bellosta, R., Hamczyk, M. R. & Andres, V. Novel phosphate-activated macrophages prevent ectopic calcification by increasing extracellular ATP and pyrophosphate. *PLoS One* **12**, e0174998, doi:10.1371/journal.pone.0174998 (2017).
24. Zhang, H., Chen, X. & Sairam, M. R. Novel hormone-regulated genes in visceral adipose tissue: cloning and identification of proinflammatory cytokine-like mouse and human MEDA-7: implications for obesity, insulin resistance and the metabolic syndrome. *Diabetologia* **54**, 2368-2380, doi:10.1007/s00125-011-2212-7 (2011).
25. Riera-Borrull, M. *et al.* Palmitate Conditions Macrophages for Enhanced Responses toward Inflammatory Stimuli via JNK Activation. *J Immunol* **199**, 3858-3869, doi:10.4049/jimmunol.1700845 (2017).
26. Al-Rashed, F. *et al.* Neutral sphingomyelinase 2 regulates inflammatory responses in monocytes/macrophages induced by TNF-alpha. *Sci Rep* **10**, 16802, doi:10.1038/s41598-020-73912-5 (2020).
27. Jaiswal, A., Maurya, M., Maurya, P. & Barthwal, M. K. Lin28B Regulates Angiotensin II-Mediated Let-7c/miR-99a MicroRNA Formation Consequently Affecting Macrophage Polarization and Allergic Inflammation. *Inflammation* **43**, 1846-1861, doi:10.1007/s10753-020-01258-1 (2020).
28. Min, B. K. *et al.* Pyruvate Dehydrogenase Kinase Is a Metabolic Checkpoint for Polarization of Macrophages to the M1 Phenotype. *Front Immunol* **10**, 944, doi:10.3389/fimmu.2019.00944 (2019).

Point-by-point reply Table 1. List of upregulated and downregulated genes in Phago⁺ MDMs as compared to Phago⁻ MDMs (in PPR Fig. 2).

The submitted Fig. 2j was replaced by the PPR Fig. 2 (revised Fig. 1j) and the submitted Extended Table 2 was corrected in the revised version of our manuscript (revised Supplementary Table 2), according to the major point of critique 2 raised by reviewer #1.

Major point of critique 3 raised by reviewer #1: The authors claim that p21 activation is not mediated by p53. This conclusion is based on the observation that shRNAs targeting p53 do not impair phagocytosis. However, if one compares the efficiency of the knock down of p21 with that of p53, it is clear that p53 knock down is very inefficient. The authors should use more efficient shRNAs or other strategies to target p53 and thus exclude its role in phagocytosis. The authors could also use a p53 inhibitor.

Our response. We agree with reviewer #1. The knockdown of p53 shown in submitted Extended Data Fig. 3a appears inefficient. To improve p53 knockdown efficacy, we set up new transfection conditions and used a smart pool siRNAs containing four specific siRNAs. As shown in PPR Fig. 3, we efficiently knockdown p53 in MDMs and observed that p53 knockdown strongly reduces p21 protein expression (PPR Fig. 3a) without affecting the viability of MDMs (PPR Fig. 3b). Accordingly, the phagocytosis of MOLT4 cells by p53-depleted MDMs was significantly inhibited (PPR Fig. 3c). We also confirmed these results

using a pharmacological inhibitor of p53, Pifithrin- α (PFT α) (PPR Fig. 3d-f). Altogether, these results demonstrate that p53-dependent p21 expression dictates the phagocytosis of the leukemic cells by MDMs.

Point-by-point reply Figure 3. p53 regulates p21 protein expression and dictates macrophage phagocytosis of leukemic cells. *a-c*, Expressions of p53 and p21 by WB analysis (*a*), cell proliferation and viability assessed by a WST-1 assay (*b*) and phagocytosis percentage of MOLT4 cells after 8 h coculture (***p*=0.001) (*c*) in p53 knockdown MDMs (*sip53*) or control MDMs (*siCo.*) 24 h after siRNAs transfection. *d,e*, Expressions of p53 and p21 by WB analysis (*d*) and cell proliferation and viability assessed by WST-1 assay (*e*) in pifithrin- α (PFT α)-treated (10 μ M) or control MDMs 24h after treatments. *f*, Phagocytosis percentage of MOLT4 cells after 8 h coculture with PFT α -pretreated MDMs (**p*=0.0270). In (*a*, *d*), the data are representative of *n*=3 donors. In (*b*, *c*, *e*, *f*), the data are donor matched from *n*=3 donors. **p* < 0.05 and ***p* < 0.01; determined with two-tailed (*c*) and one-tailed (*f*) paired *t* tests.

We only added the results of p53 knockdown (PPR Fig. 3a-c) in the revised Supplementary Fig. 5a-c.

Major point of critique 4 raised by reviewer #1: *The authors overexpress p21 using a lentiviral vector in MDMs and report that transduced cells had increased phagocytosis of leukemic cells. Is cell cycle and viability of macrophages impaired when overexpressing p21? Is there any correlation between proliferation and phagocytic capacity of MDMs?*

Our response. In response to the major point of critique 4 raised by reviewer #1, we analyzed the cell cycle progression, the proliferation and the viability of p21-depleted MDMs using siRNAs (as shown in submitted Fig. 2a), p21-overexpressing MDMs using lentiviral vectors (as shown in submitted Fig. 2p) and control MDMs at indicated times after siRNA transfection (PPR Fig. 4a-c, g, i) or lentiviral transduction (PPR Fig. 4d-f, h, j). We observed that control (*siCo.* or *Co.TD*), p21-depleted (*sip21*) or p21-overexpressing (*p21TD*) MDMs were mainly arrested in G0/G1 phase (PPR Fig. 4a-f), did not divide (PPR Fig. 4g, h) and were not affected in their viability (PPR Fig. 4i, j) until 30 days after transfection or transduction, thus demonstrating that the modulation of p21 expression did not alter the

terminal differentiation and the survival of macrophages. In addition, these results also reveal that the phagocytic capacity of MDMs is independent of MDMs proliferation.

Point-by-point reply Figure 4. p21 knockdown or overexpression does not affect cell cycle, proliferation and viability of macrophages. *a-c*, FACS histograms (*a, b*) and percentages (*c*) of cell phases (G0/G1, S, G2/M) analyzed by DNA content staining with propidium iodide (PI) of control (siCo.) or p21 knockdown (sip21) MDMs, as shown in **submitted Fig. 2a**, at 24 h, 15 d and 30 d after siRNA transfection. *d-f*, FACS histograms (*d, e*) and percentages (*f*) of cell phases (G0/G1, S, G2/M) analyzed by DNA content staining with PI of MDMs derived from Co.TD or p21TD genetically engineered monocytes and differentiated into MDMs for 7d, as shown in **submitted Fig. 2p**, at 7 d, 15 d and 30 d after lentiviral transductions. *g, i*, Cell proliferation assessed by a WST-1 assay (*g*) and viability determined by counting viable cells (*i*) of siCo. or sip21 MDMs at 24 h, 15 d and 30 d after siRNAs transfections. *h, j*, Cell proliferation assessed by a WST-1 assay (*h*) and viability determined by counting viable cells numbers (*j*) of Co.TD or p21TD MDMs at 7 d, 15 d and 30 d after lentiviral transductions. In (*a, b, d, e*) the data are representative of $n=3$ donors. The data in (*c, f, g-j*) are presented as the mean \pm SEM from $n=3$ donors.

These results were added in the revised Supplementary Fig. 4 and 8.

Major point of critique 5 raised by reviewer #1: What is the mechanism whereby SIRP1- α expression (suppressed by p21) inhibits macrophage-mediated phagocytosis of leukemic cells, if independent on CD47 expression? Probably most importantly, there are several outstanding issues pertaining the mechanism(s) underlying the anti-tumor effect observed in vivo in the mouse model that need to be addressed to provide confidence on the proposed interpretations.

Our response. We agree with the major point of critique 5 raised by reviewer #1. In the submitted manuscript, we proposed that macrophage-mediated phagocytosis of leukemia cells could be independent of CD47 expression (submitted Extended Data Fig. 1h-j) and thus, could be in conflict with previous publications (Majeti *et al.*, *Cell*, 2009; Weiskopf *et al.*, *Science* (2013); Peluso *et al.*, *J Immunother Cancer*, (2020)). To better characterize the relationship between CD47 expression on leukemic cells and p21-dependent tumor phagocytosis, we first compared the experimental procedure that we used in our phagocytosis assays to those previously published (Majeti *et al.*, *Cell*, 2009; Chao *et al.*, *Cancer Res*, 2011) and then, we determined the effect of CD47 depletion in leukemia cells on the macrophage-mediated phagocytosis. We agree that the phagocytosis rates of leukemic cells by macrophages (ranging from 15% to 45%) that we observed in our control cocultures (submitted Fig. 1c and Extended Data Fig. 1j) were higher than those previously reported (ranging from 2% to 10%) (Majeti *et al.*, *Cell*, 2009; Chao *et al.*, *Cancer Res*, 2011; Weiskopf *et al.*, *Science*, 2013; Peluso *et al.*, *J Immunother Cancer*, 2020). As indicated in the submitted Methods section, our phagocytosis assays were performed in 10% Heat Inactivated (HI) serum-supplemented medium and analyzed after 8 hours, while previous published assays were performed with macrophages that were cultured for 2 hours in serum-free medium prior the phagocytosis assays (performed in the same culture medium) and analyzed after 2 hours. To determine the effects of these differences in our experimental procedures on phagocytosis rates, we explored the expressions of p21 and SIRP α in MDMs that were cultured in absence or in presence of 10% HI serum. We observed after two-hours without serum that p21 protein expression was decreased (PPR Fig. 5a), and both protein (PPR Fig. 5a) and cell surface (PPR Fig. 5b) expressions of SIRP α strongly increased, as compared to MDMs that were cultured in presence of serum, thus revealing that the culture conditions positively or negatively impact the phagocytic activity of MDMs. Indeed, the increased expression of SIRP α positively correlated with low phagocytosis rates of MOLT4 cells detected in serum-free phagocytosis assays with respect to phagocytosis assays performed in presence of serum (PPR Fig. 5c). Furthermore, in serum-free phagocytosis assays, cocultures in presence of anti-CD47 antibodies showed a significant enhancement of macrophage-mediated phagocytosis of MOLT4 cells, while phagocytosis assays performed in presence of serum did not increase macrophage-mediated phagocytic activity, as compared to control cocultures performed in presence of isotype control (IgG) (PPR Fig. 5c). These results indicate that the apparent discrepancy between our results (submitted Extended Data Fig. 1j) and previous publications (Majeti *et al.*, *Cell*, 2009; Chao *et al.*, *Cancer Res*, 2011) remains on the experimental procedures that were used. Nevertheless, the results obtained in absence of serum suggested that macrophage-mediated phagocytosis of leukemic cells may be dependent of CD47.

To further emphasize the role of CD47 during macrophage-mediated phagocytosis, MOLT4 cells were transduced with lentiviral vectors encoding for control or specific CD47 CRISPR guide RNAs (gRNAs) and CAS9 gene. Then, MDMs were cocultured with stably CD47-depleted (CrCD47MOLT4) and control (CrCo.MOLT4) MOLT4 cells (PPR Fig. 5d, e) and analyzed for tumor phagocytosis. Given that the depletion of CD47 in MOLT4 cells did not affect their proliferation (PPR Fig. 5f, g) and that the percentage and the efficiency of the phagocytosis of CrCD47MOLT4 by MDMs were significantly enhanced with respect to CrCo.MOLT4 cells (PPR Fig. 5h-j), these results demonstrate that the phagocytosis of leukemic T cells by MDMs is inhibited by the expression of CD47 in target cells. To further investigate the molecular mechanisms regulating p21-dependent phagocytosis of leukemia cells, p21-upregulating (p21TD) or control (Co.TD) MDMs were cocultured with control (CrCo.) or CD47-depleted (CrCD47) MOLT4 cells and analyzed for phagocytosis. The

cocultures of p21TD-MDMs with CrCo.MOLT4 cells exhibited a significant increase of the percentage of phagocytic MDMs, while the cocultures of Co.TD-MDMs or p21TD-MDMs with CrCD47MOLT4 cells did not show significant enhancement in the phagocytosis of the leukemia cells (PPR Fig. 5k). Accordingly, coculturing p21TD-MDMs with MOLT4 cells in the presence of anti-CD47 blocking antibodies did not increase the phagocytosis of MOLT4 cells compared to coculturing p21TD-MDMs with MOLT4 cells in the presence of isotype control (IgG) or coculturing Co.TD-MDMs with MOLT4 cells in the presence of anti-CD47 blocking antibodies (PPR Fig. 5l). To appreciate the antitumor effect of the combination of p21TD-Mo-based cellular therapy with CD47 blockade (Majeti *et al.*, *Cell* (2009)), mCherry⁺ MOLT4 cell-engrafted NSG mice were treated 15 days after the adoptive transfer of p21TD-Mos, SIRP α TD-Mos, p21+SIRP α TD-Mos or Co.TD-Mos with anti-CD47 antibodies. Anti-CD47 antibodies significantly prolonged the survival of treated mice, with the exception of the mice infused with p21TD-Mos, which did not show an additional enhancement of survival compared with that of the corresponding control mice (PPR Fig. 5m). These results were confirmed by adoptively transferring p21TD-Mos or Co.TD-Mos into NSG mice engrafted with CD47-depleted (CrCD47MOLT4) or control (CrCo.MOLT4) MOLT4 cells (PPR Fig. 5n). Altogether, these results revealed that p21 overexpression enables the phagocytosis of MOLT4 cells through the disruption of antiphagocytic CD47-SIRP α axis on the side of phagocytic macrophages and highlighted that the adoptive transfer of p21TD-Mos promotes the phagocytosis of CD47-expressing leukemia cells *in vivo* and significantly inhibits leukemia progression through the abrogation of antiphagocytic CD47-SIRP α axis by repressing SIRP α expression in TAMs differentiated from p21TD-Mos.

Point-by-point reply Figure 5. CD47 dictates p21-mediated macrophage phagocytosis and the survival of mCherry+MOLT4-engrafted mice. a, b, p21 and SIRP α expressions analyzed by WB (a) and SIRP α cell-surface expression (MFI) determined by FACS (b) on MDMs cultured for 2 h in serum-free or 10% HI serum-supplemented culture medium before phagocytosis assays. c, The percentage of phagocytosis of MOLT4 cells by MDMs precultured for 2 h in serum-free or 10% HI serum-supplemented culture medium and cocultured in the same media for an additional 2 h in the presence of an isotype control (IgG) or anti-CD47 blocking antibody (B6H12.2) (7 μ g/ml) (* p =0.0271, * p =0.0004, **** p <0.0001). d, e, FACS histograms (d) and cell-surface expression (MFI) (**** p <0.0001) (e) of human CD47 (hCD47) on stable CD47-depleted (CrCD47MOLT4) or control (CrCo.MOLT4) MOLT4 cells obtained through transduction with lentiviral vectors encoding specific CD47 CRISPR guide RNAs (gRNAs) and the CAS9 gene or with control lentiviral vectors, respectively. f, g, Cell proliferation of CrCD47 and CrCo. MOLT4 cells assessed by a WST-1 assay (**** p <0.0001) (f) and viable cell number determination (**** p <0.0001) (g) on d 1, 2 and 5. h, Confocal micrographs of CMFDA-labeled MDMs and CMTMR-labeled CrCD47 or CrCo. MOLT4 cells cocultured for 8 h. i, j, Percentages of phagocytosis of CrCD47 or CrCo. MOLT4 cells by MDMs after 8 h coculture**

(* $p=0.0307$) (i) and the related percentage of Phago⁺ MDMs that engulfed more than one MOLT4 target cells (** $p=0.0056$) (j). **k**, Percentages of stably CD47-depleted MOLT4 (CrCD47MOLT4) or control MOLT4 (CrCo.MOLT4) cell phagocytosis, after 8 h of coculture, by MDMs transduced with control (Co.TD) or p21-expressing (p21TD) lentiviral vectors after 72 h transduction (** $p=0.0061$, ** $p=0.0053$, ** $p=0.0048$). **l**, The percentage of phagocytosis of MOLT4 cells by Co.TD-MDMs or p21TD-MDMs at 72 h after lentiviral transduction; the MDMs were precultured for 2 h in serum-free medium before the phagocytosis assay was performed with MOLT4 cells in the same medium for an additional 2 h in the presence of an isotype control (IgG) or anti-CD47 blocking antibody (B6H12.2) (7 $\mu\text{g/ml}$) (**** $p<0.0001$). **m**, Survival of engrafted mice that received Co.TD-Mos, p21TD-Mos, SIRP α TD-Mos or p21+SIRP α TD-Mos and were treated on d 15 after Mo transfer for 14 d with daily injections of isotype control (IgG) or anti-CD47 blocking antibodies (100 $\mu\text{g/mouse}$) (**** $p<0.0001$). **n**, Survival of NSG mice engrafted with stable CD47-depleted (CrCD47MOLT4) or control (CrCo.MOLT4) MOLT4 cells that received Co.TD-Mos or p21TD-Mos (**** $p<0.0001$, ** $p=0.0054$). In (a, h) and (d), the data are representative data of $n=3$ donors or $n=3$ independent experiments, respectively. In (b, i, j), the data are donor matched from $n=3$ donors. In (c, k, l) and (e, f, g) the data are presented as the mean \pm SEM from $n=3$ donors or $n=3$ independent experiments, respectively. In (m, n), survival data are from $n=5$ mice/group. * $p<0.05$, ** $p<0.01$ and **** $p<0.0001$; determined with two-tailed unpaired *t* (b,e), two-tailed (i) and one-tailed (j) paired tests, with ANOVA with Tukey's multiple comparison test (f, g), with ANOVA with Sidak's multiple comparison test (k, l) and Mantel-Cox test (m, n).

These results were added in the revised Fig. 2g, 3n and the revised Supplementary Fig. 2, Supplementary Fig. 6 and Supplementary Fig. 15.

Major point of critique 6 raised by reviewer #1: The authors show that adoptively transferred p21-overexpressing MDMs have a survival effect on immunodeficient mice challenged with MOLT4 ALL cells. According to fig. 3o there would be no cross-activation of human phago⁻ by phago⁺ MDMs *in vivo*. One would then question whether the fraction of infused phago⁺ MDM and their *in vivo* survival could be sufficient to explain the observed benefit. What is the percentage of macrophages transduced with p21 LV? How long do they survive *in vivo*? Are mouse macrophages cross-activated by phago⁺ MDMs? If so, through what mechanism, as human IFN-gamma may have species-specific activity.

Our response. We thank reviewer #1 for her/his positive constructive critiques. Despite the fact that we initially revealed that Phago⁺ MDMs underwent a proinflammatory activation and also triggered the proinflammatory activation of neighboring Phago⁻ MDMs *in vitro* (submitted Fig. 1j-m and PPR Fig. 1 and 2), we agree that the results showing that after 21 days of monocyte transfer, Phago⁺ MDMs obtained from mCherry⁺ MOLT4-engrafted NSG mice exhibited a proinflammatory activation without affecting neighboring Phago⁻ MDMs (as revealed by the expression of CD163 (submitted Fig. 3o)), could suggest that the proinflammatory activation of neighboring MDMs may not occur *in vivo*. In this context, we studied at different time points (day 21 and 35) the occurrence of the proinflammatory reprogramming of Phago⁺ and Phago⁻ MDMs *in vivo*. Fluorescence microscopy (at d 21 and 35) analysis of FACS-sorted nonphagocytic CFSE⁺ (Phago⁻) or phagocytic (Phago⁺) mCherry⁺CFSE⁺ MDMs from the spleen and the bone marrow of treated mice showed, respectively, that Phago⁺ MDMs highly expressed iNOS (PPR Fig. 6a, b). These results confirm that *in vivo*, phagocytosis of leukemic cells promoted the activation of MDMs toward a proinflammatory phenotype, as shown *in vitro* (submitted Fig. 1j-m and PPR Fig. 1 and 2).

In contrast to MDMs sorted from tumor-free NSG mice that were infused with Co.TD-Mos or p21TD-Mos (PPR Fig. 6c), Phago⁻ MDMs sorted from mCherry⁺ MOLT4-engrafted NSG mice that were injected with p21TD-Mos exhibited a significant increased iNOS expression at day 35 (PPR Fig. 6d), thus suggesting that Phago⁺ MDMs support the proinflammatory reprogramming of surrounding TAMs in the TME.

Point-by-point reply Figure 6. Prophylactic adoptive transfer of p21TD-Mos triggers the proinflammatory activation of TAMs. **a**, Schematic representation showing prophylactic adoptive transfer of CFSE-labeled, control (Co.TD) or p21 (p21TD) genetically engineered human monocytes (Mos) into total body irradiated (TBI) NSG mice which will be engrafted after 7d with mCherry⁺ MOLT4 cells. Then, Co. TD or p21TD (CFSE⁺) MDMs, Co. TD or p21TD Phago⁻ (mCherry⁻CFSE⁺) MDMs, or Co. TD or p21TD Phago⁺ (mCherry⁺CFSE⁺) MDMs were sorted by FACS and analyzed for iNOS expression. **b-d**, Percentages of iNOS-expressing (iNOS⁺) cells, determined by immunofluorescence staining, among Phago⁺ MDMs (**b**), MDMs (**c**) or Phago⁻ MDMs (*****p*<0.0001, ****p*= 0.0004) (**d**) sorted from tumor-free (**c**) or mCherry⁺ MOLT4 cell-engrafted (**b, d**) NSG mice that received Co.TD-Mos or p21TD-Mos; cells collected on d 21 and 35 after monocyte transfer. In (**b-d**), the data are presented as the mean±SEM from *n*=3 mice/group. ****p*<0.001 and *****p*<0.0001; determined with ANOVA with Tukey's multiple comparison test (**d**).

To test this hypothesis, FACS-sorted Phago⁻ and Phago⁺ MDMs from Co.TD-Mo- or p21TD-Mo-infused, mCherry⁺ MOLT4 cell-engrafted NSG mice were seeded in the upper chambers of Transwell devices and cocultured with hCD14⁺hCD11b⁺hCD163⁺CFSE⁺ MDMs (TAMs) (from mCherry⁺ MOLT4 cell-engrafted NSG mice infused with CFSE⁺ human Mos) in the bottom chambers for 15 days; then, and iNOS expression was analyzed (PPR Fig. 7a). In accordance with our *in vitro* findings (submitted Fig. 1n), TAMs were cocultured with Phago⁺ MDMs sorted from tumor-bearing mice that were injected with Co.TD-MDMs or p21TD-MDMs showed a significant increase of iNOS expression (PPR Fig. 7b). We noticed that TAMs that were cocultured with Phago⁺ p21TD-MDMs exhibited significantly increased expression of iNOS compared to those that were cocultured with Phago⁺ Co.TD-MDMs.

Point-by-point reply Figure 7. p21TD-Mo-based therapy triggers proinflammatory reprogramming of TAMs in human T-ALL model. *a*, Schematic representation showing the experimental procedure used to determine the percentage of iNOS⁺ TAMs, using immunofluorescence microscopy. hCD14⁺hCD11b⁺hCD163⁺CFSE⁺ MDMs (TAMs) were sorted from the BM and spleen of mCherry⁺ MOLT4 cell-engrafted NSG mice by FACS at day 35 after human Mo transfer, and cocultured in the bottom chamber of Transwell devices with Phago⁻ MDMs or Phago⁺ MDMs (in the upper chambers), which were sorted from the BM and the spleen of mCherry⁺ MOLT4 cell-engrafted NSG mice adoptively transferred with Co.TD-Mos or p21TD-Mos (cells collected at 35 d after Mo transfer). Control TAMs were cocultured with Co.TD-MDMs or p21TD-MDMs obtained by in vitro differentiation of Co.TD-Mos or p21TD-Mos. After 15 d of coculture, the percentage of iNOS⁺ TAMs was determined by immunofluorescence staining and microscopy. *b*, Percentages of iNOS⁺ TAMs, determined by immunofluorescence staining as shown in (a). The data are presented as the mean±SEM from n=3 mice/group. ****p<0.0001; determined with ANOVA with Sidak's multiple comparison test.

To exclude a potential role of mouse macrophages in the antitumor effect elicited by p21TD-Mo-based therapy, the expression of proinflammatory mouse macrophage markers (murine cell-surface MHCII and iNOS) were analyzed. No increased expression of those markers was detected in mCD45⁺mF4/80⁺ mouse macrophages sorted on day 21 and 35 after Mo transfer from the spleen and BM of mCherry⁺ MOLT4 cell-engrafted or tumor-free NSG mice adoptively infused with Co.TD-Mos or p21TD-Mos as compared to corresponding cells sorted from control mice (PPR Fig. 8), suggesting that the observed bystander proinflammatory activation of human TAMs (PPR Fig. 6d and 7b) mainly depended on factors secreted by the p21TD-Phago⁺ MDMs. Altogether, these results indicate that p21TD-Mo-based cellular therapy drives the engraftment of p21-transduced phagocytes, which, in addition to directing the elimination of leukemia cells, triggers the secretion of the proinflammatory cytokines such as IFN γ (submitted Fig. 1o) and supports the proinflammatory reprogramming of TAMs, which in turn participate in the regression of leukemia (submitted Fig. 3p).

Point-by-point reply Figure 8. p21TD-Mo-based therapy does not trigger the proinflammatory activation of murine macrophages in the human T-ALL model. a-j, Cell-surface expression (MFI) of murine MHCII (mMHCII) and the percentage of cells expressing iNOS (iNOS⁺) were analyzed by FACS (a-d) and immunofluorescence staining and microscopy (e-h) of murine macrophages (mCD45⁺mF4/80⁺). Murine macrophages were sorted from the BM (a, b, e, f) and the spleen (c, d, g, h) of tumor-free (a, c, e, g) or mCherry⁺MOLT4 cell-engrafted (b, d, f, h) NSG mice on d 21 and d 35 after adoptive transfer of Co.TD-Mos or p21TD-Mos or harvested from mice not treated with infusion (NI). Proinflammatory activated murine Raw264.7 macrophages treated with murine IFN γ (mIFN γ) (1 μ g/ml) for 48 h were used as positive controls for mMHCII (p=0.0063) (i) and iNOS (****p<0.0001) (j) staining. In (a-h) and (i, j), the data are presented as the mean \pm SEM from n=3 mice/group and n=3 independent experiments. **p<0.001 and ****p<0.0001; determined with one-tailed (i) and two-tailed (j) unpaired t tests.**

In response to the major point of critique 6 raised by reviewer #1, we also determined the percentage of macrophage transduced with p21 LV and analyzed their survival *in vivo*. Using lentiviral vectors encoding p21 cDNA or control vectors that encode a GFP reporter gene (Co-GFPTD and p21-GFPTD), we first revealed that human Mos were effectively transduced and that GFP expression was stably detected after Mo differentiation into macrophages until 30 days after lentiviral transduction (PPR Fig 9a). In addition, we also demonstrated that 72 days after their adoptive transfer into NSG mice, FACS-sorted Co.TD-MDMs or p21TD-MDMs from BM and spleen exhibited integrated lentiviral copy numbers (4 to 6 lentiviral vector copies per MDM) similar to those of MDMs transduced and differentiated *in vitro* over 7 days (PPR Fig 9b, c), indicating that Co.TD-MDMs or p21TD-MDMs stably persisted for at least 72 days in NSG mice.

Point-by-point reply Figure 9. Lentiviral transduction efficiency and persistence of p21-engineered macrophages in tumor-free mice. a Percentage of live GFP-positive expressing MDMs (GFP^+ MDMs) derived *in vitro* from monocytes transduced with lentiviral vectors encoding p21 and GFP cDNAs (*pp21-GFP*) or a control vector (*pCo-GFP*), using the lentiviral transduction experimental procedure used for *Co.TD-Mos* and *p21TD-Mos* and determined using immunofluorescence microscopy at d 7, 15 and 30 after transduction. The data are presented as the $mean \pm SEM$ from $n=3$ donors. **b, c**, Schematic representation showing the experimental procedure used to analyze the persistence of p21-engineered macrophages (**b**), and quantification of integrated lentiviral copy numbers per MDM (**c**) among MDMs derived from *Co.TD-Mos* or *p21TD-Mos* that were either differentiated *in vitro* for 7 d or sorted ($hCD45^+hCD11b^+hCD71^+hCD14^+$) from the BM and spleen on d 72 after adoptive transfer into tumor-free NSG mice. Integrated lentiviral copy numbers were determined from genomic DNA analyzed by qPCR. In (**a**) and (**b**), the data are presented as the $mean \pm SEM$ from $n=3$ donors and $n=3$ donors (*in vitro*), $n=3$ mice/group (*in vivo*).

All the results obtained were added in the revised Fig. 4c-f and the revised Supplementary Fig. 10, 12, 16 and 17.

Major point of critique 7 raised by reviewer #1: The authors ablate the therapeutic effect of p21-overexpressing MDMs by delivering clodronate after 21 days after MDM transfer. It is known that macrophages need *CSF1R* activation to survive, whereas mouse M-CSF is not active on human *CSF1R*. How long do the human MDMs survive in NSG mice? $CFSE^+$ cells in the spleen may not be human MDMs but mouse spleen macrophages that uptake CFSE from apoptotic/debris MDMs. Authors should label hCD45 to identify human MDMs in mice at 21 days post infusion.

Our response. In response to the major point of critique 6 raised by reviewer #1, we demonstrated that *Co.TD-MDMs* or *p21TD-MDMs* stably persisted for at least 72 days in NSG mice (PPR Fig 9). According to reviewer #1 recommendation, spleen autopsies from NSG mice were analyzed 21 days after the adoptive transfer of $CFSE^+$ *Co.TD-Mos* or $CFSE^+$ *p21TD-Mos* for the expression of human leukocyte ($hCD45$) and macrophage ($hCD68$) markers. We observed that more than 96% of the $CFSE^+$ MDMs detected in the spleen expressed both human leukocyte ($hCD45^+$) and macrophage ($hCD68^+$) markers (PPR Fig. 10a, b).

Point-by-point reply Figure 10. MDMs detected in the spleen of infused mice expressed human leucocyte and macrophage markers. a, b, Confocal micrographs (a) and percentage (b) of CFSE⁺ cells detected in mouse spleens evaluated 21 d after Mo transfer and expressing human leucocyte (hCD45⁺) and human macrophage (hCD68⁺) markers. In (a), the data are representative of n=3 mice/group. In (b), the data are presented as the mean±SEM from n=3 mice/group.

These results were added in the revised Supplementary Fig. 11f, g.

Major point of critique 8 raised by reviewer #1: *Is the therapeutic effect mediated by phagocytosis or by a direct effect of human IFN-gamma on the leukemic cells? The authors show that the therapeutic effect can be abolished when using anti-human IFN-gamma antibodies, as shown in Fig.3p. Can they obtain the same effect by administering IFN-gamma and omitting MDM? The authors could also attempt to block CD47 in vivo using specific antibodies; indeed, even if correlation studies performed in vitro suggest that CD47 does not play a role in inhibiting MDM-mediated phagocytoses, CD47 may still play an important function in vivo.*

Our response. In response to major point of critique 8 raised by reviewer #1, we treated MOLT4 cells with increasing concentrations of human IFN γ (hIFN γ) during 1, 3 and 6 days, analyzed the cell proliferation (by a WST-1 assays) and determined viable cell number using trypan-blue. Thus, we observed that hIFN γ significantly decreased the proliferation (PPR Fig. 11a) and the viability (PPR Fig. 11b) of treated MOLT4 cells in a dose-dependent manner. Moreover, the intraperitoneal administration of hIFN γ into mCherry⁺ MOLT4 cell-engrafted NSG mice elongated the survival of leukemic mice (PPR Fig. 11c). Consistently to the previously demonstrated role of IFN γ to direct anti-proliferative and pro-apoptotic effects against malignant cells (Castro *et al.*, *Front. Immunol*, 2018; Kotredes and Gamero, *J Interferon Cytokine Res*, 2013), our results indicate that IFN γ treatments may also prolong the survival of leukemic mice through the modulation of the leukemic MOLT4 cells viability. Additional investigations will be required to characterize the molecular mechanisms involved in the modulation of MOLT4 cell proliferation and viability. Nevertheless, these results emphasize the fact that, as suggested by reviewer #1, IFN γ secreted by phagocytic macrophages could direct, beside the bystander proinflammatory activation of TAMs, the cell killing of leukemic cells.

Point-by-point reply Figure 11. IFN γ reduces the viability of MOLT4 leukemia cells and prolongs the survival of mCherry⁺ MOLT4 cell-engrafted NSG mice. a, b, Cell proliferation (using a WST-1 assay (** p <0.0001)) (a) and viability (viable cell counts) (**** p <0.0001)) (b) of MOLT4 cells treated (or not) with the indicated hIFN γ concentrations (in International Units/ml (IU/ml)) and during the indicated times. c, Survival of mCherry⁺ MOLT4 cell-engrafted NSG mice treated (or not) with hIFN γ (10 μ g/mouse) on d 7 after leukemia cell engraftment (** p =0.0028). In (a, b), the data represent the mean \pm SEM from n =3 independent experiments. In (c), the survival data are from n =5 mice/group. ** p <0.01, **** p <0.0001; determined with ANOVA with Tukey's test (a, b) and the Mantel-Cox (c) test.**

These results were added in the revised Supplementary Fig. 19.

As shown in our response to the major point of critique 5 raised by reviewer #1 (PPR Fig. 5), our results demonstrated that p21 overexpression enables the phagocytosis of MOLT4 cells through the disruption of anti-phagocytic CD47-SIRP α axis on the side of phagocytic macrophages and highlighted that the adoptive transfer of p21TD-Mos promotes the phagocytosis of CD47-expressing leukemic cells *in vivo* and significantly inhibits leukemia progression through the abrogation of anti-phagocytosis CD47-SIRP α axis, by repressing SIRP α expression on TAMs differentiated from p21TD-Mos.

These results were added in the revised Fig. 2g, 3n and the revised Supplementary Fig. 2, 6 and 15.

Major point of critique 9 raised by reviewer #1: In these survival studies, the control arm is (correctly) performed by infusing mice with untransduced MDM but there is no data showing the survival of mice not infused with any cell. Do untransduced MDM have an effect?

Our response. In response to major point of critique 9 raised by reviewer #1 and to exclude any impact of different steps of the cell manufacturing process in the antitumor effect observed, NSG mice were not treated with adoptive transfer or infused with control untransduced human monocytes (UTD-Mos); human monocytes treated with Vpx viral-like particles (Vpx-Mos), which were used to enhance lentiviral transduction efficiency of myeloid cells (Berger et al., *PloS Pathog* (2011); Laguette et al., *Nature* (2011); p21TD-Mos or Co.TD-Mos prior the engraftment of mCherry⁺ MOLT4 cells. We observed that only mice infused with p21TD-Mos showed significant prolongation of survival (PPR Fig. 10). Altogether, these results demonstrated that the adoptive transfer of p21TD-Mos strongly reduced the progression of leukemia.

Point-by-point reply Figure 12. p21TD-Mo-based therapy prolongs survival in the human T-ALL model. Survivals of *mCherry*⁺ MOLT4 cell-engrafted NSG mice not treated with adoptive transfer or adoptively transferred, as in (submitted Fig. 3a), with untransduced *Mos* (UTD-*Mos*), *Vpx* viral-like particle-treated *Mos* (*Vpx*-*Mos*), *Co.TD*-*Mos* or *p21TD*-*Mos* (****p*<0.0001). The survival data are from *n*=5 mice/group. ****p*<0.0001; determined with Mantel-Cox test.

We added these results in the revised Supplementary Fig. 13b.

Major point of critique 10 raised by reviewer #1: *The authors used MDMs as a model of human macrophages. However, it is well established that macrophages are polarized according to the microenvironment (e.g. organ, tissue, etc) where they are found. Authors' finding may not apply to in vivo (endogenously) differentiated macrophages but it would still be interesting as manipulation/interventional strategy. This should be discussed.*

Our response. We agree with reviewer #1. Considering that tissue-resident macrophages exhibit distinct biology, plasticity and inflammatory activation status that depend on their microenvironment, our findings obtained with *in vitro* differentiated MDMs may not apply to *in vivo* differentiated macrophages. As shown in our response to major point of critique 6 raised by reviewer #1, we revealed that *hCD14*⁺*hCD11b*⁺*hCD163*⁺*CFSE*⁺ MDMs (TAMs) sorted from *mCherry*⁺ MOLT4 cell-engrafted NSG mice that were infused with *CFSE*⁺ human monocytes can be reprogrammed into proinflammatory macrophages after 15 d of coculture with *Phago*⁺ *Co.TD*-MDMs or *Phago*⁺ *p21TD*-MDMs. We noticed that TAMs that were cocultured with *Phago*⁺ *p21TD*-MDMs exhibited significantly increased expression of *iNOS* compared to those that were cocultured with *Phago*⁺ *Co.TD*-MDMs (PPR Fig. 7). Altogether, these results indicate that *p21TD*-Mo-based cellular therapy drives the proinflammatory reprogramming of TAMs.

These results were added in the revised Fig. 4f and revised Supplementary Fig. 16.

Minor point of critique 1 raised by reviewer #1: *Abstract: Refer to CDKN1A as p21 once the term p21 has been introduced.*

Our response. We agree with the minor point of critique 1 raised by reviewer #1 and addressed this critique in the revised manuscript text (page 2, line 5).

Minor point of critique 2 raised by reviewer #1: *The authors exclude efferocytosis as the mechanism of MDM-mediated uptake of leukemic cells, since caspase inhibitor and anti-annexin antibodies did not impair human MDM engulfment of cancer cells (Jurkat and AML blasts). However, CDDP is not described in the results. This figure should be described better in the results section.*

Our response. We agree with the minor point of critique 2 raised by reviewer #1 and we better described CDDP results in the revised manuscript (page 5, lines 22 and 23).

Minor point of critique 3 raised by reviewer #1: *Authors claim that MDMs were treated with intravenous immunoglobulins (IVIG). Here the IVIG term seems ambiguous. Authors should explain better what IVIGs are and how MDMs were treated.*

Our response. We agree with the minor point of critique 3 raised by reviewer #1 and we better explained what IVIGs are and how MDMs were treated in the revised manuscript (page 8, lines 10 and 11).

Minor point of critique 4 raised by reviewer #1: *Extended Data Fig. 1e and d are quoted before Extended Data Fig. 1b, c. Authors should quote figures in a chronological order.*

Our response. We agree. The minor point of critique 4 raised by reviewer #1 was addressed in the revised manuscript (page 5, lines 19-24) and in the revised Supplementary Fig. 1b-e.

Minor point of critique 5 raised by reviewer #1: *It is unclear the rationale used by the authors to investigate the role of p21. There are many transcription factors, including the IRF5, that could be in part responsible for the observed phenotype. If possible, the authors should briefly indicate their rationale in the result section before describing the strategy to knock down p21.*

Our response. We agree with the minor point of critique 5 raised by reviewer #1 and we indicated our rationale in the revised manuscript (page 7, lines 18-22), as followed “Considering (i) the shared cellular features between macrophages and senescent cells (Behmoaras et al., *J Cell Biol* (2021)), (ii) the ability of senescent cells to phagocytose live tumor cells (Tonnessen-Murray et al., *J Cell Biol* (2019)) and (iii) the pivotal roles of p21 during the senescence (Abbas and Dutta, *Nat Rev Cancer* (2009)), terminal differentiation and survival of macrophages (Asada et al., *EMBO J* (1999); Kramer et al., *Br J Haematol* (2002); Comalda et al., *Eur J Immunol* (2004); Gazova et al., *Front Cell Dev Biol* (2020)), we assessed the role of p21 in order to identify molecular mechanisms regulating the macrophage-mediated phagocytosis of leukemia cells.”

Reviewer 2 (expert on T-ALL)

General critique raised by reviewer 2: In this work, the authors identified p21 as a trigger of phagocytosis-guided pro-inflammatory reprogramming of TAMs and demonstrate the potential for p21TD-Mo-based cell therapy in cancer immunotherapy. I have a few questions that need to be clarified before this paper can be considered for publication.

Our response. We thank reviewer #2 for her/his positive constructive critiques that we addressed in the revised manuscript.

Major point of critique 1 raised by reviewer #2: Figure 1c. MDMs fail to engulf HEL and K562 cells. Why is that?

Our response. We understood the major point of critique 1 raised by reviewer #2. Despite the fact that we demonstrated that CD47 is a key determinant for the repression of leukemia T cell phagocytosis (PPR Fig. 5) and that CD47 cell-surface expression in HEL cells or K562 cells was lower than those in Jurkat cells or MOLT4 cells, we did not detect phagocytosis of HEL cells and K562 cells by MDMs (submitted Fig. 1c and Extended Fig. 1h, i). These results suggest that proteins other than CD47 that are expressed or released by HEL cells and K562 cells may also regulate macrophage phagocytosis. In this context, we studied the cell-surface expression of SLAMF7, which is a prophagocytosis signal that can be expressed on both tumor cells and macrophages (Chen *et al.*, *Nature* (2017)), on MOLT4 cells, HEL cells and K562 cells. Using flow cytometry analysis, we detected that SLAMF7 cell-surface expression in HEL cells or K562 cells was significantly lower than those of MOLT4 cells (PPR Fig. 11), thus suggesting that the cell-surface expression of SLAMF7 on tumor cells may play a key role as “eat-me ligand” during CD47-dependent phagocytosis by macrophages, as previously published (Chen *et al.*, *Nature* (2017)). Further investigations are needed to decipher the role of SLAMF7 cell-surface expression on both tumor cells and macrophages during p21-mediated tumor phagocytosis and phagocytosis-guided proinflammatory macrophage reprogramming. Hence we did not add these preliminary results in the revised manuscript.

Point-by-point reply Figure 11. Cell-surface expression of SLAMF7 in MOLT4 cells, HEL cells and K562 cells. The mean fluorescence intensities (MFI) of SLAMF7 cell-surface expression were determined by FACS (***p*<0.001 and *****p*<0.0001). The data are presented as the mean±SEM from *n*=3 independent experiments. ****p*<0.001 and *****p*<0.0001; determined with ANOVA with Tukey’s multiple comparison test.

Major point of critique 2 raised by reviewer #2: Extended data Fig 1. It is remarkable that leukemic T cells could be refractory to phagocytosis induced by anti-CD47 blockade. What could be the reason for this?

Our response. We agree with the major point of critique 2 raised by reviewer #2 that was also raised by reviewer #1 and reviewer #3. As indicated in our response to reviewer #1 (PPR Fig. 5), we better characterized the relationship between CD47 expression on the surface of leukemia cells and p21-dependent tumor phagocytosis. Thus, we demonstrated using different *in vitro* and *in vivo* experimental procedures with blocking antibodies (PPR Fig. 5c, 5l and 5m) and CRISPR/Cas9 gene-editing (PPR Fig. 5d-k and 5n) specific for CD47 that (i) p21 overexpression enables the phagocytosis of MOLT4 cells through the disruption of antiphagocytic CD47-SIRP α axis in phagocytic macrophages and that (ii) the adoptive transfer of p21TD-Mos promotes the phagocytosis of CD47-expressing leukemic cells *in vivo* and significantly inhibits leukemia progression through the abrogation of antiphagocytosis CD47-SIRP α axis mediated by repressing SIRP α expression in TAMs differentiated from p21TD-Mos.

These results were added in the revised Fig. 2g, 3n and the revised Supplementary Fig. 2, 6 and 15.

Major point of critique 3 raised by reviewer 2: Next, the authors compare Phago⁺MDM with Phago⁻MDM cells. Although a variety of differences are identified. The authors decided to focus on IFN γ . Why did the authors pick this specific cytokine?

Our response. We thank reviewer #2 for her/his positive constructive critique. Considering that IFN γ can be released by proinflammatory macrophages (Wu et al., *Cell Death Differ* (2018)) and is a key modulator of antitumor immune response (Hu and Ivashkiv, *Immunity* (2009)), we paid a particular attention to this cytokine in our study. Furthermore, using cytokine array, western blot and ELISA (submitted Fig. 1m-o, submitted Extended Fig.2 and PPR Fig 1), we demonstrated that IFN γ is secreted by phagocytic macrophages (Phago⁺MDMs), after the degradation of engulfed leukemic cells and that secreted IFN γ dictates the proinflammatory reprogramming of neighboring nonphagocytic anti-inflammatory macrophages. In addition and consistently to the previously demonstrated antiproliferative and proapoptotic effects of IFN γ on malignant cells (Castro et al., *Front. Immunol*, 2018; Kotredes and Gamero, *J Interferon Cytokine Res*, 2013), our results indicate that IFN γ treatments may prolong the survival of leukemic mice through the modulation of the leukemia MOLT4 cell viability (PPR Fig. 11). Additional investigations are needed to characterize the molecular mechanisms involved in the modulation of MOLT4 cell proliferation and viability. Altogether, these results emphasize the fact that the IFN γ secreted by phagocytic macrophages could direct (i) the killing of leukemic cells and (ii) the proinflammatory reprogramming of both phagocytic and nonphagocytic anti-inflammatory macrophages, *in vitro* and *in vivo*.

These results were added in the revised Fig. 1n and the revised Supplementary Fig. 19.

Major point of critique 4 raised by reviewer 2: Next, the authors focus on p21 to identify molecular mechanisms that regulate the macrophage-mediated phagocytosis of leukemic cells. Why did they pick p21?

Our response. In response to the major point of critique 4 raised by reviewer #2 that was also raised by reviewers #1 and #3, we indicated our rationale in the revised manuscript (page 7, lines 18-22), as followed “Considering (i) the shared cellular features between macrophages and senescent cells (Behmoaras et al., *J Cell Biol* (2021)), (ii) the ability of senescent cells to phagocytose live tumor cells (Tonnessen-Murray et al., *J Cell Biol* (2019)) and (iii) the pivotal roles of p21 during the senescence (Abbas and Dutta, *Nat Rev Cancer* (2009)) and during the terminal differentiation and the survival of macrophages (Asada et al., *EMBO J* (1999); Kramer et al., *Br J Haematol* (2002); Comalda et al., *Eur J Immunol* (2004); Gazova et al., *Front Cell Dev Biol* (2020)), we assessed the role of p21 in order to identify molecular mechanisms regulating the macrophage-mediated phagocytosis of leukemia cells”.

Major point of critique 5 raised by reviewer 2: *How would the authors envision clinical translation of their findings in daily clinical practice?*

Our response. We thank reviewer #2 for her/his positive and constructive question regarding the clinical translation of our findings. We agree with reviewer #2 that our findings highlighted a new concept using patient’s engineered autologous monocytes to infiltrate tumors, engulf cancer cells and harness antitumor immunity for cancer cure that should be translated to clinic. To translate our findings to clinical stage, we will partner with contract manufacturing organizations (CMOs) and contract development and manufacturing organization (CDMOs) to engineer patient’s monocytes, according to good manufacturing practices (GMP) guidelines and regulatory affairs. In daily clinical practice, monocytes will be collected from whole blood of cancer patients through leukapheresis. With the support of CMOs/CDMOs, patient’s monocytes will be genetically engineered *ex vivo* by cotransducing p21-expressing self-inactivated lentiviral vectors with viral-like particles containing Vpx protein during 4 hours and assessed for p21 expression and SIRP α repression. As for autologous CAR T cellular therapy, cancer patients will be then infused with their own engineered monocytes and followed for leukemia regression (disease free survival and remission rate), side effects such neurotoxicity and cytokine release syndrome (CRS) and persistence of engineered monocytes in blood and engineered macrophages in tumor beds. Considering that the manufacturing time for CAR T cells (such as anti-CD19 CAR T cells CTL019 (Kymriah (Novartis)) and KTE-C19 (Yescarta) (Kite/Gilead)) is ranging for 17 to 22 days, we envision that the efficient manufacturing of p21-engineered monocytes within few days (considering all validation and release steps) will improve gene therapy manufacturing processes and production.

In addition, considering the ability of human monocytes to naturally patrol through human body and to infiltrate both hematological malignancies and solid tumors (Laguette et al., *Nature* (2011); Berger et al., *Gene Ther* (2011); Arwert et al., *Cell Rep* (2018)), and our recent experimental evidence demonstrating that p21-transduced MDMs can overcome SIRP α -mediated phagocytosis restriction and immune escape, and that phagocytic engineered MDMs can specifically eliminate cancer cells and alleviate immunosuppressive tumor microenvironment (TME), we are convinced that phagocytosis-guided, monocyte-based cell therapy should also overcome several limitations restricting the efficacy of CAR T cell therapies and should improve the treatment of CD47⁺ hematological malignancies (other than T-ALL) and CD47⁺ solid tumors (such as glioblastoma and lung adenocarcinoma).

Reviewer #3 (expert on macrophage, immunosurveillance and cancer)

General critiques raised by reviewer #3: *In this manuscript, the authors discovered a novel role of CDKN1A in regulating macrophage phagocytosis of leukemic cells. They showed that CDKN1A (p21) regulated the expression of Sirpα, a receptor for “don’t eat me” signal CD47. This is an exciting discovery and the human monocytes engineered to overexpress p21 demonstrated impressive anticancer efficacy in in vivo MOLT4 cell model and T-ALL PDX models. While their findings are novel and exciting, the mechanism of p21 in regulating phagocytosis was not well explored in this study, raising certain big concerns. The authors revealed a correlation between p21 and Sirpα expression-knockdown of p21 promoted Sirpα expression whereas overexpression of p21 inhibited Sirpα expression in MDMs. These data would suggest that p21 regulate phagocytosis through Sirpα signaling. However, the data in Fig1 showed that phagocytosis of the leukemic T cells was independent of their CD47 expression and was resistant to CD47 blockade, excluding a possible role of CD47-Sirpα signaling axis in this process. So the interpretation would be that p21 regulates the expression of Sirpα, which controls phagocytosis ability of MDM through a novel CD47-independent mechanism - however, this has not been addressed at all in this manuscript.*

Our response. We thank reviewer #3 for her/his positive, enthusiastic and constructive general critiques concerning the manuscript that we submitted. We agree with reviewer #3 concerning the role of CD47-SIRPα axis during p21-directed, macrophage-mediated phagocytosis of leukemic cells. As mentioned in our response to the major critique 5 of reviewer #1 and the major critique 2 of reviewer #2, we addressed this critique and better characterized the relationship between CD47 expression on the surface of leukemia cells and p21-dependent tumor phagocytosis. Thus, we demonstrated using different *in vitro* and *in vivo* experimental procedures with blocking antibodies (PPR Fig. 5c, 5l and 5m) and CRISPR/Cas9 gene-editing (PPR Fig. 5d-k and 5n) specific for CD47 that (i) p21 overexpression enables the phagocytosis of MOLT4 cells through the disruption of antiphagocytic CD47-SIRPα axis in phagocytic macrophages and that (ii) the adoptive transfer of p21TD-Mos promotes the phagocytosis of CD47-expressing leukemic cells *in vivo* and significantly inhibits leukemia progression through the abrogation of antiphagocytosis CD47-SIRPα axis mediated by repressing SIRPα expression in TAMs differentiated from p21TD-Mos.

These results were added in the revised Fig. 2g, 3n and the revised Supplementary Fig. 2, 6 and 15.

Major point of critique 1 raised by reviewer #3: *It’s very interesting that the authors showed IFNγ secreted by Phago⁺ MDMs stimulated Phago⁻ MDMs to a pro-inflammatory phenotype. The authors should investigate would such changes of phenotype on Phago⁻ MDMs also enhance their phagocytic ability.*

Our response. We would like to thank the reviewer #3 for raising this major point of critique 1. To address this major point of critique, CMFDA-labeled MDMs were cocultured with CMTMR-labeled MOLT4 leukemic cells during 2 hours and sorted by FACS on the basis of their phagocytic activity. Sorted CMTMR⁺CMFDA⁺ (Phago⁺) MDMs or CMTMR⁻CMFDA⁺ (Phago⁻) MDMs (seeded in the upper chambers of Transwell devices) were cocultured with CMTMR⁻CMFDA⁺ (Phago⁻) MDMs (seeded in the bottom chambers) (PPR Fig. 12a). After 15 days, the upper chambers were removed and CMTMR⁺ MOLT4 cells were added to CMTMR⁻CMFDA⁺ (Phago⁻) MDMs in the bottom chamber for coculture during 8 hours. We

then assessed the phagocytic capacity of these MDMs and observed that MDMs that were cocultured with CMTMR⁺CMFDA⁺ (Phago⁺) MDMs exhibited a significant enhancement of their phagocytic activity (PPR Fig. 12b), as compared to MDMs that were cocultured with CMTMR⁻CMFDA⁺ (Phago⁻) MDMs. These results demonstrate that phagocytic macrophages could enhance the phagocytic capacities of neighboring nonphagocytic macrophages. Additional investigations will be required to characterize the molecular mechanisms regulating phagocytic capacities of neighboring macrophages and biological consequences of this process.

While very interesting and unexpected, these results were not added to the revised manuscript.

Point-by-point reply Figure 12. Phagocytic macrophages enhanced the phagocytic abilities of bystander macrophages. **a**, Schematic representation showing the experimental procedure used to analyze phagocytic capacities of bystander macrophages. FACS-sorted CMTMR⁺CMFDA⁺ (Phago⁺) MDMs or CMTMR⁻CMFDA⁺ (Phago⁻) MDMs (upper chambers) were cocultured in Transwell devices during 15 days with CMTMR⁻CMFDA⁺ (Phago⁻) MDMs (bottom chambers). Then, CMTMR⁻CMFDA⁺ (Phago⁻) MDMs were cocultured with CMTMR⁺MOLT4 cells during 8 hours and analyzed for phagocytosis. **b**, Phagocytosis percentage of MOLT4 cells by MDMs are shown (* $p=0.0258$). In (**b**), the data are donor matched from $n=3$ donors. * $p<0.05$ is determined by one-tailed paired t test.

Major point of critique 2 raised by reviewer #3: Experiments performed in Fig1 were not well connected to the rest of the paper. In addition, the author started to investigate the role of p21 in macrophage phagocytosis but there was a lack of interpretation of why was p21 studied for this purpose and what the connection is between p21 and the experiments performed in Fig1 for the inflammatory phenotype of Phago⁺ or Phago⁻ MDMs.

Our response. We agree with reviewer #3. In response to her/his major point of critique 2, we corrected the submitted manuscript. In the revised manuscript, we better introduced our work and better presented the main results in the revised manuscript (page 3, line 17, page 3 to page 4, line 25). We also divided the Results section into distinct paragraphs with subheadings (page 5 to page 16) and better connected these paragraphs together (page 5, line 3 and page 7, line 18).

As indicated in our response to the minor point of critique 5 raised by reviewer #1 and the major point of critique 4 raised by reviewer #2, we better indicated our rationale to investigate the role of p21 in macrophage phagocytosis in the revised manuscript (page 7, lines 18-22), as followed “Considering (i) the shared cellular features between macrophages and senescent cells (Behmoaras et al., *J Cell Biol* (2021)), (ii) the ability of senescent cells to phagocytose

live tumor cells (Tonnessen-Murray *et al.*, *J Cell Biol* (2019)) and (iii) the pivotal roles of p21 during the senescence (Abbas and Dutta, *Nat Rev Cancer* (2009)) and during the terminal differentiation and the survival of macrophages (Asada *et al.*, *EMBO J* (1999); Kramer *et al.*, *Br J Haematol* (2002); Comalda *et al.*, *Eur J Immunol* (2004); Gazova *et al.*, *Front Cell Dev Biol* (2020)), we assessed the role of p21 in order to identify molecular mechanisms regulating the macrophage-mediated phagocytosis of leukemia cells". We expected that the rewriting of manuscript and the corrections added to the revised manuscript will better introduce and explain the rationale of our study to characterize the molecular basis and the antitumor consequences of a novel non-cell autonomous modality of macrophage proinflammatory activation that starts with the phagocytosis of live cancer cells.

Major point of critique 3 raised by reviewer #3: In fig2, western blot was used for assessing the expression of p21 and Sirp α in MDMs. It would be more informative if FACS can be performed to examine cell surface Sirpa (to evaluate the percentage of cells whose surface expression of Sirpa were impacted) of these MDMs. The efficiency of MDM transduction by lentiviruses should be examined as well – what is the percentage of MDMs that were transduced by lentiviruses?

Our response: We agree with the major point of critique 3 raised by reviewer #3. In response, we analyzed using flow cytometry, the cell-surface expression of SIRP α in MDMs that were depleted for p21 (sip21), that overexpressed p21 (p21TD) and on their corresponding controls (siCo. or Co.TD). Though the percentage of MDMs expressing SIRP α was not significantly affected by the depletion or the overexpression of p21 (PPR Fig. 13a, c, respectively), the cell-surface expression of SIRP α was significantly increased after 7 days of p21 knockdown (PPR Fig. 13b) and significantly decreased after 7 and 15 days of p21 overexpression (PPR Fig. 13d), as compared to their corresponding controls. Considering that (i) MDMs are terminally differentiated cells that did not divide (PPR Fig. 4) and (ii) p21 does not affect their cell cycle progression or viability (PPR Fig. 4), the increased or decreased SIRP α cell-surface expression by, respectively, p21 knockdown or by p21 overexpression persisted for long time after siRNA transfection or transduction of p21-encoding lentiviral vectors. Altogether, these results are in agreement with the repression of SIRP α transcription by p21 that we reported in the submitted manuscript (submitted Fig. 2).

Point-by-point reply Figure 13. p21 expression modulates SIRP α cell-surface expression. **a, b,** Percentage of cells expressing SIRP α (**a**) and SIRP α cell-surface expression (***p*=0.0023) (**b**) determined by FACS of siCo. or sip21 MDMs at 7 d after siRNA transfection. **c, d,** Percentage of cells expressing SIRP α (**c**) and SIRP α cell-surface expression (**p*=0.0333, ****p*=0.0005) (**d**) determined by FACS of Co.TD-MDMs or p21TD-MDMs at 7 d and 15 d after lentiviral transduction. Mean fluorescence intensity (MFI) of SIRP α are shown in (**b**) and (**d**). In (**a, b**) and (**c, d**) the data are donor matched from *n*=4 and

n=3 donors. **p*<0.05, ***p*<0.01 and ****p*<0.001 are determined with two-tailed paired *t* test (b) and with ANOVA with Sidak's multiple comparison test (d).

The results shown in PPR Fig. 13b, d were added in the revised Supplementary Fig. 7a, c.

As shown in our response the major point of critique 6 raised by reviewer #1, we also determined the percentage of macrophage transduced with p21 LV and analyzed their survival *in vivo*. Using lentiviral vectors encoding p21 cDNA or control vectors that encode a GFP reporter gene (Co-GFPTD and p21-GFPTD), we first revealed that human Mos were effectively transduced and that GFP expression was stably detected after Mo differentiation into macrophages until 30 days after lentiviral transduction (PPR Fig 9a). In addition, we also demonstrated that 72 days after their adoptive transfer into NSG mice, FACS-sorted Co.TD-MDMs or p21TD-MDMs from BM and spleen exhibited integrated lentiviral copy numbers (4 to 6 lentiviral vector copies per MDM) similar to those of MDMs transduced and differentiated *in vitro* over 7 days (PPR Fig 9b, c), indicating that Co.TD-MDMs or p21TD-MDMs stably persisted for at least 72 days in NSG mice. Altogether, these results revealed that MDMs were efficiently and stably transduced with lentiviruses.

These results were added in the revised Supplementary Fig. 10 and 12.

Major point of critique 4 raised by reviewer #3: *The authors' finding regarding leukemic T cells are resistant to CD47 blockade-induced phagocytosis is contradictory to many of previous studies (There are many studies showing Jurkat cells were efficiently phagocytosed upon CD47 blockade, eg. Weiskopf et al, Science, 2013; Peluso et al, JITC, 2020). In addition, it's very surprising that as shown in Fig1c etc. the phagocytosis rate of Jurkat cells (and MOLT4, CEM, THP1) by resting MDMs (without treatment of antibodies etc.) could reach 20-50%, which seems to be inconsistent with previous studies in which phagocytosis rate by resting MDMs were usually 5-10% or lower. An interpretation of such inconsistency is needed.*

Our response. We agree with the major point of critique 4 raised by reviewer #3. As shown in our response to the major point of critique 5 raised by reviewer #1, we proposed in the submitted manuscript that macrophage-mediated phagocytosis of leukemia cells could be independent on CD47 expression (submitted Extended Data Fig. 1h-j) and thus, could be in conflict with previous publications (Weiskopf *et al.*, *Science* (2013); Peluso *et al.*, *J Immunother Cancer*, (2020)). To better characterize the relationship between CD47 expression on leukemia cells and p21-dependent tumor phagocytosis, we compared our experimental procedure to those previously published (Majeti *et al.*, *Cell*, 2009; Chao *et al.*, *Cancer Res*, 2011) and determined the effect of CD47 depletion in leukemia cells on the macrophage-mediated phagocytosis. We agree that the phagocytosis rates of leukemia cells by macrophages (15-45%) that we observed in control cocultures (submitted Fig. 1c and Extended Data Fig. 1j) were higher than those previously reported (2-10%) (Majeti *et al.*, *Cell*, 2009; Chao *et al.*, *Cancer Res*, 2011; Weiskopf *et al.*, *Science*, 2013; Peluso *et al.*, *J Immunother Cancer*, 2020). As indicated in the submitted Methods section, our phagocytosis assays were performed in 10% Heat Inactivated (HI) serum-supplemented medium and analyzed after 8 hours, while previous published assays were performed with MDMs that were cultured for 2 hours in serum-free medium prior the phagocytosis (performed in the same culture medium) and analyzed after 2 hours (Majeti *et al.*, *Cell*, 2009; Chao *et al.*, *Cancer Res*, 2011). To determine the effects of these differences on phagocytosis rates, we first explored the expressions of p21 and SIRP α in MDMs that were cultured in absence or in

presence of 10% HI serum. We observed after two-hours of serum deprivation that p21 protein expression was decreased (PPR Fig. 5a), and both protein (PPR Fig. 5a) and cell-surface (PPR Fig. 5b) expressions of SIRP α strongly increased. These expressions positively correlated with low phagocytosis rates of MOLT4 cells detected in serum-free phagocytosis assays with respect to phagocytosis assays performed in presence of serum (PPR Fig. 5c). Furthermore, in serum-free phagocytosis assays, cocultures in presence of anti-CD47 antibodies showed a significant enhancement of macrophage-mediated phagocytosis of MOLT4 cells, while phagocytosis assays performed in presence of serum did not increase macrophage-mediated phagocytic activity, as compared to control cocultures performed in presence of isotype control (IgG) (PPR Fig. 5c). These results indicate that the apparent discrepancy between our results (submitted Extended Data Fig. 1j) and previous publications (Majeti *et al.*, *Cell*, 2009; Chao *et al.*, *Cancer Res*, 2011) remains on the experimental procedure used. Furthermore, phagocytosis assays with p21-overexpressing MDMs using either anti-CD47 antibodies or CD47-depleted MOLT4 cells (CrCD47MOLT4 cells) showed that p21 overexpression enables the phagocytosis of MOLT4 cells through the disruption of antiphagocytic CD47-SIRP α axis on the side of phagocytic macrophages (PPR Fig. 5d-l). In addition, these results that were obtained *in vitro* (PPR Fig. 5c-l) were confirmed *in vivo* using anti-CD47 blocking antibodies (PPR Fig. 5m) or CrCD47MOLT4 cells (PPR Fig. 5n). Altogether, these results demonstrated that macrophage p21-mediated phagocytosis of leukemia cells is dependent on the expression of CD47 in leukemia target cells.

These results were added in the revised Fig. 2g, 3n and revised Supplementary Fig. 2, 6 and 15.

Major point of critique 5 raised by reviewer 3: *It's difficult to interpret the data in Fig2 that p21 regulated phagocytosis through the regulation of Sirp α . If the changes of Sirp α have such significant effects on phagocytosis as shown in Figure 2c-e and Fig2r, why didn't blockade of CD47 have any effects (Extended Fig 1J.), given that CD47 functions through binding to Sirp α to send phagocytosis-inhibitory signals? Were the findings in figure2 due to certain mechanisms independent of CD47- Sirp α axis? If so, such mechanisms should be investigated.*

Our response. We agree with the major point of critique 5 raised by reviewer #3. As shown in our response to the major point of critique 5 raised by reviewer #1, we further emphasized that the phagocytosis of leukemia cells by macrophages depends on CD47 expression. MOLT4 cells were thus transduced with lentiviral vectors encoding for control or specific CD47 CRISPR guide RNAs (gRNAs) and CAS9 gene. Then, MDMs were cocultured with stably CD47-depleted (CrCD47MOLT4) and control (CrCo.MOLT4) MOLT4 cells (PPR Fig. 5d, e) and analyzed for tumor phagocytosis. Giving that the depletion of CD47 in MOLT4 cells did not affect their proliferation (PPR Fig. 5f, g) and that the percentage and the efficiency of the phagocytosis of CrCD47MOLT4 by MDMs were significantly enhanced with respect to CrCo.MOLT4 cells (PPR Fig. 5h-j), these results demonstrate that the phagocytosis of leukemic T cells by MDMs is inhibited by the expression of CD47 in target cells. To further investigate the molecular mechanisms regulating p21-dependent phagocytosis of leukemia cells, p21-upregulating (p21TD) or control (Co.TD) MDMs were cocultured with control (CrCo.) or CD47-depleted (CrCD47) MOLT4 cells and analyzed for phagocytosis. The cocultures of p21TD-MDMs with CrCo.MOLT4 cells exhibited a significant increase of the percentage of phagocytic MDMs, while the cocultures of Co.TD-MDMs or p21TD-MDMs with CrCD47 MOLT4 cells did not show significant enhancement in the phagocytosis of the leukemia cells (PPR Fig. 5k). Accordingly, coculturing p21TD-

MDMs with MOLT4 cells in the presence of anti-CD47 blocking antibodies did not increase the phagocytosis of MOLT4 cells compared to coculturing p21TD-MDMs with MOLT4 cells in the presence of isotype control (IgG) or coculturing Co.TD-MDMs with MOLT4 cells in the presence of anti-CD47 blocking antibodies (PPR Fig. 5l). To appreciate the anti-tumor effect of the combination of p21TD-Mo-based cellular therapy with CD47 blockade (Majeti *et al.*, *Cell* (2009)), mCherry⁺ MOLT4 cell-engrafted NSG mice were treated 15 days after the adoptive transfer of p21TD-Mos, SIRP α TD-Mos, p21+SIRP α TD-Mos or Co.TD-Mos with anti-CD47 antibodies. Anti-CD47 antibodies significantly prolonged the survival of treated mice, with the exception of the mice infused with p21TD-Mos, which did not show an additional enhancement of survival compared with that of the corresponding control mice (PPR Fig. 5m). These results were confirmed by adoptively transferring p21TD-Mos or Co.TD-Mos into NSG mice engrafted with CD47-depleted (CrCD47MOLT4) or control (CrCo.MOLT4) MOLT4 cells (PPR Fig. 5n). Altogether, these results revealed that p21 overexpression enables the phagocytosis of MOLT4 cells through the disruption of antiphagocytic CD47-SIRP α axis on the side of phagocytic macrophages and highlighted that the adoptive transfer of p21TD-Mos promotes the phagocytosis of CD47-expressing leukemia cells *in vivo* and significantly inhibits leukemia progression through the abrogation of antiphagocytic CD47-SIRP α axis by repressing SIRP α expression in TAMs differentiated from p21TD-Mos.

These results were added in the revised Fig. 2g, 3n and the revised Supplementary Fig. 2, 6 and 15.

Major point of critique 6 raised by reviewer #3: *Is p21 differentially expressed in Phago⁺ vs Phago⁻ MDMs? Is the expression of p21 in Phago⁺ and Phago⁻ MDMs correlated to their phagocytosis capacity?*

Our response. In response to the major point of critique 6 raised by reviewer #3, CMFDA⁺ MDMs were cocultured with CMTMR⁺ MOLT4 cells for two hours and then, CMTMR⁺CMFDA⁺ (Phago⁺) MDMs and CMTMR⁻CMFDA⁺ (Phago⁻) MDMs were analyzed for p21 expression. No significant difference between phagocytic (Phago⁺) or nonphagocytic (Phago⁻) MDMs was detected for the expression of p21 (PPR Fig. 14a) and for the percentage of MDMs showing p21 expression (PPR Fig. 14b). These results suggest that p21 seems to not be differentially expressed in nonphagocytic (Phago⁻) macrophages and in (Phago⁺) macrophages that spontaneously engulfed MOLT4 cells, thus revealing in this experimental setting, that the modulation of the basal expression level of p21 did not positively correlate with the spontaneous phagocytosis capacity of macrophages. Regarding the ability of p21 overexpression or p21 depletion to respectively induce or inhibit the phagocytosis of leukemia cells by macrophages, more in-depth investigations at single cell level are needed to further characterize the relationship between the endogenous modulation of macrophage p21 expression level and the spontaneous phagocytosis of leukemia cells by macrophages. Thus, we did not include these results in the revised manuscript.

Point-by-point reply Figure 14. Expression of p21 in phagocytic (Phago⁺) and nonphagocytic (Phago⁻) macrophages. a, b, Confocal micrographs (a) and percentage of phagocytic (Phago⁺) and nonphagocytic (Phago⁻) MDMs expressing p21 (b) after 2h of cocultures of CMFDA-stained MDMs with CMTMR-stained MOLT4 (b). In (a), the data is representative of n=3 donors. In (b) the data are presented as the mean±SEM from n=3 donors.

Major point of critique 7 raised by reviewer 3: Does p21 depletion or overexpression have an impact on the viability of MDMs? Could the reduction of phagocytic ability of p21 knockdown MDMs be due to their compromised viability?

Our response. In response to the major point of critique 4 raised by reviewer #1, we analyzed the cell cycle progression, the proliferation and the viability of p21-depleted MDMs using siRNA (as shown in submitted Fig. 2a), p21-overexpressing MDMs using lentiviral vectors (as shown in submitted Fig. 2p) and control MDMs (siCo. or Co.TD) at indicated times after siRNA transfection (PPR Fig. 4a-c, g, i) or lentiviral transduction (PPR Fig. 4d-f, h, j). We observed that control (siCo. or Co.TD), p21-depleted (sip21) or p21-overexpressing (p21TD) MDMs were mainly arrested in G0/G1 phase (81-90%) (PPR Fig. 4a-f), did not divide (PPR Fig. 4g-j) and were not affected in their viability (PPR Fig. 4g, h) until 30 days after transfection or transduction, thus demonstrating that the modulation of p21 expression did not alter the terminal differentiation and the survival of macrophages. In addition, these results also reveal that the phagocytic capacity of MDMs is independent of MDM proliferation and that the reduction of phagocytosis ability of p21-depleted MDMs is not related to the alteration of their viability.

These results were added in the revised Supplementary Fig. 4 and 8.

Major point of critique 8 raised by reviewer 3: Given that p21 is a multi-functional regulators of macrophage functions, the authors should perform a function rescue experiment to exclude the off-target effects – eg. To express p21 in p21 knockdown MDMs and examine whether phagocytosis can be reversed to a similar level as that of the WT cells.

Our response. We thank reviewer #3 for her/his positive constructive critique. In response to her/his major point of critique 8, we transduced MDMs with lentiviral vectors encoding short hairpin RNA (shRNA) targeting the 3' untranslated region of p21 (sh3'UTRp21) and/or p21 cDNA (p21TD) resistant to sh3'UTRp21 (PPR Fig 15a). Accordingly, the silencing or the upregulation of p21 in MDMs, respectively, impaired or enhanced the phagocytosis of MOLT4 cells, as indicated by comparison to control MDMs (Co.TD) (PPR Fig 15b). Importantly, the exogenous expression of p21 cDNA in p21-depleted MDMs (p21TD + sh3'UTRp21) restored the phagocytic activity of these complemented MDMs as compared to

that of control or p21-depleted cells (Co.TD or sh3'UTRp21) (PPR Fig 15b), further corroborating the specific key role of p21 in promoting the phagocytosis of leukemia cells by macrophages. Furthermore, these results confirm the results obtained after p21 silencing (shown in submitted Fig. 2), thus excluding the off-target effects.

Point-by-point reply Figure 15. p21 exogenous expression restores phagocytosis in p21-depleted MDMs. **a**, Expression of p21 in MDMs transduced with control (Co.TD) lentiviral vectors, p21-encoding (p21TD) lentiviral vectors and/or lentiviral vectors encoding p21 short hairpin RNA targeting the 3' untranslated region of the p21 gene-encoding (sh3'UTRp21) at 72 h after transduction. **b**, Percentage of phagocytosis of MOLT4 cells by the indicated transduced MDMs shown in (a) after 8 h of coculture (*p=0.0462, ***p=0.0003, *p=0.0172). In (a), the data are representative from n=3 donors. The data in (b) are presented as the mean±SEM from n=3 donors. *p<0.05, **p<0.01, ***p<0.001; determined with ANOVA with Tukey's multiple comparison test (b).

These results were added in the revised Supplementary Fig. 5k,l.

Major point of critique 9 raised by reviewer #3: In the experiment depicted in figure 3, a control group is missing – mice only transplanted with MOLT4 cells but not the MDMs. As the authors showed in figure 1, WT MDMs demonstrated significant basal level phagocytosis of MOLT4 cells, therefore, it would be expected that the mice transplanted with Co. TD MDMs should demonstrate certain level of inhibition of MOLT4 cells, as compared to the mice only transplanted with MOLT4 but not MDMs.

Our response. We agree with reviewer #3. As shown in our response to the major point of critique 9 raised by reviewer #1 and to exclude any impact of different steps of the cell manufacturing process in the antitumor effect observed, NSG mice were not treated with adoptive transfer or infused with control untransduced human monocytes (UTD-Mos); human monocytes treated with Vpx viral-like particles (Vpx-Mos), which were used to enhance lentiviral transduction efficiency of myeloid cells (Berger et al., *PloS Pathog* (2011); Laguette et al., *Nature* (2011)); p21TD-Mos or Co.TD-Mos prior the engraftment of mCherry⁺ MOLT4 cells. We observed that only mice infused with p21TD-Mos showed significant prolongation of survival (PPR Fig. 12). Altogether, these results demonstrated the adoptive transfer of genetically controlled, p21-overexpressing human Mos is required to efficiently subvert immunosuppressive TAMs and reduce the tumor growth.

We added these results in the revised Supplementary Fig. 13b.

Minor point of critique 1 raised by reviewer #3: Fig 1g, was the x-axis “phagocytic MDMs - +” mis-labeled?

Our response. We corrected this error in the revised Fig. 1g.

Minor point of critique 2 raised by reviewer #3: Fig 1j-m, please indicate how long after the initiation of phagocytosis was the comparison for CD163, IRF5, etc. performed?

Our response. According to the minor point of critique 2 raised by reviewer #3, we added in the revised Fig.1j-m and the corresponding figure legends how long after the initiation of phagocytosis was the comparison for transcriptomic analysis, for CD163 and IRF5 expression, and for the secretion of IL1 β , IL6, IL8 and IFN γ .

We addressed this comment in the revised Fig. 1j-m.

REVIEWER COMMENTS

Reviewer #1 (Remarks to the Author):

The Authors have commendably and satisfactorily addressed nearly all issues raised by the reviewers with substantial new experimental work, which have enriched the manuscript and helped clarifying and amending the original interpretation of the findings.

Reviewer #2 (Remarks to the Author):

Authors have addressed all my suggestions and concerns.

Reviewer #3 (Remarks to the Author):

My comments have been adequately addressed.

Point-by-point response to Reviewer's comments:

Reviewer #1

General critique raised by reviewer #1: The Authors have commendably and satisfactorily addressed nearly all issues raised by the reviewers with substantial new experimental work, which have enriched the manuscript and helped clarifying and amending the original interpretation of the findings.

Our response. We thank reviewer #1 for her/his positive comments and for recognizing that we have addressed all issues raised by reviewers.

Reviewer #2

General critique raised by reviewer #2: Authors have addressed all my suggestions and concerns.

Our response. We thank reviewer #2 for recognizing that we have addressed all suggestions and concerns that she/he raised.

Reviewer #3

General critique raised by reviewer #3: My comments have been adequately addressed.

Our response. We thank reviewer #3 for recognizing that we have adequately addressed her/his comments.